# Small Total-Cost Constraints in Contextual Bandits with Knapsacks, with Application to Fairness

**Evgenii Chzhen**      **Christophe Giraud**
Université Paris-Saclay, CNRS, Laboratoire de mathématiques d'Orsay, 91405, Orsay, France
`{evgenii.chzhen, christophe.giraud}@universite-paris-saclay.fr`

**Zhen Li**
BNP Paribas Corporate and Institutional Banking, 20 boulevard des Italiens, 75009 Paris, France
`zhen.li@bnpparibas.com`

**Gilles Stoltz**
Université Paris-Saclay, CNRS, Laboratoire de mathématiques d'Orsay, 91405, Orsay, France
HEC Paris, 78351 Jouy-en-Josas, France
`gilles.stoltz@universite-paris-saclay.fr, stoltz@hec.fr`

## Abstract

We consider contextual bandit problems with knapsacks [CBwK], a problem where at each round, a scalar reward is obtained and vector-valued costs are suffered. The learner aims to maximize the cumulative rewards while ensuring that the cumulative costs are lower than some predetermined cost constraints. We assume that contexts come from a continuous set, that costs can be signed, and that the expected reward and cost functions, while unknown, may be uniformly estimated— a typical assumption in the literature. In this setting, total cost constraints had so far to be at least of order $T^{3/4}$, where $T$ is the number of rounds, and were even typically assumed to depend linearly on $T$. We are however motivated to use CBwK to impose a fairness constraint of equalized average costs between groups: the budget associated with the corresponding cost constraints should be as close as possible to the natural deviations, of order $\sqrt{T}$. To that end, we introduce a dual strategy based on projected-gradient-descent updates, that is able to deal with total-cost constraints of the order of $\sqrt{T}$ up to poly-logarithmic terms. This strategy is more direct and simpler than existing strategies in the literature. It relies on a careful, adaptive, tuning of the step size.

## 1  Setting, literature review, and main contributions

We consider contextual bandits with knapsacks [CBwK], a setting where at each round $t \geqslant 1$, the learner, after observing some context $\boldsymbol{x}_t \in \mathcal{X}$, where $\mathcal{X} \subseteq \mathbb{R}^n$, picks an action $a_t \in \mathcal{A}$ in a finite set $\mathcal{A}$. We do not impose the existence of a null-cost action. Contexts are independently drawn according to a distribution $\nu$. The learner may pick $a_t$ at random according to a probability distribution, denoted by $\boldsymbol{\pi}_t(\boldsymbol{x}_t) = \big(\pi_{t,a}(\boldsymbol{x}_t)\big)_{a \in \mathcal{A}}$ for consistency with the notion of policy defined later in Section 2. The action $a_t$ played leads to some scalar reward $r_t \in [0, 1]$ and some signed vector-valued cost $\boldsymbol{c}_t \in [-1, 1]^d$. Actually, $r_t$ and $\boldsymbol{c}_t$ are generated independently at random in a way such that the conditional expectations of $r_t$ and $\boldsymbol{c}_t$ given the past, $\boldsymbol{x}_t$, and $a_t$, equal $r(\boldsymbol{x}_t, a_t)$ and $\boldsymbol{c}(\boldsymbol{x}_t, a_t)$, respectively. We denoted here by $r : \mathcal{X} \times \mathcal{A} \to [0, 1]$ and $\boldsymbol{c} = (c_1, \ldots, c_d) : \mathcal{X} \times \mathcal{A} \to [-1, 1]^d$ the unknown expected-reward and expected-cost functions. The modeling and the estimation of $r$

> **BOX A: CONTEXTUAL BANDITS WITH KNAPSACKS [CBwK]**
>
> **Known parameters:** finite action set $\mathcal{A}$; context set $\mathcal{X} \subseteq \mathbb{R}^n$; number $T$ of rounds; vector of average cost constraints $\boldsymbol{B} \in [0,1]^d$
>
> **Unknown parameters:** context distribution $\nu$ on $\mathcal{X}$; scalar expected-reward function $r : \mathcal{X} \times \mathcal{A} \to [0,1]$ and vector-valued expected-cost function $\boldsymbol{c} : \mathcal{X} \times \mathcal{A} \to [-1,1]^d$, both enjoying some modeling allowing for estimation, see Section 2.2
>
> **For rounds** $t = 1, 2, 3, \ldots, T$:
>
> 1. Context $\boldsymbol{x}_t \sim \nu$ is drawn independently of the past;
> 2. Learner observes $\boldsymbol{x}_t$ and picks an action $a_t \in \mathcal{A}$, possibly at random according to a distribution $\boldsymbol{\pi}_t(\boldsymbol{x}_t) = \big(\pi_{t,a}(\boldsymbol{x}_t)\big)_{a \in \mathcal{A}}$;
> 3. Learner gets a reward $r_t \in [0,1]$ with conditional expectation $r(\boldsymbol{x}_t, a_t)$ and suffers constraints $\boldsymbol{c}_t$ with conditional expectations $\boldsymbol{c}(\boldsymbol{x}_t, a_t)$.
>
> **Goal:** Maximize $\displaystyle\sum_{t \leqslant T} r_t$ while ensuring $\displaystyle\sum_{t \leqslant T} \boldsymbol{c}_t \leqslant T\boldsymbol{B}$

and $\boldsymbol{c}$ are discussed in Section 2.2. The aim of the learner is to maximize the sum of rewards (or equivalently, to minimize the regret defined in Section 2) while controlling cumulative costs. More precisely, we denote by $\boldsymbol{B} = (B_1, \ldots, B_d) \in [0,1]^d$ the normalized (i.e., average per-instance) cost constraints, which may depend on the coordinates. The number $T$ of rounds is known, and the learner must play in a way such that $\boldsymbol{c}_1 + \ldots + \boldsymbol{c}_T \leqslant T\boldsymbol{B}$. The example of Section 2.1 illustrates how this setting allows for controlling costs in absolute values. The setting described is summarized in Box A.

**Overview of the literature review.** Contextual bandits with knapsacks [CBwK] is a setting that is a combination of the problems of contextual bandits (where only rewards are obtained, and no costs: see, among others, Lattimore and Szepesvári, 2020, Chapter 18 for a survey of this rich area) and of bandits with knapsacks (without contexts, as initially introduced by Badanidiyuru et al., 2013, 2018). The first approaches to CBwK (Badanidiyuru et al., 2014, Agrawal et al., 2016) relied on no specific modeling of the rewards and costs, and made the problem tractable by using as a benchmark a finite set of so-called static policies (but picking this set is uneasy, as noted by Agrawal and Devanur, 2016). Virtually all subsequent approaches to CBwK thus introduced structural assumptions in one way or the other. The simplest modelings are linear dependencies of expected rewards and costs on the contexts (Agrawal and Devanur, 2016) and a logistic conversion model (Li and Stoltz, 2022). The problem of CBwK attracted attention in recent past: we discuss and contrast the contributions by Chohlas-Wood et al. [2021], Slivkins et al. [2022], Han et al. [2022], Ai et al. [2022], Li and Stoltz [2022] after stating our main contributions.

**Overview of our main contributions.** Each contribution calls or allows for the next one.

**1.** This article revisits the CBwK approach by Chohlas-Wood et al. [2021] to a problem of fair spending of resources between groups while maximizing some total utility. Fairness constraints require to deal with signed costs and with possibly small total-cost constraints (typically close to $T^{1/2}$), while having no null-cost action at disposal.

**2.** We therefore provide a new CBwK strategy dealing with these three issues, the most significant being handling cost constraint $T\boldsymbol{B}$ as small as $T^{1/2+\varepsilon}$, breaking the state-of-the-art $T^{3/4}$ barrier for CBwK with continuous contexts, while preserving attractive computational efficiency.

**3.** This new strategy is a direct, simple, explicit, dual strategy, relying on projected-gradient-descent updates (instead of using existing general algorithms as subroutines). We may perform an ad hoc analysis. The latter leads to refined regret bounds, in terms of the norms of some optimal dual variables. We also discuss the optimality of the results achieved by offering a proof scheme for problem-dependent (not only minimax) regret lower bounds.

## 1.1 Detailed literature review

We now contrast our main contributions to the existing literature.

**CBwK and fairness: missing tools.** In the setting of Chohlas-Wood et al. [2021], there is a budget constraint on the total spendings on top of the wish of sharing out these spendings among groups (possibly favoring to some maximal extent some groups that are more in need). However,

Chohlas-Wood et al. [2021] incorporated the fairness constraints in the reward function (through some Lagrangian penalty) instead of doing so in the vector-cost function, which does not seem the most natural way to proceed. One reason is that (see a discussion below) the total cost constraints $T\boldsymbol{B}$ were so far typically assumed to be linear, which is undesirable in fairness applications, where one wishes that $T\boldsymbol{B}$ is sublinear—and actually, is sometimes as close as possible to the natural $\sqrt{T}$ deviations suffered even in a perfectly fair random allocation scheme. Also, there is no null-cost action and costs are signed—two twists to the typical settings of CBwK that only Slivkins et al. [2022] and our approach deal with, to the best of our knowledge. We describe our alternative modeling of this fairness problem in Section 2.1.

**Orders of magnitude for total-cost constraints $T\boldsymbol{B}$.** The literature so far mostly considered linear total-cost constraints $T\boldsymbol{B}$ (see, e.g., Han et al., 2022 or Ai et al., 2022 among recent references), with two series of exceptions: (i) the primal strategy by Li and Stoltz [2022] handling total-cost contraints of the order of $\sqrt{T}$ up to poly-logarithmic terms but critically assuming the finiteness of the context set $\mathcal{X}$; and (ii) the two-stage dual strategies of Agrawal and Devanur [2016], Han et al. [2022] handling $T^{3/4}$ total-cost constraints. These two-stage strategies use $\sqrt{T}$ preliminary rounds to learn some key hyperparameter $Z$ and then transform strategies that work with linear total-cost constraints into strategies that may handle total-cost constraints larger than $T^{3/4}$; see discussions after Lemma 2. In contrast, we provide a general theory of small cost constraints for continuous context sets, while obtaining similar regret bounds as the literature does: for some components $j$, we may have $T B_j \ll T^{3/4}$ (actually, any rate $T B_j \gg T^{1/2+\varepsilon}$ is suitable for a sublinear regret), while for other components $k$, the total-cost constraints $T B_k$ may be linearly large.

**Typical CBwK strategies: primal approaches suffer from severe limitations.** CBwK is typically solved through dual approaches, as the latter basically rely on learning a finite-dimensional parameter given by the optimal dual variables $\boldsymbol{\lambda}^\star$ (see the beginning of Section 3), while primal approaches rely on learning the distribution $\nu$ of the contexts in $\mathbb{L}^1$–distance. This may be achieved in some cases, like finiteness of the context set $\mathcal{X}$ (Li and Stoltz, 2022, Ai et al., 2022) or, at the cost of degraded regret bounds and under additional assumptions on $\nu$ when resorting to $\mathbb{L}^1$–density-estimation techniques (Ai et al., 2022). On top of these strong limitations, such primal approaches are also typically computationally inefficient as they require the computation of expectations over estimates of $\nu$.

**Typical CBwK strategies: dual approaches are modular and somewhat indirect.** Typical dual approaches in CBwK take two general forms. One, illustrated in Slivkins et al. [2022] (extending the non-contextual approach by Immorlica et al., 2019), takes a game-theoretic approach by identifying the primal-dual formulation (4) as some minimax equilibrium, which may be learned by separate regret minimization of a learner picking actions $a_t$ and an opponent picking dual variables $\boldsymbol{\lambda}_t$. The second approach (Agrawal and Devanur, 2016, Han et al., 2022) learns more directly the $\boldsymbol{\lambda}_t$ via some online convex optimization algorithm fed with the suffered costs $\boldsymbol{c}_t$; this second approach however requires, at the stage of picking $a_t$, a suitable bound $Z$ on the norms of the $\boldsymbol{\lambda}_t$.

The dual strategies discussed above are modular and indirect as they all consist of using as building blocks some general-purpose strategies. We rather propose a more direct dual approach, tailored to our needs. (We also believe that it is simpler and more elegant.) Our strategy picks the arm $a_t$ that maximizes the Lagrangian penalization of rewards by costs through the current $\boldsymbol{\lambda}_{t-1}$ (see Step 2 in Box B), as also proposed by Agrawal and Devanur [2016] (while Han et al., 2022 add some randomization to this step), and then, performs some direct, explicit, projected-gradient-descent update on $\boldsymbol{\lambda}_{t-1}$ to obtain $\boldsymbol{\lambda}_t$, with step size $\gamma$. We carefully and explicitly tune $\gamma$ (in a sequential fashion, via regimes) to achieve our goals. Such fine-tuning is more difficult to perform with approaches relying on the black-box use of subroutines given by existing general-purpose strategies.

**Regret (upper and lower) bounds typically proposed in the CBwK literature.** A final advantage of our more explicit strategy is that we master each piece of its analysis: while not needing a null-cost action, we provide refined regret bounds that go beyond the typical $(\textsc{opt}(r, \boldsymbol{c}, \boldsymbol{B}) / \min \boldsymbol{B})\sqrt{T}$ bound offered by the literature so far (see, among others, Agrawal and Devanur, 2016, Li and Stoltz, 2022, Han et al., 2022), where $\textsc{opt}(r, \boldsymbol{c}, \boldsymbol{B})$ denotes the expected reward achieved by the optimal static policy (see Section 2 for definitions). Namely, our bounds are of the order of $\|\boldsymbol{\lambda}^\star\|\sqrt{T}$, where $\boldsymbol{\lambda}^\star$ is the optimal dual variable; we relate the norm of the latter to quantities of the form $\big(\textsc{opt}(r, \boldsymbol{c}, \boldsymbol{B}) - \textsc{opt}(r, \boldsymbol{c}, \alpha\boldsymbol{B})\big) / \big((1 - \alpha) \min \boldsymbol{B}\big)$, for some $\alpha < 1$. Again, we may do so without the existence of a null-cost action, but when one such action is available, we may take $\alpha = 0$ in the interpretation of the bound. We also offer a proof scheme for lower bounds in Section 4 and

explain why $\|\boldsymbol{\lambda}^\star\|\sqrt{T}$ appears as the correct target. We compare therein our problem-dependent lower bound-approach to the minimax ones of Han et al. [2022] and Slivkins et al. [2022].

## 1.2 Outline

In Section 2, we further describe the problem of CBwK (by defining the regret and recalling how $r$ and $\boldsymbol{c}$ may be estimated) and state our motivating example of fairness. Section 3 is devoted to our new dual strategy, which we analyze first with a fixed step size $\gamma$, for which we move next to an adaptive version, and whose bounds we finally discuss. Section 4 offers a proof scheme for regret lower bounds and lists some limitations of our approach, mostly relative to optimality.

## 2 Further description of the setting, and statement of our motivating example

We define a static policy as a mapping $\boldsymbol{\pi} : \mathcal{X} \to \mathcal{P}(\mathcal{A})$, where $\mathcal{P}(\mathcal{A})$ denotes the set of probability distributions over $\mathcal{A}$. We denote by $\boldsymbol{\pi} = (\pi_a)_{a \in \mathcal{A}}$ the components of $\boldsymbol{\pi}$. We let $\mathbb{E}_{\boldsymbol{X} \sim \nu}$ indicate that the expectation is taken over the random variable $\boldsymbol{X}$ with distribution $\nu$. In the sequel, the inequalities $\leqslant$ or $\geqslant$ between vectors will mean pointwise satisfaction of the corresponding inequalities.

**Assumption 1.** *The contextual bandit problem with knapsacks* $(r, \boldsymbol{c}, \boldsymbol{B}', \nu)$ *is feasible if there exists a stationary policy* $\boldsymbol{\pi}$ *such that*

$$\mathbb{E}_{\boldsymbol{X} \sim \nu}\left[\sum_{a \in \mathcal{A}} \boldsymbol{c}(\boldsymbol{X}, a)\, \pi_a(\boldsymbol{X})\right] \leqslant \boldsymbol{B}'\,.$$

We denote the average cost constraints by $\boldsymbol{B}'$ in the assumption above as in Section 3, we will actually require feasibility for average cost constraints $\boldsymbol{B}' < \boldsymbol{B}$, not just for $\boldsymbol{B}$.

In a feasible problem $(r, \boldsymbol{c}, \boldsymbol{B}', \nu)$, the optimal static policy $\boldsymbol{\pi}^\star$ is defined as the policy $\boldsymbol{\pi} : \mathcal{X} \to \mathcal{P}(\mathcal{A})$ achieving

$$\text{OPT}(r, \boldsymbol{c}, \boldsymbol{B}') = \sup_{\boldsymbol{\pi}} \left\{ \mathbb{E}_{\boldsymbol{X} \sim \nu}\left[\sum_{a \in \mathcal{A}} r(\boldsymbol{X}, a)\, \pi_a(\boldsymbol{X})\right] \ : \ \mathbb{E}_{\boldsymbol{X} \sim \nu}\left[\sum_{a \in \mathcal{A}} \boldsymbol{c}(\boldsymbol{X}, a)\, \pi_a(\boldsymbol{X})\right] \leqslant \boldsymbol{B}' \right\}. \quad (1)$$

Maximizing the sum of the $r_t$ amounts to minimizing the regret $R_T = T\,\text{OPT}(r, \boldsymbol{c}, \boldsymbol{B}') - \sum_{t=1}^{T} r_t\,.$

### 2.1 Motivating example: fairness—equalized average costs between groups

This example is inspired from Chohlas-Wood et al. [2021] (who did not provide a strategy to minimize regret), and features a total budget constraint $T B_{\text{total}}$ together with fairness constraints on how the budget is spent among finitely many subgroups, whose set is denoted by $\mathcal{G}$. We assume that the contexts $\boldsymbol{x}_t$ include the group index $\text{gr}(\boldsymbol{x}_t) \in \mathcal{G}$, which means, in our setting, that the learner accesses this possibly sensitive attribute before making the predictions (the so-called awareness framework in the fairness literature). Each context–action pair $(\boldsymbol{x}, a) \in \mathcal{X} \times \mathcal{A}$, on top of leading to some expected reward $r(\boldsymbol{x}, a)$, also corresponds to some spendings $c_{\text{spd}}(\boldsymbol{x}, a)$. We recall that we denoted that $r_t$ and $c_t$ (here, this is a scalar for now) the individual payoffs and costs obtained at round $t$; they have conditional expectations $r(\boldsymbol{x}_t, a_t)$ and $c_{\text{spd}}(\boldsymbol{x}_t, a_t)$. We want to trade off the differences in average spendings among groups with the total reward achieved: larger differences lead to larger total rewards but generate feelings of inequity or of public money not optimally used. (Chohlas-Wood et al. [2021] consider a situation where unfavored groups should get slightly more spendings than favored groups.)

Formally, we denote by $\text{gr}(\boldsymbol{x}) \in \mathcal{G}$ the group to which a given context $\boldsymbol{x}$ belongs and introduce a tolerance factor $\tau$, which may depend on $T$. We issue a simplifying assumption: while the distribution $\nu$ of the contexts is complex and is unknown, the respective proportions $\gamma_g = \nu\{\boldsymbol{x} : \text{gr}(\boldsymbol{x}) = g\}$ of the sub-groups may be known.[1] The total-budget and fairness constraints then read:

$$\sum_{t=1}^{T} c_t \leqslant T B_{\text{total}} \qquad \text{and} \qquad \forall g \in \mathcal{G}, \qquad \left| \frac{1}{T\gamma_g} \sum_{t=1}^{T} c_t \, \mathbb{1}_{\{\text{gr}(\boldsymbol{x}_t) = g\}} - \frac{1}{T} \sum_{t=1}^{T} c_t \right| \leqslant \tau,$$

---

[1]This simplification is not unheard of in the fairness literature; see, for instance, Chzhen et al. [2021]. It amounts to having a reasonable knowledge of the breakdown of the population into subgroups. We use this assumption to have an easy rewriting of the fairness constraints in the setting of Box A. The knowledge of the $\gamma_g$ is key for determining the average-constraint vector $\boldsymbol{B}$.

which corresponds, in the setting of Box A, to the following vector-valued expected-cost function $\boldsymbol{c}$, with $d = 1 + 2|\mathcal{G}|$ components and with values in $[-1,1]^{1+2|\mathcal{G}|}$:   $\boldsymbol{c}(\boldsymbol{x}, a) =$

$$\Big( c_{\text{spd}}(\boldsymbol{x}, a), \ \big( c_{\text{spd}}(\boldsymbol{x}, a)\mathbb{1}_{\{\text{gr}(\boldsymbol{x})=g\}} - \gamma_g c_{\text{spd}}(\boldsymbol{x}, a), \ \gamma_g c_{\text{spd}}(\boldsymbol{x}, a) - c_{\text{spd}}(\boldsymbol{x}, a)\mathbb{1}_{\{\text{gr}(\boldsymbol{x})=g\}} \big)_{g \in \mathcal{G}} \Big)$$

as well as to the average-constraint vector $\boldsymbol{B} = \Big( B_{\text{total}}, \ \big( \gamma_g \, \tau, \ \gamma_g \, \tau \big)_{g \in \mathcal{G}} \Big)$.

The regimes we have in mind, and that correspond to the public-policy issues reported by Chohlas-Wood et al. [2021], are that the per-instance budget $B_{\text{total}}$ is larger than a positive constant, i.e., a constant fraction of the $T$ individuals may benefit from some costly action(s), while the fairness threshold $\tau$ must be small, and even, as small as possible. Because of central-limit-theorem fluctuations, a minimal value of the order of $1/\sqrt{T}$, or slightly larger (up to logarithmic terms, say), has to be considered for $\tau$. The salient point is that the components of $\boldsymbol{B}$ may therefore be of different orders of magnitude, some of them being as small as $1/\sqrt{T}$, up to logarithmic terms. More details on this example and numerical simulations about it are provided in Section 5 and Appendix G.

## 2.2 Modelings for, and estimation of, expected-reward and expected-cost functions

As is common in the literature (see Han et al., 2022 and Slivkins et al., 2022, who use the terminology of regression oracles), we sequentially estimate the functions $r$ and $\boldsymbol{c}$ and assume that we may do so in some uniform way, e.g., because of the underlying structure assumed. Note that the estimation procedure of Assumption 2 below does not force any specific choice of actions, it is able to exploit actions $a_t$ picked by the main strategy (this is what the "otherwise" is related to therein). We denote by $\widehat{r}_t : \mathcal{X} \times \mathcal{A} \to [0, 1]$ and $\widehat{\boldsymbol{c}}_t : \mathcal{X} \times \mathcal{A} \to [-1, 1]^d$ the estimations built based on $(\boldsymbol{x}_s, a_s, r_s, \boldsymbol{c}_s)_{s \leqslant t}$, for $t \geqslant 1$. We also assume that initial estimations $\widehat{r}_0$ and $\widehat{\boldsymbol{c}}_0$, based on no data, are available. We denote by $\mathbf{1}$ the column-vector $(1, \ldots, 1)$.

**Assumption 2.** *There exists an estimation procedure such that for all individual sequences of contexts $\boldsymbol{x}_1, \boldsymbol{x}_2, \ldots$ and of actions $a_1, a_2, \ldots$ played otherwise, there exist* known *error functions $\varepsilon_t : \mathcal{X} \times \mathcal{A} \times (0, 1] \to [0, 1]$, relying each on the pairs $(\boldsymbol{x}_s, a_s)_{s \leqslant t}$, where $t = 0, 1, 2, \ldots$, ensuring that for all $\delta \in (0, 1]$, with probability at least $1 - \delta$,*

$$\forall 0 \leqslant t \leqslant T, \ \forall \boldsymbol{x} \in \mathcal{X}, \ \forall a \in \mathcal{A}, \qquad \big| \widehat{r}_t(\boldsymbol{x}, a) - r(\boldsymbol{x}, a) \big| \leqslant \varepsilon_t(\boldsymbol{x}, a, \delta)$$

$$\text{and} \qquad \big| \widehat{\boldsymbol{c}}_t(\boldsymbol{x}, a) - \boldsymbol{c}(\boldsymbol{x}, a) \big| \leqslant \varepsilon_t(\boldsymbol{x}, a, \delta) \, \mathbf{1} \,,$$

*where we assume in addition that* $\quad \beta_{T,\delta} \overset{\text{def}}{=} \sum_{t=1}^{T} \varepsilon_{t-1}(\boldsymbol{x}_t, a_t, \delta) = O\bigg( \sqrt{T} \ln \frac{T}{\delta} \bigg)\,.$

This assumption is satisfied at least for the two modelings discussed below, which are both of the form: for known transfer functions $\boldsymbol{\varphi}$, link functions $\Phi$, and normalization function $\eta$, for some unknown finite-dimensional parameters $\boldsymbol{\mu}_\star, \boldsymbol{\theta}_{\star,1}, \ldots, \boldsymbol{\theta}_{\star,d}$,

$$\forall \boldsymbol{x} \in \mathcal{X}, \ \forall a \in \mathcal{A}, \qquad\qquad r(\boldsymbol{x}, a) = \eta_r(a, \boldsymbol{x}) \, \Phi_r\big( \boldsymbol{\varphi}_r(\boldsymbol{x}, a)^{\text{T}} \boldsymbol{\mu}_\star \big) \,,$$

$$\text{and for all } 1 \leqslant i \leqslant d, \qquad c_i(\boldsymbol{x}, a) = \eta_{c,i}(a, \boldsymbol{x}) \, \Phi_{c,i}\big( \boldsymbol{\varphi}_{c,i}(\boldsymbol{x}, a)^{\text{T}} \boldsymbol{\theta}_{\star,i} \big) \,.$$

See also the exposition by Han et al. [2022, Section 3.3].

Based on Assumption 2 and given the ranges $[0, 1]$ for rewards and $[-1, 1]^d$ for costs, we may now define the following (clipped) upper- and lower-confidence bounds : given a confidence level $1 - \delta$ where $\delta \in (0, 1]$, for all $t \leqslant T$, for all $\boldsymbol{x} \in \mathcal{X}$ and $a \in \mathcal{A}$,

$$\widehat{r}_{\delta,t}^{\,\text{ucb}}(\boldsymbol{x}, a) \overset{\text{def}}{=} \text{clip}\big[ \widehat{r}_t(\boldsymbol{x}, a) + \varepsilon_t(\boldsymbol{x}, a, \delta) \big]_0^1 \ \text{ and } \ \widehat{\boldsymbol{c}}_{\delta,t}^{\,\text{lcb}}(\boldsymbol{x}, a) \overset{\text{def}}{=} \text{clip}\big[ \widehat{\boldsymbol{c}}_t(\boldsymbol{x}, a) - \varepsilon_t(\boldsymbol{x}, a, \delta)\mathbf{1} \big]_{-1}^1 \,, \quad (2)$$

where we define clipping by $\text{clip}[x]_\ell^u = \min\big\{ \max\{x, \ell\}, u \big\}$ for a scalar value $x \in \mathbb{R}$ and lower and upper bounds $\ell$ and $u$, and apply clipping component-wise to vectors. Under Assumption 2, we have that for all $\delta \in (0, 1]$, with probability at least $1 - \delta$,

$$\forall 0 \leqslant t \leqslant T, \ \forall \boldsymbol{x} \in \mathcal{X}, \ \forall a \in \mathcal{A}, \qquad r(\boldsymbol{x}, a) \leqslant \widehat{r}_{\delta,t}^{\,\text{ucb}}(\boldsymbol{x}, a) \leqslant r(\boldsymbol{x}, a) + 2\varepsilon_t(\boldsymbol{x}, a, \delta)$$

$$\text{and} \qquad \boldsymbol{c}(\boldsymbol{x}, a) - 2\varepsilon_t(\boldsymbol{x}, a, \delta)\mathbf{1} \leqslant \widehat{\boldsymbol{c}}_{\delta,t}^{\,\text{lcb}}(\boldsymbol{x}) \leqslant \boldsymbol{c}(\boldsymbol{x}, a) \,. \qquad (3)$$

Doing so, we have optimistic estimates of rewards (they are larger than the actual expected rewards) and of costs (they are smaller than the actual expected costs) for all actions $a$, while for actions $a_{t+1}$ played, and only these, we will also use the control from the other side, which will lead to manageable sums of $\varepsilon_t(\boldsymbol{x}_{t+1}, a_{t+1}, \delta)$, given the control on $\beta_{T,\delta}$ by Assumption 2.

**Modeling 1: Linear modeling.** Agrawal and Devanur [2016] (see also Li and Stoltz, 2022, Appendix E) consider the case of a linear modeling where $\eta_r = \eta_1 = \ldots = \eta_d \equiv 1$ and $\Phi$ is the identity (together with assumptions on the range of $\varphi$ and on the norms of $\boldsymbol{\mu}_\star$ and the $\boldsymbol{\theta}_{\star,i}$). The LinUCB strategy by Abbasi-Yadkori et al. [2011] may be slightly adapted (to take care of the existence of a transfer function $\varphi$ depending on the actions taken) to show that Assumption 2 holds; see Li and Stoltz [2022, Appendix E, Lemma 3].

**Modeling 2: Conversion model based on logistic bandits.** Li and Stoltz [2022, Sections 1-5] discuss the case where $\Phi(x) = 1/(1 + \mathrm{e}^{-x})$ for rewards and costs. The Logistic-UCB1 algorithm of Faury et al. [2020] (see also an earlier version by Filippi et al., 2010) may be slightly adapted (again, because of $\varphi$) to show that Assumption 2 holds; see Li and Stoltz [2022, Lemma 1 in Appendix B, as well as Appendix C].

## 3 New, direct, dual strategy for CBwK

Our strategy heavily relies on the following equalities between the primal and the dual values of the problem considered. The underlying justifications are typically omitted in the literature (see, e.g., Slivkins et al., 2022, end of Section 2); we detail them carefully in Appendix A. We introduce dual variables $\boldsymbol{\lambda} \in [0, +\infty)^d$, denote by $\langle \cdot, \cdot \rangle$ the inner product in $\mathbb{R}^d$, and we have, provided that the problem $(r, \boldsymbol{c}, \boldsymbol{B}', \nu)$ is feasible for some $\boldsymbol{B}' < \boldsymbol{B}$:

$$
\begin{aligned}
\mathrm{OPT}(r, \boldsymbol{c}, \boldsymbol{B}) &= \sup_{\boldsymbol{\pi}:\mathcal{X}\to\mathcal{P}(\mathcal{A})} \inf_{\boldsymbol{\lambda}\geqslant\mathbf{0}} \mathbb{E}_{\boldsymbol{X}\sim\nu}\left[\sum_{a\in\mathcal{A}} r(\boldsymbol{X},a)\,\pi_a(\boldsymbol{X}) + \left\langle \boldsymbol{\lambda},\, \boldsymbol{B} - \sum_{a\in\mathcal{A}} \boldsymbol{c}(\boldsymbol{X},a)\,\pi_a(\boldsymbol{X}) \right\rangle\right] \\
&= \min_{\boldsymbol{\lambda}\geqslant\mathbf{0}} \sup_{\boldsymbol{\pi}:\mathcal{X}\to\mathcal{P}(\mathcal{A})} \mathbb{E}_{\boldsymbol{X}\sim\nu}\left[\sum_{a\in\mathcal{A}} \pi_a(\boldsymbol{X})\Big(r(\boldsymbol{X},a) - \big\langle \boldsymbol{c}(\boldsymbol{X},a) - \boldsymbol{B},\, \boldsymbol{\lambda}\big\rangle\Big)\right] \\
&= \min_{\boldsymbol{\lambda}\geqslant\mathbf{0}} \mathbb{E}_{\boldsymbol{X}\sim\nu}\left[\max_{a\in\mathcal{A}}\Big\{r(\boldsymbol{X},a) - \big\langle \boldsymbol{c}(\boldsymbol{X},a) - \boldsymbol{B},\, \boldsymbol{\lambda}\big\rangle\Big\}\right].
\end{aligned}
\tag{4}
$$

We denote by $\boldsymbol{\lambda}_{\boldsymbol{B}}^\star$ a vector $\boldsymbol{\lambda} \geqslant \mathbf{0}$ achieving the minimum in the final equality. Virtually all previous contributions to (contextual) bandits with knapsacks learned the dual optimal $\boldsymbol{\lambda}^\star$ in some way; see the discussion in Main contribution #3 in Section 1. Our strategy is stated in Box B. A high-level description is that it replaces (Step 2) the ideal dual problem above by an empirical version, with estimates of $\boldsymbol{r}$ and $\boldsymbol{C}$, with the expectation over $\boldsymbol{X} \sim \nu$ replaced by a point evaluation at $\boldsymbol{x}_t$, and that it sequentially chooses some $\boldsymbol{\lambda}_t$ based on projected-gradient-descent steps (Step 4 uses the subgradient of the function in $\boldsymbol{\lambda}$ defined in Step 2). The gradient-descent steps are performed based on fixed step sizes $\gamma$ in Section 3.1, and Section 3.2 will explain how to sequentially set the step sizes based on data. We also take some margin $b\mathbf{1}$ on $\boldsymbol{B}$.

---

**BOX B: PROJECTED GRADIENT DESCENT FOR CBwK WITH FIXED STEP SIZE**

**Inputs:** step size $\gamma > 0$; number of rounds $T$; confidence level $1 - \delta$; margin $b$ on the average constraints; estimation procedure and error functions $\varepsilon_t$ of Assumption 2; optimistic estimates (2)

**Initialization:** $\boldsymbol{\lambda}_0 = \mathbf{0}$; initial estimates $\widehat{r}_{\delta,0}^{\,\mathrm{ucb}}$ and $\widehat{c}_{\delta,0}^{\,\mathrm{lcb}}$

**For rounds** $t = 1, 2, 3, \ldots, T$:

    1. Observe the context $\boldsymbol{x}_t$;

    2. Pick an action $a_t \in \mathcal{A}$ achieving

$$
\max_{a\in\mathcal{A}}\Big\{\widehat{r}_{\delta,t-1}^{\,\mathrm{ucb}}(\boldsymbol{x}_t,a) - \big\langle \widehat{\boldsymbol{c}}_{\delta,t-1}^{\,\mathrm{lcb}}(\boldsymbol{x}_t,a) - (\boldsymbol{B} - b\mathbf{1}),\, \boldsymbol{\lambda}_{t-1}\big\rangle\Big\};
$$

    3. Observe the payoff $r_t$ and the costs $\boldsymbol{c}_t$;

    4. Compute $\boldsymbol{\lambda}_t = \Big(\boldsymbol{\lambda}_{t-1} + \gamma\big(\widehat{\boldsymbol{c}}_{\delta,t-1}^{\,\mathrm{lcb}}(\boldsymbol{x}_t,a_t) - (\boldsymbol{B} - b\mathbf{1})\big)\Big)_+$;

    5. Compute the estimates $\widehat{r}_{\delta,t}^{\,\mathrm{ucb}}$ and $\widehat{\boldsymbol{c}}_{\delta,t}^{\,\mathrm{lcb}}$.

---

## 3.1 Analysis for a fixed step size $\gamma$

The strategy of Box B is analyzed through two lemmas. We let $\min \boldsymbol{v} = \min\{v_1, \ldots, v_d\} > 0$ denote the smallest component of a vector $\boldsymbol{v} \in (0, +\infty)^d$ and take a margin $b$ in $(0, \min \boldsymbol{B})$. We assume that the problem is feasible for some $\boldsymbol{B}' < \boldsymbol{B} - b\mathbf{1}$, and denote by $\boldsymbol{\lambda}_{\boldsymbol{B}-b\mathbf{1}}^{\star}$ the vector achieving the minimum in (4) when the per-round cost constraint is $\boldsymbol{B} - b\mathbf{1}$. We use $\| \cdot \|$ for the Euclidean norm. The bounds of Lemmas 1 and 2 critically rely on $\|\boldsymbol{\lambda}_{\boldsymbol{B}-b\mathbf{1}}^{\star}\|$, which we bound and interpret below in Section 3.3. We denote

$$\Upsilon_{T,\delta} = \max\left\{ \beta_{T,\delta/4}, \ 2\sqrt{dT \ln \frac{T^2}{\delta/4}}, \ \sqrt{2T \ln \frac{2(d+1)T}{\delta/4}} \right\}.$$

**Lemma 1.** *Fix $\delta \in (0,1)$ and $0 < b < \min \boldsymbol{B}$. If Assumption 2 holds and if the CBwK problem is feasible for some $\boldsymbol{B}' < \boldsymbol{B} - b\mathbf{1}$, the Box B strategy run with $\delta/4$ ensures that the costs satisfy, with probability at least $1 - \delta$,*

$$\forall\, 1 \leqslant t \leqslant T, \qquad \left\| \left( \sum_{\tau=1}^{t} \boldsymbol{c}_\tau - t\,(\boldsymbol{B} - b\mathbf{1}) \right)_+ \right\| \leqslant \frac{1 + 3\|\boldsymbol{\lambda}_{\boldsymbol{B}-b\mathbf{1}}^{\star}\|}{\gamma} + 20\sqrt{d}\,\Upsilon_{T,\delta}\,.$$

**Lemma 2.** *Fix $\delta \in (0,1)$ and $0 < b < \min \boldsymbol{B}$. Under the same assumptions as in Lemma 1 and on the same event with probability at least $1 - \delta$ as therein, the regret of the Box B strategy run with $\delta/4$ satisfies*

$$\forall\, 1 \leqslant t \leqslant T, \qquad R_t \leqslant \|\boldsymbol{\lambda}_{\boldsymbol{B}-b\mathbf{1}}^{\star}\| \left( t\,b\sqrt{d} + 6\Upsilon_{T,\delta} \right) + 36\gamma\sqrt{d}\left( \Upsilon_{T,\delta} \right)^2 + 8\ln \frac{T^2}{\delta/4}\,.$$

**Ideal choice of $\gamma$.** The margin $b$ will be taken proportional to the high-probability deviation provided by Lemma 1, divided by $T$. Now, the bounds of Lemmas 1 and 2 are respectively of the order

$$Tb \sim \left( 1 + \|\boldsymbol{\lambda}_{\boldsymbol{B}-b\mathbf{1}}^{\star}\| \right)/\gamma + \sqrt{T} \qquad \text{and} \qquad \|\boldsymbol{\lambda}_{\boldsymbol{B}-b\mathbf{1}}^{\star}\|\left( Tb + \sqrt{T} \right) + \gamma T\,,$$

again, up to logarithmic factors. To optimize each of these bounds, we therefore wish that $b$ be of order $1/\sqrt{T}$ and that $\gamma$ be of the order of $\|\boldsymbol{\lambda}_{\boldsymbol{B}-b\mathbf{1}}^{\star}\|/\sqrt{T}$. Of course, this wish stumbles across the same issues of ignoring a beforehand bound on $\|\boldsymbol{\lambda}_{\boldsymbol{B}-b\mathbf{1}}^{\star}\|$, exactly like, e.g., in Agrawal and Devanur [2016], Han et al. [2022] (where this bound is denoted by $Z$). We were however able to overcome this issue in a satisfactory way, i.e., by obtaining the ideal bounds implied above through adaptive choices of $\gamma$, discussed in Section 3.2.

**Suboptimal choices of $\gamma$.** When $\gamma$ is badly set, e.g., $\gamma = 1/\sqrt{T}$, the order of magnitude of the regret remains larger than $\|\boldsymbol{\lambda}_{\boldsymbol{B}-b\mathbf{1}}^{\star}\|\, Tb$ and the cost deviations to $T(\boldsymbol{B} - b\mathbf{1})$ become of the larger order $\|\boldsymbol{\lambda}_{\boldsymbol{B}-b\mathbf{1}}^{\star}\|\sqrt{T}$, which dictates a larger choice for $T\,b$. Namely, if in addition $\|\boldsymbol{\lambda}_{\boldsymbol{B}-b\mathbf{1}}^{\star}\|$ is bounded in some worst-case way by $1/\min \boldsymbol{B}$ (see Section 3.3), the margin $Tb$ is set of the order of $\sqrt{T}/\min \boldsymbol{B}$. But of course, this margin must remain smaller than the total-cost constraints $T\boldsymbol{B}$. This is the fundamental source of the typical $\min \boldsymbol{B} \geqslant T^{-1/4}$ constraints on the average cost constraints $\boldsymbol{B}$ observed in the literature: see, e.g., Agrawal and Devanur, 2016, Han et al., 2022, where the hyperparameter $Z$ plays exactly the role of $\|\boldsymbol{\lambda}_{\boldsymbol{B}-b\mathbf{1}}^{\star}\|$.

*Proof sketch.* Lemmas 1 and 2 are proved in detail in Appendix B; the two proofs are similar and we sketch here the one for Lemma 1. The derivations below hold on the intersection of four events of probability at least $1 - \delta/4$ each: one for Assumption 2, one for the application of the Hoeffding-Azuma inequality, and two for the application of a version of Bernstein's inequality. In this sketch of proof, $\lesssim$ denotes inequalities holding up to small terms.

By the Hoeffding-Azuma inequality and by Assumption 2, the sum of the $\boldsymbol{c}_\tau - (\boldsymbol{B} - b\mathbf{1})$ is not much larger than the sum of the $\Delta\widehat{\boldsymbol{c}}_\tau = \widehat{\boldsymbol{c}}_{\delta/4,\tau-1}^{\mathrm{lcb}}(\boldsymbol{x}_\tau, a_\tau) - (\boldsymbol{B} - b\mathbf{1})$. We bound the latter by noting that by definition, $\boldsymbol{\lambda}_\tau \geqslant \boldsymbol{\lambda}_{\tau-1} + \gamma\Delta\widehat{\boldsymbol{c}}_\tau$, so that after telescoping,

$$\left\| \left( \sum_{\tau=1}^{t} \boldsymbol{c}_\tau - t(\boldsymbol{B} - b\mathbf{1}) \right)_+ \right\| \lesssim \left\| \left( \sum_{\tau=1}^{t} \Delta\widehat{\boldsymbol{c}}_\tau \right)_+ \right\| \leqslant \frac{\|\boldsymbol{\lambda}_t\|}{\gamma} \leqslant \frac{\|\boldsymbol{\lambda}_{\boldsymbol{B}-b\mathbf{1}}^{\star}\|}{\gamma} + \frac{\|\boldsymbol{\lambda}_{\boldsymbol{B}-b\mathbf{1}}^{\star} - \boldsymbol{\lambda}_t\|}{\gamma}\,. \quad (5)$$

We are thus left with bounding the $\|\boldsymbol{\lambda}_{\boldsymbol{B}-b\mathbf{1}}^{\star} - \boldsymbol{\lambda}_t\|$. The classical bound in (projected) gradient-descent analyses (see, e.g., Zinkevich, 2003) reads:

$$\|\boldsymbol{\lambda}_\tau - \boldsymbol{\lambda}^\star_{\boldsymbol{B}-b\mathbf{1}}\|^2 - \|\boldsymbol{\lambda}_{\tau-1} - \boldsymbol{\lambda}^\star_{\boldsymbol{B}-b\mathbf{1}}\|^2 \leqslant 2\gamma\langle\Delta\widehat{\boldsymbol{c}}_\tau,\,\boldsymbol{\lambda}_{\tau-1}-\boldsymbol{\lambda}^\star_{\boldsymbol{B}-b\mathbf{1}}\rangle + \gamma^2 \overbrace{\|\Delta\widehat{\boldsymbol{c}}_\tau\|^2}^{\leqslant 4d}.$$

The estimates of Assumption 2, and the definition of $a_t$ as the argument of some maximum, entail

$$\langle\Delta\widehat{\boldsymbol{c}}_\tau,\,\boldsymbol{\lambda}_{\tau-1}-\boldsymbol{\lambda}^\star_{\boldsymbol{B}-b\mathbf{1}}\rangle = \langle\widehat{\boldsymbol{c}}^{\,\mathrm{lcb}}_{\delta/4,\tau-1}(\boldsymbol{x}_\tau,a_\tau)-(\boldsymbol{B}-b\mathbf{1}),\,\boldsymbol{\lambda}_{\tau-1}-\boldsymbol{\lambda}^\star_{\boldsymbol{B}-b\mathbf{1}}\rangle \lesssim g_\tau(\boldsymbol{\lambda}^\star_{\boldsymbol{B}-b\mathbf{1}})-g_\tau(\boldsymbol{\lambda}_{\tau-1})\,,$$

$$\text{where}\qquad g_\tau(\boldsymbol{\lambda}) = \max_{a\in\mathcal{A}}\Big\{r(\boldsymbol{x}_\tau,a) - \langle\boldsymbol{c}(\boldsymbol{x}_\tau,a) - (\boldsymbol{B}-b\mathbf{1}),\,\boldsymbol{\lambda}\rangle\Big\}.$$

We also introduce $\Lambda_t = \sqrt{d}\,\max_{1\leqslant\tau\leqslant t}\|\boldsymbol{\lambda}^\star_{\boldsymbol{B}-b\mathbf{1}} - \boldsymbol{\lambda}_{\tau-1}\|$. We sum up the bounds above, use telescoping, and get

$$\|\boldsymbol{\lambda}_t - \boldsymbol{\lambda}^\star_{\boldsymbol{B}-b\mathbf{1}}\|^2 \lesssim \|\boldsymbol{\lambda}^\star_{\boldsymbol{B}-b\mathbf{1}}\|^2 + 4d\,\gamma^2 t + 2\gamma\underbrace{\sum_{\tau\leqslant t} g_\tau(\boldsymbol{\lambda}^\star_{\boldsymbol{B}-b\mathbf{1}}) - g_\tau(\boldsymbol{\lambda}_{\tau-1})}_{\lesssim 0+3\Lambda_t\sqrt{t\ln(T/\delta)}}\,, \tag{6}$$

where the $\lesssim$ of the sum in the right-hand side comes from a version of Bernstein's inequality, the fact that $\mathbb{E}\big[g_\tau(\boldsymbol{\lambda}^\star_{\boldsymbol{B}-b\mathbf{1}})\big] = \mathrm{OPT}(r,\boldsymbol{c},\boldsymbol{B}-b\mathbf{1})$ is larger than the conditional expectation of $g_\tau(\boldsymbol{\lambda}_{\tau-1})$. Since (6) holds for all $t$, we may then show by induction that for some constants $\square$ and $\triangle$,

$$\|\boldsymbol{\lambda}_t - \boldsymbol{\lambda}^\star_{\boldsymbol{B}-b\mathbf{1}}\| \lesssim \square\|\boldsymbol{\lambda}^\star_{\boldsymbol{B}-b\mathbf{1}}\| + \triangle\gamma\Upsilon_T\,,$$

and we conclude by substituting this inequality into (5). $\qquad\square$

### 3.2 Meta-stategy with adaptive step size $\gamma$

As indicated after the statements of Lemmas 1 and 2, we want a (meta-)strategy adaptively setting $\gamma$ to learn $\|\boldsymbol{\lambda}^\star_{\boldsymbol{B}-b\mathbf{1}}\|$. We do so via a doubling trick using the Box B strategy as a routine and step sizes of the form $\gamma_k = 2^k/\sqrt{T}$ in regimes $k \geqslant 0$; the resulting meta-strategy is stated in Box C (a more detailed pseudo-code is provided in Appendix C). As the theorem below will reveal, the margin $b$ therein can be picked based solely on $T$, $\delta$, and $d$.

---

**BOX C: PROJECTED GRADIENT DESCENT FOR CBwK WITH ADAPTIVE STEP SIZE**

**Inputs:** number of rounds $T$; confidence level $1-\delta$; margin $b$ on the average constraints; estimation procedure and error functions $\varepsilon_t$ of Assumption 2; optimistic estimates (2)

**Initialization:** $T_0 = 1$; sequence $\gamma_k = 2^k/\sqrt{T}$ of step sizes;
sequence $M_{T,\delta,k} = 4\sqrt{T} + 20\sqrt{d}\,\Upsilon_{T,\delta/(k+2)^2}$ of cost deviations

**For regimes** $k = 0, 1, 2, \ldots$:

- Take a fresh start of the Box B strategy with step size $\gamma_k$, risk level $\delta/\big(4(k+2)^2\big)$, and margin $b$;

- Run it for $t \geqslant T_k$ until $\left\|\left(\sum_{\tau=T_k}^{t}\boldsymbol{c}_\tau - (t - T_k + 1)(\boldsymbol{B}-b\mathbf{1})\right)_+\right\| > M_{T,\delta,k}$;

- Stop regime $k$, and move to regime $k+1$ for the next round, i.e., $T_{k+1} = t+1$.

---

Bandit problems containing a null-cost action (which is a typical assumption in the CBwK literature) are feasible for all cost constraints $\boldsymbol{B}' \geqslant \mathbf{0}$; the assumption of $(\boldsymbol{B}-2b_T\mathbf{1})$–feasibility in Theorem 1 below is therefore a mild assumption. Apart from this, the following theorem only imposes that the average cost constraints are larger than $1/\sqrt{T}$, up to poly-logarithmic terms. We let $\widetilde{\mathcal{O}}$ denote orders of magnitude in $T$ up to poly-logarithmic terms.

For $x > 0$, we denote by $\mathrm{ilog}\,x = \lceil\log_2 x\rceil$ the upper integer part of the base-2 logarithm of $x$.

**Theorem 1.** *Fix $\delta \in (0,1)$ and let Assumption 2 hold. Consider the Box C meta-strategy, run with confidence level $1-\delta$ and with margin $b_T = (1 + \mathrm{ilog}\,T)\big(M_{T,\delta,\mathrm{ilog}\,T} + 2\sqrt{d}\big)/T = \widetilde{\mathcal{O}}\big(1/\sqrt{T}\big)$. If the average cost constraints $\boldsymbol{B}$ are such that $\boldsymbol{B} \geqslant 2b_T\mathbf{1}$ and if the problem $(r,\boldsymbol{c},\boldsymbol{B}-2b_T\mathbf{1},\nu)$ is feasible, then this meta-strategy satisfies, with probability at least $1-\delta$:*

$$R_T \leqslant \widetilde{\mathcal{O}}\Big(\big(1 + \|\boldsymbol{\lambda}^{\star}_{\boldsymbol{B}-b_T\mathbf{1}}\|\big)\sqrt{T}\Big) \qquad \text{and} \qquad \sum_{t\leqslant T}\boldsymbol{c}_t \leqslant T\boldsymbol{B}\,,$$

*where a fully closed-form, non-asymptotic, regret bound is stated in* (19) *in Appendix C.*

A complete proof may be found in Appendix C. Its structure is the following; all statements hold with high probability. We denote by $\ell_k$ the realized lengths of the regime. First, the number of regimes is bounded by noting that if regime $K = \mathrm{ilog}\,\|\boldsymbol{\lambda}^{\star}_{\boldsymbol{B}-b_T\mathbf{1}}\|$ is achieved, then by Lemma 1 and the choice of $M_{T,\delta,K}$, the stopping condition of regime $K$ will never be met; thus, at most $K+1$ regimes take place. We also prove that $\mathrm{ilog}\,\|\boldsymbol{\lambda}^{\star}_{\boldsymbol{B}-b_T\mathbf{1}}\| \leqslant K_T = \mathrm{ilog}\,T$. Second, on each regime $k$, the difference of cumulated costs to $\ell_k(\boldsymbol{B}-b_T\mathbf{1})$ is smaller than $M_{T,\delta,k}$ by design, so that the total cumulative costs are smaller than $T(\boldsymbol{B}-b_T\mathbf{1})$ plus something of the order of $(K_T+1)\,M_{T,\delta,K_T}$, which is a quantity that only depends on $T$, $\delta$, $d$, and not on the unknown $\|\boldsymbol{\lambda}^{\star}_{\boldsymbol{B}-b_T\mathbf{1}}\|$, and that we take for $T\,b_T$. Third, we similarly sum the regret bounds of Lemma 2 over the regimes: keeping in mind that $b_T = \widetilde{\mathcal{O}}\big(\sqrt{T}\big)$, we have sums of the form

$$\|\boldsymbol{\lambda}^{\star}_{\boldsymbol{B}-b_T\mathbf{1}}\| \sum_{k\leqslant K}\big(b_T\ell_k + \sqrt{T}\big) \leqslant \|\boldsymbol{\lambda}^{\star}_{\boldsymbol{B}-b_T\mathbf{1}}\|\,\widetilde{\mathcal{O}}\big(\sqrt{T}\big) \quad \text{and} \quad \sum_{k\leqslant K}\gamma_k T \leqslant \frac{2^{K+1}}{\sqrt{T}} \leqslant \frac{4\|\boldsymbol{\lambda}^{\star}_{\boldsymbol{B}-b_T\mathbf{1}}\|}{\sqrt{T}}\,.$$

### 3.3 Discussion of the obtained regret bounds

The regret bound of Theorem 1 features a multiplicative factor $\|\boldsymbol{\lambda}^{\star}_{\boldsymbol{B}-b_T\mathbf{1}}\|$ instead of the typical factor $\mathrm{OPT}(r,\boldsymbol{c},\boldsymbol{B})/\min\boldsymbol{B}$ proposed by the literature (see, e.g., Agrawal and Devanur, 2016, Li and Stoltz, 2022, Han et al., 2022). In view of the lower bound proof scheme discussed in Section 4, this bound $\|\boldsymbol{\lambda}^{\star}_{\boldsymbol{B}-b_T\mathbf{1}}\|\sqrt{T}$ seems the optimal bound. We thus now provide bounds on $\|\boldsymbol{\lambda}^{\star}_{\boldsymbol{B}-b_T\mathbf{1}}\|$ that have some natural interpretation. We recall that feasibility was defined in Assumption 1. The elementary proofs of Lemma 3 and Corollary 1 are based on (4) and may be found in Appendix D.

**Lemma 3.** *Let $b \in [0, \min\boldsymbol{B})$ and let $\mathbf{0} \leqslant \widetilde{\boldsymbol{B}} < \boldsymbol{B} - b\mathbf{1}$. If the contextual bandit problem with knapsacks $(r, \boldsymbol{c}, \boldsymbol{B}', \nu)$ is feasible for some $\boldsymbol{B}' < \widetilde{\boldsymbol{B}}$, then*

$$\|\boldsymbol{\lambda}^{\star}_{\boldsymbol{B}-b\mathbf{1}}\| \leqslant \frac{\mathrm{OPT}(r,\boldsymbol{c},\boldsymbol{B}-b\mathbf{1}) - \mathrm{OPT}(r,\boldsymbol{c},\widetilde{\boldsymbol{B}})}{\min(\boldsymbol{B}-b\mathbf{1}-\widetilde{\boldsymbol{B}})}\,.$$

**Corollary 1.** *When there exists a null-cost action, $\|\boldsymbol{\lambda}^{\star}_{\boldsymbol{B}-b\mathbf{1}}\| \leqslant 2\dfrac{\mathrm{OPT}(r,\boldsymbol{c},\boldsymbol{B}) - \mathrm{OPT}(r,\boldsymbol{c},\mathbf{0})}{\min\boldsymbol{B}}$ for all $b \in [0, \min\boldsymbol{B}/2]$.*

The bounds provided by Lemma 3 and Corollary 1 must be discussed in each specific case. They are null (as expected) when the average cost constraints $\boldsymbol{B}$ are large enough so that costs do not constrain the choice of actions; this was not the case with the typical $\mathrm{OPT}(r,\boldsymbol{c},\boldsymbol{B})/\min\boldsymbol{B}$ bounds of the literature. Another typical example is the following.

**Example 1.** *Consider a problem featuring a baseline action $a_{\mathrm{null}}$ with some positive expected rewards and no cost, and additional actions with larger rewards and scalar expected costs $c(\boldsymbol{x}, a) \geqslant \alpha > 0$. Denote by $B$ the average cost constraint. Then actions $a \neq a_{\mathrm{null}}$ may only be played at most $B/\alpha$ times on average, and therefore, $\mathrm{OPT}(r, c, B) - \mathrm{OPT}(r, c, 0) \leqslant B/\alpha$. In particular, the bound stated in Corollary 1 may be further upper bounded by $1/(2\alpha)$, which is fully independent of $T$ and $B$; i.e., a $\sqrt{T}$–regret only is suffered in Theorem 1.*

## 4 Optimality of the upper bounds, and limitations

The main limitations to our work are relative, in our eyes, to the optimality of the bounds achieved— up to one limitation, already discussed in Section 2.1: our fairness example is intrinsically limited to the awareness set-up, where the learner has access to the group index before making the predictions.

**A proof scheme for problem-dependent lower bounds.** Results offering lower bounds on the regret for CBwK are scarce in the literature. They typically refer to some minimax regret, and either do so by indicating that lower bounds for CBwK must be larger in a minimax sense than the ones obtained for non-contextual bandits with knapsacks (see, e.g., Agrawal and Devanur, 2016, Li and Stoltz, 2022), or by leveraging a minimax regret lower bound for a subroutine of the strategy used (Slivkins et al., 2022). We rather focus on problem-dependent regret bounds but only offer a proof scheme, to be made more formal—see Appendix E.1. Interestingly, this proof scheme follows closely the

analysis of a primal strategy performed in Appendix F. First, an argument based on the law of the iterated logarithm shows the necessity of a margin $b_T$ of order $1/\sqrt{T}$ to satisfy the total $TB$ cost constraints. This imposes that only $T\text{OPT}(r, c, B - b_T 1)$ is targeted, thus the regret is at least of order

$$T\big(\text{OPT}(r, c, B) - \text{OPT}(r, c, B - b_T 1)\big) + \sqrt{T}, \qquad \text{i.e.,} \qquad \big(1 + \|\lambda_B^\star\|\big)\sqrt{T}\,.$$

This matches the bound of Theorem 1, but with $\lambda_{B-b_T 1}^\star$ replaced by $\lambda_B^\star$.

We now turn to the list of limitations pertaining to the optimality of our bounds.

**Limitation 1: Large constants.** First, we must mention that in the bounds of Lemmas 1 and 2, we did not try to optimize the constants but targeted readability. The resulting values in Theorem 1, namely, the recommended choice of $b_T$ and the closed-form bound (19) on the regret, therefore involve constants that are much larger than needed.

**Limitation 2: Not capturing possibly fast regret rates in some cases.** Second, a careful inspection of our proof scheme for lower bounds shows that it only works in the absence of a null-cost action. When such a null-cost action exists and costs are non-negative, Example 1 shows that costly actions are played at most $B/\alpha$ times on average. If the null-cost action has also a null reward, the costly actions are the only ones generating some stochasticity, thus, we expect that the $\sqrt{T}$ factors (coming, among others, from a version of Bernstein's inequality) be replaced by $\sqrt{B/\alpha}$ factors. As a consequence, in the setting of Example 1, bounds growing much slower than $\sqrt{T}$ should be achievable, see the discussion in Appendix E.1.

**Limitation 3: $\sqrt{T}$ regret not captured in case of softer constraints.** For the primal strategy discussed in Appendix F, we could prove a general $\sqrt{T}$ regret bound in the case where total costs smaller than $TB + \widetilde{\mathcal{O}}\big(\sqrt{T}\big)$ are allowed. We were unable to do so for our dual strategy.

## 5   Brief overview of the numerical experiments performed

In Appendix G, we implement a specific version of the motivating example of Section 2.1, as proposed by Chohlas-Wood et al. [2021] in the public repository `https://github.com/stanford-policylab/learning-to-be-fair`. Fairness costs together with spendings related to rideshare assistance or transportation vouchers are considered. We report below the performance of the strategies considered only in terms of average rewards and rideshare costs, and for the fairness threshold $\tau = 10^{-7}$. The Box B strategies with fixed step sizes may fail to control the budget when $\gamma$ is too large: we observe this for $\gamma = 0.01$. We see overall a tradeoff between larger average rewards and larger average costs. The Box C strategy navigates between three regimes, $\gamma = 0.01$ to start (regime $k = 0$), then $\gamma = 0.02$ (regime $k = 1$), and finally $\gamma = 0.04$ (regime $k = 2$), which it sticks to. It overall adaptively finds a decent tuning of $\gamma$.

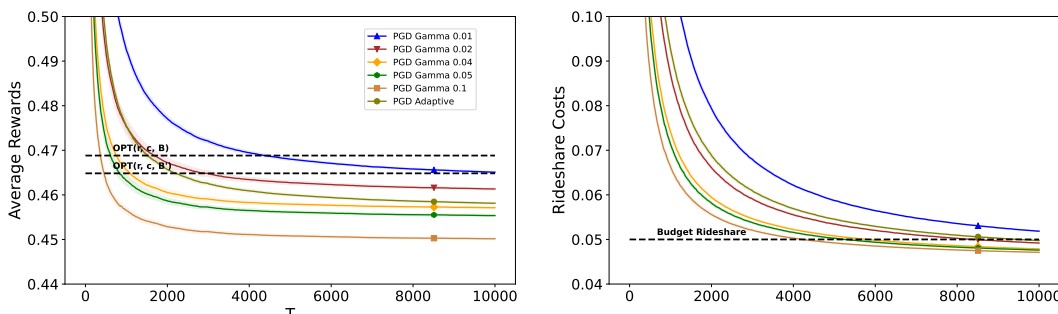

## Acknowledgments and Disclosure of Funding

Evgenii Chzhen, Christophe Giraud, Zhen Li, and Gilles Stoltz have no direct funding to acknowledge other than the salaries paid by their employers, CNRS, Université Paris-Saclay, BNP Paribas, and HEC Paris. They have no competing interests to declare.

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

## A  Proof of the strong duality (4)

In this section, we explain why the equalities (4) hold when the problem $(r, \boldsymbol{c}, \boldsymbol{B}', \nu)$ is feasible for some $\boldsymbol{B}' < \boldsymbol{B}$. We restate first these inequalities for the convenience of the reader:

$$
\mathrm{OPT}(r, \boldsymbol{c}, \boldsymbol{B}) = \sup_{\boldsymbol{\pi} : \mathcal{X} \to \mathcal{P}(\mathcal{A})} \ \inf_{\boldsymbol{\lambda} \geqslant \boldsymbol{0}} \ \mathbb{E}_{\boldsymbol{X} \sim \nu} \left[ \sum_{a \in \mathcal{A}} r(\boldsymbol{X}, a) \, \pi_a(\boldsymbol{X}) + \Big\langle \boldsymbol{\lambda}, \ \boldsymbol{B} - \sum_{a \in \mathcal{A}} \boldsymbol{c}(\boldsymbol{X}, a) \, \pi_a(\boldsymbol{X}) \Big\rangle \right]
$$

$$
\overset{(\star)}{=} \min_{\boldsymbol{\lambda} \geqslant \boldsymbol{0}} \ \sup_{\boldsymbol{\pi} : \mathcal{X} \to \mathcal{P}(\mathcal{A})} \ \mathbb{E}_{\boldsymbol{X} \sim \nu} \left[ \sum_{a \in \mathcal{A}} \pi_a(\boldsymbol{X}) \Big( r(\boldsymbol{X}, a) - \big\langle \boldsymbol{c}(\boldsymbol{X}, a) - \boldsymbol{B}, \boldsymbol{\lambda} \big\rangle \Big) \right]
$$

$$
= \min_{\boldsymbol{\lambda} \geqslant \boldsymbol{0}} \ \mathbb{E}_{\boldsymbol{X} \sim \nu} \left[ \max_{a \in \mathcal{A}} \Big\{ r(\boldsymbol{X}, a) - \big\langle \boldsymbol{c}(\boldsymbol{X}, a) - \boldsymbol{B}, \boldsymbol{\lambda} \big\rangle \Big\} \right].
$$

We deal here with a linear program, with primal variables in functional space $\mathcal{X} \to \mathbb{R}^{\mathcal{A}}$ and dual variables in the finite-dimensional space $\mathbb{R}^d$.

The first and third equalities are straightforward. The first equality holds because in $\mathrm{OPT}(r, \boldsymbol{c}, \boldsymbol{B})$, we want to consider only policies with expected cost smaller than or equal to $\boldsymbol{B}$; the infimum over $\boldsymbol{\lambda} \geqslant \boldsymbol{0}$ equals $-\infty$ for policies with expected costs strictly larger than $\boldsymbol{B}$ and is achieved at $\boldsymbol{\lambda} = 0$ otherwise. The third inequality follows from identifying that for a given $\boldsymbol{\lambda}$, the best policy may be defined pointwise as the argument of the maximum written in the expectation. Thus, only the middle equality $(\star)$ deserves a proof. We obtain it by applying a general theorem of strong duality (which requires feasibility for slightly smaller cost constraints).

**Statement of a general theorem of strong duality.**  We restate a result extracted from the monograph by Luenberger [1969]. It relies on the dual functional $\varphi$, whose expression we recall below.

**Theorem 2** (stated as Theorem 1 in Section 8.6, page 224 in Luenberger, 1969). *Let $f$ be a real-valued convex functional defined on a convex subset $\Omega$ of a vector space $X$, and let $G$ be a convex mapping of $X$ into a normed space $Z$. Suppose there exists an $x_1$ such that $G(x_1) < \theta$ and that $\mu_0 = \inf \big\{ f(x) : \ G(x) \leqslant \theta, \ x \in \Omega \big\}$ is finite. Then*

$$
\inf_{\substack{G(x) \leqslant \theta \\ x \in \Omega}} f(x) = \max_{z^\star \geqslant \theta} \varphi(z^\star)
$$

*and the maximum on the right is achieved by some $z_0^\star \geqslant \theta$.*

The dual function $\varphi$ is defined as follows (Luenberger, 1969, pages 215, 223, and 224, which we quote). We denote by $P$ a positive convex cone $P \subset Z$, and let $P^\star = \big\{ z^\star \in Z^\star : \ \forall z \in P, \ \langle z, z^\star \rangle \geqslant 0 \big\}$ be its corresponding positive cone in the dual space $Z^\star$. The dual functional is defined, for each element $z^\star \in P^\star$, as

$$
\varphi(z^\star) = \inf_{x \in \Omega} \big\{ f(x) + \langle G(x), z^\star \rangle \big\}.
$$

**Application.**  To prove the middle equality $(\star)$, we apply Theorem 2 with the vector space $X$ of all functions $\mathcal{X} \to \mathbb{R}^{\mathcal{A}}$, its convex subset $\Omega$ formed by functions $\mathcal{X} \to \mathcal{P}(\mathcal{A})$, and note that the function $f$ is actually linear in our situation:

$$
f(\pi) = -\mathbb{E}_{\boldsymbol{X} \sim \nu} \left[ \sum_{a \in \mathcal{A}} r(\boldsymbol{X}, a) \, \pi_a(\boldsymbol{X}) \right].
$$

(We take a $-$ sign to match the form of the minimization problem over $\pi$ in Theorem 2.) We take $Z = \mathbb{R}^d = Z^\star$ with positive cones $P = P^\star = [0, +\infty)^d$. The mapping $G$ is a linear mapping from $X$ to $Z = \mathbb{R}^d$:

$$
G(\pi) = \mathbb{E}_{\boldsymbol{X} \sim \nu} \left[ \sum_{a \in \mathcal{A}} \boldsymbol{c}(\boldsymbol{X}, a) \, \pi_a(\boldsymbol{X}) \right] - \boldsymbol{B},
$$

and we take $\theta = \boldsymbol{0}$. We have that $\mu_0 = \mathrm{OPT}(r, \boldsymbol{c}, \boldsymbol{B})$ is indeed finite. We note that the condition "$(r, \boldsymbol{c}, \boldsymbol{B}', \nu)$ is feasible for some $\boldsymbol{B}' < \boldsymbol{B}$" is required to apply the theorem. The result follows from noting that we exactly have, for $z = \boldsymbol{\lambda} \in P^\star$,

$$
\varphi(\boldsymbol{\lambda}) = \inf_{\boldsymbol{\pi} : \mathcal{X} \to \mathcal{P}(\mathcal{A})} \ \mathbb{E}_{\boldsymbol{X} \sim \nu} \left[ -\sum_{a \in \mathcal{A}} r(\boldsymbol{X}, a) \, \pi_a(\boldsymbol{X}) + \sum_{a \in \mathcal{A}} \big\langle \boldsymbol{c}(\boldsymbol{X}, a) - \boldsymbol{B}, \boldsymbol{\lambda} \big\rangle \pi_a(\boldsymbol{X}) \right].
$$

## B  Proofs of Lemmas 1 and 2

In both proofs, we will use the following deviation inequalities, holding on an event $\mathcal{E}_{\text{H-Az}} \cap \mathcal{E}_\beta$ of probability at least $1 - \delta/2$, where the event $\mathcal{E}_\beta$ is defined in Assumption 2 with a confidence level $1 - \delta/4$ and the event $\mathcal{E}_{\text{H-Az}}$ is defined below: on $\mathcal{E}_{\text{H-Az}} \cap \mathcal{E}_\beta$,

$$\forall\, 1 \leqslant t \leqslant T, \qquad \sum_{\tau=1}^t \boldsymbol{c}_\tau \leqslant \left(\alpha_{t,\delta/4} + 2\beta_{t,\delta/4}\right)\mathbf{1} + \sum_{\tau=1}^t \widehat{\boldsymbol{c}}^{\,\text{lcb}}_{\delta/4,\tau-1}(\boldsymbol{x}_\tau, a_\tau) \tag{7}$$

$$\text{and} \qquad \sum_{\tau=1}^t r_\tau \geqslant -\left(\alpha_{t,\delta/4} + 2\beta_{t,\delta/4}\right) + \sum_{\tau=1}^t \widehat{r}^{\,\text{ucb}}_{\delta/4,\tau-1}(\boldsymbol{x}_\tau, a_\tau), \tag{8}$$

where $\beta_{t,\delta/4}$ is also defined in Assumption 2 and

$$\alpha_{t,\delta/4} = \sqrt{2t \ln \frac{2(d+1)T}{\delta/4}}\,.$$

These inequalities come, on the one hand, by the Hoeffding-Azuma inequality applied $(d+1)T$ times on the range $[-1, 1]$: it ensures that on an event $\mathcal{E}_{\text{H-Az}}$ with probability at least $1 - \delta/4$, for all $1 \leqslant t \leqslant T$,

$$\left\| \sum_{\tau=1}^t \boldsymbol{c}_\tau - \sum_{\tau=1}^t \boldsymbol{c}(\boldsymbol{x}_\tau, a_\tau) \right\|_\infty \leqslant \alpha_{t,\delta/4}\,\mathbf{1} \qquad \text{and} \qquad \left| \sum_{\tau=1}^t r_\tau - \sum_{\tau=1}^t r(\boldsymbol{x}_\tau, a_\tau) \right| \leqslant \alpha_{t,\delta/4}$$

(where we denoted by $\|\cdot\|_\infty$ the supremum norm). On the other hand, Assumption 2 and the clipping (3) entail, in particular, that on an event $\mathcal{E}_\beta$ of probability at least $1 - \delta/4$, for all $1 \leqslant t \leqslant T$,

$$\sum_{\tau=1}^t \boldsymbol{c}(\boldsymbol{x}_\tau, a_\tau) \leqslant \sum_{\tau=1}^t \left(\widehat{\boldsymbol{c}}^{\,\text{lcb}}_{\delta/4,\tau-1}(\boldsymbol{x}_\tau, a_\tau) + 2\varepsilon_{\tau-1}(\boldsymbol{x}_\tau, a_\tau, \delta/4)\,\mathbf{1}\right) = 2\beta_{t,\delta/4}\,\mathbf{1} + \sum_{\tau=1}^t \widehat{\boldsymbol{c}}^{\,\text{lcb}}_{\delta/4,\tau-1}(\boldsymbol{x}_\tau, a_\tau)$$

$$\text{and (similarly)} \qquad \sum_{\tau=1}^t r(\boldsymbol{x}_\tau, a_\tau) \geqslant -2\beta_{t,\delta/4} + \sum_{\tau=1}^t \widehat{r}^{\,\text{ucb}}_{\delta/4,\tau-1}(\boldsymbol{x}_\tau, a_\tau)\,.$$

### B.1  Proof of Lemma 1

For $\tau \geqslant 1$, by definition of $\boldsymbol{\lambda}_\tau$ in Box B,

$$\boldsymbol{\lambda}_\tau - \boldsymbol{\lambda}_{\tau-1} = \left(\boldsymbol{\lambda}_{\tau-1} + \gamma\left(\widehat{\boldsymbol{c}}^{\,\text{lcb}}_{\delta/4,\tau-1}(\boldsymbol{x}_\tau, a_\tau) - (\boldsymbol{B} - b\mathbf{1})\right)\right)_+ - \boldsymbol{\lambda}_{\tau-1}$$

$$\geqslant \gamma\left(\widehat{\boldsymbol{c}}^{\,\text{lcb}}_{\delta/4,\tau-1}(\boldsymbol{x}_\tau, a_\tau) - (\boldsymbol{B} - b\mathbf{1})\right).$$

For $t \geqslant 1$, as $\boldsymbol{\lambda}_0 = \mathbf{0}$ and $\boldsymbol{\lambda}_t \geqslant \mathbf{0}$, we get, after telescoping and taking non-negative parts,

$$\boldsymbol{\lambda}_t \geqslant \gamma\left(\sum_{\tau=1}^t \left(\widehat{\boldsymbol{c}}^{\,\text{lcb}}_{\delta/4,\tau-1}(\boldsymbol{x}_\tau, a_\tau) - (\boldsymbol{B} - b\mathbf{1})\right)\right)_+,$$

$$\text{thus} \qquad \left\| \left(\sum_{\tau=1}^t \left(\widehat{\boldsymbol{c}}^{\,\text{lcb}}_{\delta/4,\tau-1}(\boldsymbol{x}_\tau, a_\tau) - (\boldsymbol{B} - b\mathbf{1})\right)\right)_+ \right\| \leqslant \frac{\|\boldsymbol{\lambda}_t\|}{\gamma}\,. \tag{9}$$

Up to the deviation terms (7), $\|\boldsymbol{\lambda}_t\|/\gamma$ bounds how larger the cost constraints till round $t$ are from $t(\boldsymbol{B} - b\mathbf{1})$. Most of the rest of the proof, namely, the Steps 1–4 below, thus focus on upper bounding the $\|\boldsymbol{\lambda}_t\|$, while Step 5 will collect all bounds together and conclude.

**Step 1: Gradient-descent analysis.**  We introduce

$$\Delta\widehat{\boldsymbol{c}}_\tau = \widehat{\boldsymbol{c}}^{\,\text{lcb}}_{\delta/4,\tau-1}(\boldsymbol{x}_\tau, a_\tau) - (\boldsymbol{B} - b\mathbf{1}) \tag{10}$$

and prove the following deterministic inequality: for all $1 \leqslant t \leqslant T$,

$$\forall\, \boldsymbol{\lambda} \geqslant \mathbf{0}, \qquad \|\boldsymbol{\lambda}_t - \boldsymbol{\lambda}\|^2 \leqslant \|\boldsymbol{\lambda}\|^2 + 4d\,\gamma^2 t + 2\gamma \sum_{\tau=1}^t \langle \Delta\widehat{\boldsymbol{c}}_\tau,\, \boldsymbol{\lambda}_{\tau-1} - \boldsymbol{\lambda}\rangle\,. \tag{11}$$

To do so, we proceed as is classical in (projected) gradient-descent analyses; see, e.g., Zinkevich [2003]. Namely, for all $1 \leqslant \tau \leqslant T$,

$$2\langle -\Delta\widehat{\boldsymbol{c}}_\tau, \, \boldsymbol{\lambda}_{\tau-1} - \boldsymbol{\lambda}\rangle = \frac{1}{\gamma}\Big(\|\boldsymbol{\lambda}_{\tau-1} - \boldsymbol{\lambda}\|^2 + \big\|\gamma\Delta\widehat{\boldsymbol{c}}_\tau\big\|^2 - \big\|\boldsymbol{\lambda}_{\tau-1} - \boldsymbol{\lambda} + \gamma\Delta\widehat{\boldsymbol{c}}_\tau\big\|^2\Big)$$

$$= \gamma\big\|\Delta\widehat{\boldsymbol{c}}_\tau\big\|^2 + \frac{1}{\gamma}\Big(\|\boldsymbol{\lambda}_{\tau-1} - \boldsymbol{\lambda}\|^2 - \big\|\boldsymbol{\lambda}_{\tau-1} + \gamma\Delta\widehat{\boldsymbol{c}}_\tau - \boldsymbol{\lambda}\big\|^2\Big)$$

$$\leqslant \gamma\big\|\Delta\widehat{\boldsymbol{c}}_\tau\big\|^2 + \frac{1}{\gamma}\Big(\|\boldsymbol{\lambda}_{\tau-1} - \boldsymbol{\lambda}\|^2 - \|\boldsymbol{\lambda}_\tau - \boldsymbol{\lambda}\|^2\Big),$$

where the inequality comes from the definition $\boldsymbol{\lambda}_\tau = \big(\boldsymbol{\lambda}_{\tau-1} + \gamma\Delta\widehat{\boldsymbol{c}}_\tau\big)_+$ and the fact that

$$\forall\, x \in \mathbb{R}, \ \ \forall\, y \geqslant 0, \qquad \big|(x)_+ - y\big| \leqslant |x - y|$$

(which may be proved by distinguishing the cases $x \leqslant 0$ and $x \geqslant 0$).

We note that

$$\boldsymbol{B} - b\mathbf{1} \in [0,1]^d \quad \text{and} \quad \widehat{\boldsymbol{c}}^{\,\text{lcb}}_{\delta/4,\tau-1}(\boldsymbol{x}_\tau, a_\tau) \in [-1,1]^d, \qquad \text{so that} \qquad \big\|\Delta\widehat{\boldsymbol{c}}_\tau\big\|^2 \leqslant 4d.$$

Collecting all bounds above, we get, after summation and telescoping,

$$\sum_{\tau=1}^{t}\langle -\Delta\widehat{\boldsymbol{c}}_\tau, \, \boldsymbol{\lambda}_{\tau-1} - \boldsymbol{\lambda}\rangle \leqslant 2d\,\gamma t + \frac{1}{2\gamma}\Big(\|\boldsymbol{\lambda}_0 - \boldsymbol{\lambda}\|^2 - \|\boldsymbol{\lambda}_t - \boldsymbol{\lambda}\|^2\Big).$$

Rearranging and substituting $\boldsymbol{\lambda}_0 = \mathbf{0}$ yields the claimed inequality (11).

**Step 2: Relating estimated costs (and rewards) to true conditional expectations.** In this part, we upper bound the right-hand side of (11) by showing that on the event $\mathcal{E}_\beta$,

$$\forall\, \boldsymbol{\lambda} \geqslant \mathbf{0}, \quad \forall\, 1 \leqslant t \leqslant T,$$

$$\sum_{\tau=1}^{t}\langle \Delta\widehat{\boldsymbol{c}}_\tau, \, \boldsymbol{\lambda}_{\tau-1} - \boldsymbol{\lambda}\rangle \leqslant 2\big(1 + \|\boldsymbol{\lambda}\|_1\big)\beta_{t,\delta/4} + \sum_{\tau=1}^{t}\big(g_\tau(\boldsymbol{\lambda}) - g_\tau(\boldsymbol{\lambda}_{\tau-1})\big), \tag{12}$$

$$\text{where} \qquad g_\tau(\boldsymbol{\lambda}) = \max_{a \in \mathcal{A}}\Big\{r(\boldsymbol{x}_\tau, a) - \big\langle \boldsymbol{c}(\boldsymbol{x}_\tau, a) - (\boldsymbol{B} - b\mathbf{1}), \, \boldsymbol{\lambda}\big\rangle\Big\}$$

and where we recall that $\|\cdot\|_1$ denotes the $\ell_1$–norm.

Adding and subtracting $\widehat{r}^{\,\text{ucb}}_{\delta/4,\tau-1}(\boldsymbol{x}_\tau, a_\tau)$, here, we deal with

$$\sum_{\tau=1}^{t}\langle \Delta\widehat{\boldsymbol{c}}_\tau, \, \boldsymbol{\lambda}_{\tau-1} - \boldsymbol{\lambda}\rangle = \sum_{\tau=1}^{t}\Big(\widehat{r}^{\,\text{ucb}}_{\delta/4,\tau-1}(\boldsymbol{x}_\tau, a_\tau) - \big\langle \widehat{\boldsymbol{c}}^{\,\text{lcb}}_{\delta/4,\tau-1}(\boldsymbol{x}_\tau, a_\tau) - (\boldsymbol{B} - b\mathbf{1}), \, \boldsymbol{\lambda}\big\rangle\Big)$$

$$- \sum_{\tau=1}^{t}\Big(\widehat{r}^{\,\text{ucb}}_{\delta/4,\tau-1}(\boldsymbol{x}_\tau, a_\tau) - \big\langle \widehat{\boldsymbol{c}}^{\,\text{lcb}}_{\delta/4,\tau-1}(\boldsymbol{x}_\tau, a_\tau) - (\boldsymbol{B} - b\mathbf{1}), \, \boldsymbol{\lambda}_{\tau-1}\big\rangle\Big).$$

Now, for each $\tau$, by (3) and the fact that $\boldsymbol{\lambda} \geqslant \mathbf{0}$,

$$\widehat{r}^{\,\text{ucb}}_{\delta/4,\tau-1}(\boldsymbol{x}_\tau, a_\tau) - \big\langle \widehat{\boldsymbol{c}}^{\,\text{lcb}}_{\delta/4,\tau-1}(\boldsymbol{x}_\tau, a_\tau) - (\boldsymbol{B} - b\mathbf{1}), \, \boldsymbol{\lambda}\big\rangle$$

$$\leqslant r(\boldsymbol{x}_\tau, a_\tau) + 2\varepsilon_{\tau-1}(\boldsymbol{x}_\tau, a_\tau, \delta/4) - \big\langle \boldsymbol{c}(\boldsymbol{x}_\tau, a_\tau) - (\boldsymbol{B} - b\mathbf{1}), \, \boldsymbol{\lambda}\big\rangle + 2\varepsilon_{\tau-1}(\boldsymbol{x}_\tau, a_\tau, \delta/4)\|\boldsymbol{\lambda}\|_1$$

$$\leqslant g_\tau(\boldsymbol{\lambda}) + 2\big(1 + \|\boldsymbol{\lambda}\|_1\big)\varepsilon_{\tau-1}(\boldsymbol{x}_\tau, a_\tau, \delta/4).$$

On the other hand, by definition of $a_\tau$ for the equality, and then by the other inequalities in (3), for each $\tau$, and the fact that $\boldsymbol{\lambda}_{\tau-1} \geqslant \mathbf{0}$,

$$\widehat{r}^{\,\text{ucb}}_{\delta/4,\tau-1}(\boldsymbol{x}_\tau, a_\tau) - \big\langle \widehat{\boldsymbol{c}}^{\,\text{lcb}}_{\delta/4,\tau-1}(\boldsymbol{x}_\tau, a_\tau) - (\boldsymbol{B} - b\mathbf{1}), \, \boldsymbol{\lambda}_{\tau-1}\big\rangle \tag{13}$$

$$= \max_{a \in \mathcal{A}}\Big\{\widehat{r}^{\,\text{ucb}}_{\delta/4,\tau-1}(\boldsymbol{x}_\tau, a) - \big\langle \widehat{\boldsymbol{c}}^{\,\text{lcb}}_{\delta/4,\tau-1}(\boldsymbol{x}_\tau, a) - (\boldsymbol{B} - b\mathbf{1}), \, \boldsymbol{\lambda}_{\tau-1}\big\rangle\Big\}$$

$$\geqslant \max_{a \in \mathcal{A}}\Big\{r(\boldsymbol{x}_\tau, a) - \big\langle \boldsymbol{c}(\boldsymbol{x}_\tau, a) - (\boldsymbol{B} - b\mathbf{1}), \, \boldsymbol{\lambda}_{\tau-1}\big\rangle\Big\} = g_\tau(\boldsymbol{\lambda}_{\tau-1}).$$

Collecting the two series of bounds concludes this part.

**Step 3: Application of a Bernstein-Freedman inequality.** We recall that we denoted by $\boldsymbol{\lambda}^\star_{\boldsymbol{B}-b\mathbf{1}}$ the optimal dual variable in (4) for $\mathrm{OPT}(r, \boldsymbol{c}, \boldsymbol{B} - b\mathbf{1})$; it exists because we assumed feasibility of a problem with average cost constraints $\boldsymbol{B}' < \boldsymbol{B} - b\mathbf{1}$.

We now upper bound the sum appearing in the right hand side of (12) at $\boldsymbol{\lambda} = \boldsymbol{\lambda}^\star_{\boldsymbol{B}-b\mathbf{1}}$ by showing that on an event $\mathcal{E}_{\text{Bern-}\boldsymbol{c}}$ of probability at least $1 - \delta/4$, for all $1 \leqslant t \leqslant T$,

$$\sum_{\tau=1}^t \big(g_\tau(\boldsymbol{\lambda}^\star_{\boldsymbol{B}-b\mathbf{1}}) - g_\tau(\boldsymbol{\lambda}_{\tau-1})\big) \leqslant (1 + 2\Lambda_t)\sqrt{2t \ln \frac{T^2}{\delta/4}} + 2K_t \ln \frac{T^2}{\delta/4}, \tag{14}$$

where $\qquad \Lambda_t = \max_{1 \leqslant \tau \leqslant t} \|\boldsymbol{\lambda}^\star_{\boldsymbol{B}-b\mathbf{1}} - \boldsymbol{\lambda}_{\tau-1}\|_1 \qquad$ and $\qquad K_t = 4\big(1 + \|\boldsymbol{\lambda}^\star_{\boldsymbol{B}-b\mathbf{1}}\|_1 + 2d\gamma t\big)$.

We will do so by applying a version of the Bernstein-Freedman inequality for martingales stated in Cesa-Bianchi et al. [2005, Corollary 16] involving the sum of the conditional variances (and not only a deterministic bound thereon); it is obtained via peeling based on the classic version of Bernstein's inequality (Freedman, 1975). We restate it here for the convenience of the reader (after applying some simple boundings).

**Lemma 4** (a version of the Bernstein-Freedman inequality by Cesa-Bianchi et al., 2005). *Let $X_1, X_2, \ldots$ be a martingale difference with respect to the filtration $\mathcal{F} = (\mathcal{F}_s)_{s \geqslant 0}$ and with increments bounded in absolute values by $K$. For all $t \geqslant 1$, let*

$$\mathfrak{S}_t = \sum_{\tau=1}^t \mathbb{E}\big[X_\tau^2 \,\big|\, \mathcal{F}_{\tau-1}\big]$$

*denote the sum of the conditional variances of the first $t$ increments. Then, for all $\delta \in (0, 1)$ and all $t \geqslant 1$, with probability at least $1 - \delta$,*

$$\sum_{\tau=1}^t X_\tau \leqslant \sqrt{2\mathfrak{S}_t \ln \frac{t}{\delta}} + 2K \ln \frac{t}{\delta}.$$

We introduce, for all $\boldsymbol{\lambda} \geqslant \mathbf{0}$, the common expectation of the $g_\tau(\boldsymbol{\lambda})$, namely,

$$G(\boldsymbol{\lambda}) = \mathbb{E}\big[g_\tau(\boldsymbol{\lambda})\big] = \mathbb{E}_{\boldsymbol{X} \sim \nu}\left[\max_{a \in \mathcal{A}}\Big\{r(\boldsymbol{X}, a) - \big\langle \boldsymbol{c}(\boldsymbol{X}, a) - (\boldsymbol{B} - b\mathbf{1}), \boldsymbol{\lambda}\big\rangle\Big\}\right],$$

and consider the martingale increments

$$X_\tau = \big(g_\tau(\boldsymbol{\lambda}^\star_{\boldsymbol{B}-b\mathbf{1}}) - g_\tau(\boldsymbol{\lambda}_{\tau-1})\big) - \big(G(\boldsymbol{\lambda}^\star_{\boldsymbol{B}-b\mathbf{1}}) - G(\boldsymbol{\lambda}_{\tau-1})\big).$$

As $\boldsymbol{B} - b\mathbf{1} \in [0, 1]^d$ and $\boldsymbol{c}$ takes values in $[-1, 1]^d$, for all $\boldsymbol{x} \in \mathcal{X}$, all $a \in \mathcal{A}$, and all $\boldsymbol{v} \in \mathbb{R}^d$, the quantities $r(\boldsymbol{x}, a) - \big\langle \boldsymbol{c}(\boldsymbol{x}, a) - (\boldsymbol{B} - b\mathbf{1}), \boldsymbol{v}\big\rangle$ take absolute values smaller than $1 + 2\|\boldsymbol{v}\|_1$. Using that a difference of maxima is smaller than the maximum of the differences, we get, in particular,

$$\big|g_\tau(\boldsymbol{\lambda}^\star_{\boldsymbol{B}-b\mathbf{1}}) - g_\tau(\boldsymbol{\lambda}_{\tau-1})\big| \leqslant 1 + 2\|\boldsymbol{\lambda}^\star_{\boldsymbol{B}-b\mathbf{1}} - \boldsymbol{\lambda}_{\tau-1}\|_1 \quad \text{a.s.} \tag{15}$$

and $\qquad \big|G(\boldsymbol{\lambda}^\star_{\boldsymbol{B}-b\mathbf{1}}) - G(\boldsymbol{\lambda}_{\tau-1})\big| \leqslant 1 + 2\|\boldsymbol{\lambda}^\star_{\boldsymbol{B}-b\mathbf{1}} - \boldsymbol{\lambda}_{\tau-1}\|_1$.

Now, given the update step in Step 3 of Box B, we have the deterministic bound $\|\boldsymbol{\lambda}_t\|_1 \leqslant 2d\gamma t$ for all $1 \leqslant t \leqslant T$. Therefore, by a triangle inequality, the martingale increments are bounded in absolute values by $K_t$, as

$$2\Big(1 + 2\|\boldsymbol{\lambda}^\star_{\boldsymbol{B}-b\mathbf{1}}\|_1 + 1 + 2\max_{\tau \leqslant t}\|\boldsymbol{\lambda}_{\tau-1}\|_1\Big) \leqslant 4\big(1 + \|\boldsymbol{\lambda}^\star_{\boldsymbol{B}-b\mathbf{1}}\|_1 + 2d\gamma t\big) = K_t.$$

The conditional variance of $X_\tau$ is smaller than the squared half-width of the conditional range (Popoviciu's inequality on variances); in particular, (15) thus entails

$$\mathfrak{S}_t \leqslant \sum_{\tau=1}^t \big(1 + 2\|\boldsymbol{\lambda}^\star_{\boldsymbol{B}-b\mathbf{1}} - \boldsymbol{\lambda}_{\tau-1}\|_1\big)^2 \leqslant (1 + 2\Lambda_t)^2 t.$$

We now get the claimed inequalities (14) first by noting that by (4), for all $\tau \leqslant T$,

$$G(\boldsymbol{\lambda}^\star_{\boldsymbol{B}-b\mathbf{1}}) - G(\boldsymbol{\lambda}_{\tau-1}) \leqslant 0,$$

and second, by applying Lemma 4 for each $1 \leqslant t \leqslant T$, using a confidence level $\delta/(4T)$. By a union bound, this indeed defines an event $\mathcal{E}_{\text{Bern-}\boldsymbol{c}}$ of probability at least $1 - \delta/4$.

**Step 4: Induction to bound the $\Lambda_t$.** In this step, we show by induction that, with high probability, the norms $\|\boldsymbol{\lambda}_{\boldsymbol{B}-\boldsymbol{b}\boldsymbol{1}}^\star - \boldsymbol{\lambda}_t\|$ satisfy the bound (18) stated below.

To do so, we combine the outcomes of Steps 1–3 and obtain that on the event $\mathcal{E}_\beta \cap \mathcal{E}_{\text{Bern-}\boldsymbol{c}}$, for all $1 \leqslant t \leqslant T$,

$$\|\boldsymbol{\lambda}_{\boldsymbol{B}-\boldsymbol{b}\boldsymbol{1}}^\star - \boldsymbol{\lambda}_t\|^2$$

$$\leqslant \|\boldsymbol{\lambda}_{\boldsymbol{B}-\boldsymbol{b}\boldsymbol{1}}^\star\|^2 + 4d\,\gamma^2 t + 2\gamma \left( 2\big(1 + \|\boldsymbol{\lambda}_{\boldsymbol{B}-\boldsymbol{b}\boldsymbol{1}}^\star\|_1\big)\beta_{t,\delta/4} + \sum_{\tau=1}^{t}\big(g_\tau(\boldsymbol{\lambda}_{\boldsymbol{B}-\boldsymbol{b}\boldsymbol{1}}^\star) - g_\tau(\boldsymbol{\lambda}_{\tau-1})\big) \right)$$

$$\leqslant \|\boldsymbol{\lambda}_{\boldsymbol{B}-\boldsymbol{b}\boldsymbol{1}}^\star\|^2 + 4d\,\gamma^2 t + 4\gamma\big(1 + \|\boldsymbol{\lambda}_{\boldsymbol{B}-\boldsymbol{b}\boldsymbol{1}}^\star\|_1\big)\beta_{t,\delta/4} + 2\gamma(1 + 2\Lambda_t)\sqrt{2t \ln \frac{T^2}{\delta/4}} + 4\gamma K_t \ln \frac{T^2}{\delta/4}\,,$$

where we recall that norms not indexed by a subscript are Euclidean norms, and

$$\Lambda_t = \max_{1 \leqslant \tau \leqslant t} \|\boldsymbol{\lambda}_{\boldsymbol{B}-\boldsymbol{b}\boldsymbol{1}}^\star - \boldsymbol{\lambda}_{\tau-1}\|_1 \qquad \text{and} \qquad K_t = 4\big(1 + \|\boldsymbol{\lambda}_{\boldsymbol{B}-\boldsymbol{b}\boldsymbol{1}}^\star\|_1 + 2d\gamma t\big)\,.$$

We upper bound $\|\boldsymbol{\lambda}_{\boldsymbol{B}-\boldsymbol{b}\boldsymbol{1}}^\star\|_1$ and $\Lambda_t$ in terms of Euclidean norms,

$$\|\boldsymbol{\lambda}_{\boldsymbol{B}-\boldsymbol{b}\boldsymbol{1}}^\star\|_1 \leqslant \sqrt{d}\,\|\boldsymbol{\lambda}_{\boldsymbol{B}-\boldsymbol{b}\boldsymbol{1}}^\star\| \qquad \text{and} \qquad \Lambda_t \leqslant \sqrt{d}\, \max_{1 \leqslant \tau \leqslant t} \|\boldsymbol{\lambda}_{\boldsymbol{B}-\boldsymbol{b}\boldsymbol{1}}^\star - \boldsymbol{\lambda}_{\tau-1}\|\,,$$

perform some crude boundings like $\sqrt{t} \leqslant \sqrt{T}$ and $\beta_{t,\delta/4} \leqslant \beta_{T,\delta/4}$, and obtain the following induction relationship: on the event $\mathcal{E}_\beta \cap \mathcal{E}_{\text{Bern-}\boldsymbol{c}}$, for all $1 \leqslant t \leqslant T$,

$$\|\boldsymbol{\lambda}_{\boldsymbol{B}-\boldsymbol{b}\boldsymbol{1}}^\star - \boldsymbol{\lambda}_t\|^2 \leqslant A + B\,t + C \max_{1 \leqslant \tau \leqslant t} \|\boldsymbol{\lambda}_{\boldsymbol{B}-\boldsymbol{b}\boldsymbol{1}}^\star - \boldsymbol{\lambda}_{\tau-1}\|\,, \tag{16}$$

where

$$A = \|\boldsymbol{\lambda}_{\boldsymbol{B}-\boldsymbol{b}\boldsymbol{1}}^\star\|^2 + \gamma\left(4\big(1 + \sqrt{d}\,\|\boldsymbol{\lambda}_{\boldsymbol{B}-\boldsymbol{b}\boldsymbol{1}}^\star\|\big)\beta_{T,\delta/4} + 2\sqrt{2T \ln \frac{T^2}{\delta/4}} + 16\big(1 + \sqrt{d}\,\|\boldsymbol{\lambda}_{\boldsymbol{B}-\boldsymbol{b}\boldsymbol{1}}^\star\|\big)\ln \frac{T^2}{\delta/4}\right),$$

$$B = 4d\gamma^2 + 4 \times 4 \times 2d\gamma^2 \ln \frac{T^2}{\delta/4} = \left(4 + 32 \ln \frac{T^2}{\delta/4}\right)d\gamma^2 \leqslant 36\,d\gamma^2 \ln \frac{T^2}{\delta/4}\,,$$

$$C = 4\gamma\sqrt{2dT \ln \frac{T^2}{\delta/4}}\,.$$

We now show that (16) implies that on $\mathcal{E}_\beta \cap \mathcal{E}_{\text{Bern-}\boldsymbol{c}}$,

$$\forall\, 0 \leqslant t \leqslant T, \qquad \|\boldsymbol{\lambda}_{\boldsymbol{B}-\boldsymbol{b}\boldsymbol{1}}^\star - \boldsymbol{\lambda}_t\| \leqslant M \stackrel{\text{def}}{=} \frac{C}{2} + \sqrt{A + BT + \frac{C^2}{4}}\,. \tag{17}$$

Indeed, for $t = 0$, given that $\boldsymbol{\lambda}_0 = \boldsymbol{0}$, we have $\|\boldsymbol{\lambda}_{\boldsymbol{B}-\boldsymbol{b}\boldsymbol{1}}^\star - \boldsymbol{\lambda}_t\| = \|\boldsymbol{\lambda}_{\boldsymbol{B}-\boldsymbol{b}\boldsymbol{1}}^\star\| \leqslant \sqrt{A}$. Now, if the bound (17) is satisfied for all $0 \leqslant \tau \leqslant t$, where $0 \leqslant t \leqslant T - 1$, then (16) implies that

$$\|\boldsymbol{\lambda}_{\boldsymbol{B}-\boldsymbol{b}\boldsymbol{1}}^\star - \boldsymbol{\lambda}_{t+1}\| \leqslant A + B\,(t+1) + CM \leqslant A + BT + CM \leqslant M^2\,,$$

where the final inequality follows from the fact that (by definition of $M$, and this explains how we picked $M$)

$$M^2 - CM = \left(M - \frac{C}{2}\right)^2 + \frac{C^2}{4} = A + BT\,.$$

Below, we will make repeated uses of $\sqrt{x+y} \leqslant \sqrt{x} + \sqrt{y}$, of $xy \leqslant 2(x^2 + y^2)$, and of $\sqrt{x} \leqslant 1 + x$, for $x, y \geqslant 0$. From (17), we conclude that on the event $\mathcal{E}_\beta \cap \mathcal{E}_{\text{Bern-}\boldsymbol{c}}$,

$$\forall\, 0 \leqslant t \leqslant T, \qquad \|\boldsymbol{\lambda}_{\boldsymbol{B}-\boldsymbol{b}\boldsymbol{1}}^\star - \boldsymbol{\lambda}_t\| \leqslant \sqrt{A} + \sqrt{BT} + C$$

$$= \sqrt{A} + 6\gamma\sqrt{dT \ln \frac{T^2}{\delta/4}} + 4\gamma\sqrt{2dT \ln \frac{T^2}{\delta/4}} \leqslant \sqrt{A} + 6\gamma\beta'_{T,\delta/4}\,,$$

where we denoted

$$\beta'_{T,\delta/4} = \max\left\{\beta_{T,\delta/4},\ 2\sqrt{dT \ln \frac{T^2}{\delta/4}}\right\} \geqslant \ln \frac{T^2}{\delta/4}\,.$$

For the sake of readability, we may further bound $\sqrt{A}$ as follows (in some crude way):

$$\sqrt{A} \leqslant \|\boldsymbol{\lambda}^\star_{\boldsymbol{B}-\boldsymbol{b}\mathbf{1}}\| + \sqrt{\gamma}\sqrt{4\big(1 + \sqrt{d}\,\|\boldsymbol{\lambda}^\star_{\boldsymbol{B}-\boldsymbol{b}\mathbf{1}}\|\big)\beta_{T,\delta/4} + 2\sqrt{2T\ln\frac{T^2}{\delta/4}} + 16\big(1 + \sqrt{d}\,\|\boldsymbol{\lambda}^\star_{\boldsymbol{B}-\boldsymbol{b}\mathbf{1}}\|\big)\ln\frac{T^2}{\delta/4}}$$

$$\leqslant \|\boldsymbol{\lambda}^\star_{\boldsymbol{B}-\boldsymbol{b}\mathbf{1}}\| + \sqrt{\gamma}\sqrt{22\beta'_{T,\delta/4} + 20\sqrt{d}\,\|\boldsymbol{\lambda}^\star_{\boldsymbol{B}-\boldsymbol{b}\mathbf{1}}\|\beta'_{T,\delta/4}}$$

$$\leqslant \|\boldsymbol{\lambda}^\star_{\boldsymbol{B}-\boldsymbol{b}\mathbf{1}}\| + 1 + 6\gamma\beta'_{T,\delta/4} + \|\boldsymbol{\lambda}^\star_{\boldsymbol{B}-\boldsymbol{b}\mathbf{1}}\| + 5\gamma\sqrt{d}\,\beta'_{T,\delta/4}\,,$$

where we used the facts that $\sqrt{20xy} = 2\sqrt{5xy} \leqslant x + 5y$ and $\sqrt{22x} = 2\sqrt{22x/4} \leqslant 1 + 6x$.

All in all, we proved that on the event $\mathcal{E}_\beta \cap \mathcal{E}_{\text{Bern-}\boldsymbol{c}}$,

$$\forall\, 0 \leqslant t \leqslant T, \qquad \|\boldsymbol{\lambda}^\star_{\boldsymbol{B}-\boldsymbol{b}\mathbf{1}} - \boldsymbol{\lambda}_t\| \leqslant 2\|\boldsymbol{\lambda}^\star_{\boldsymbol{B}-\boldsymbol{b}\mathbf{1}}\| + 17\gamma\sqrt{d}\,\beta'_{T,\delta/4} + 1\,. \tag{18}$$

**Step 5: Conclusion.** We combine the bound (18) with the bound (9) of Step 1 and the bound (7): on the intersection of events $\mathcal{E}_\beta \cap \mathcal{E}_{\text{Bern-}\boldsymbol{c}} \cap \mathcal{E}_{\text{H-Az}}$, which has a probability at least $1 - 3\delta/4$, for all $1 \leqslant t \leqslant T$,

$$\left\|\left(\sum_{\tau=1}^{t} \boldsymbol{c}_\tau - t(\boldsymbol{B} - \boldsymbol{b}\mathbf{1})\right)_+\right\| \leqslant \sqrt{d}\big(\alpha_{T,\delta/4} + 2\beta_{T,\delta/4}\big) + \frac{\|\boldsymbol{\lambda}_t\|}{\gamma}$$

$$\leqslant \sqrt{d}\big(\alpha_{T,\delta/4} + 2\beta_{T,\delta/4}\big) + \frac{\|\boldsymbol{\lambda}^\star_{\boldsymbol{B}-\boldsymbol{b}\mathbf{1}}\|}{\gamma} + \frac{\|\boldsymbol{\lambda}^\star_{\boldsymbol{B}-\boldsymbol{b}\mathbf{1}} - \boldsymbol{\lambda}_t\|}{\gamma}$$

$$\leqslant \frac{3\|\boldsymbol{\lambda}^\star_{\boldsymbol{B}-\boldsymbol{b}\mathbf{1}}\| + 1}{\gamma} + \sqrt{d}\big(\alpha_{T,\delta/4} + 19\beta'_{T,\delta/4}\big)\,.$$

This entails in particular the stated result, given the definition of $\Upsilon_{T,\delta}$ as $\max\{\beta'_{T,\delta/4},\ \alpha_{T,\delta/4}\}$.

### B.2 Proof of Lemma 2

The proof is similar to (but much simpler and shorter than) the one of Lemma 1 and borrows some of its arguments. We use throughout this section the notation introduced therein; we also define a new event $\mathcal{E}_{\text{Bern-}r}$ of probability at least $1 - \delta/4$.

We start from (8) and introduce the same $\Delta\widehat{\boldsymbol{c}}_t$ quantity as in (10): on the event $\mathcal{E}_{\text{H-Az}} \cap \mathcal{E}_\beta$, for all $1 \leqslant t \leqslant T$,

$$\sum_{\tau=1}^{t} r_\tau \geqslant -\big(\alpha_{t,\delta/4} + 2\beta_{t,\delta/4}\big) + \sum_{\tau=1}^{t} \widehat{r}^{\,\text{ucb}}_{\delta/4,\tau-1}(\boldsymbol{x}_\tau, a_\tau)$$

$$\geqslant -\big(\alpha_{t,\delta/4} + 2\beta_{t,\delta/4}\big) + \sum_{\tau=1}^{t}\Big(\widehat{r}^{\,\text{ucb}}_{\delta/4,\tau-1}(\boldsymbol{x}_\tau, a_\tau) - \langle\Delta\widehat{\boldsymbol{c}}_\tau, \boldsymbol{\lambda}_{\tau-1}\rangle\Big) + \sum_{\tau=1}^{t}\langle\Delta\widehat{\boldsymbol{c}}_\tau, \boldsymbol{\lambda}_{\tau-1}\rangle\,.$$

On the one hand, the result (11) with $\boldsymbol{\lambda} = \boldsymbol{0}$ exactly states that

$$\sum_{\tau=1}^{t}\langle\Delta\widehat{\boldsymbol{c}}_\tau, \boldsymbol{\lambda}_{\tau-1}\rangle \geqslant \frac{\|\boldsymbol{\lambda}_t\|^2}{2\gamma} - 2d\gamma t \geqslant -2d\gamma t\,.$$

On the other hand, the result (13) states that on $\mathcal{E}_\beta$, for all $1 \leqslant \tau \leqslant T$,

$$\widehat{r}^{\,\text{ucb}}_{\delta/4,\tau-1}(\boldsymbol{x}_\tau, a_\tau) - \langle\Delta\widehat{\boldsymbol{c}}_\tau, \boldsymbol{\lambda}_{\tau-1}\rangle = \widehat{r}^{\,\text{ucb}}_{\delta/4,\tau-1}(\boldsymbol{x}_\tau, a_\tau) - \langle\widehat{\boldsymbol{c}}^{\,\text{lcb}}_{\delta/4,\tau-1}(\boldsymbol{x}_\tau, a_\tau) - (\boldsymbol{B} - \boldsymbol{b}\mathbf{1}), \boldsymbol{\lambda}_{\tau-1}\rangle$$

$$\geqslant g_\tau(\boldsymbol{\lambda}_{\tau-1})\,.$$

A similar application of Lemma 4 as in the proof of Lemma 1 shows that on a new event $\mathcal{E}_{\text{Bern-}r}$ of probability at least $1 - \delta/4$, for all $1 \leqslant t \leqslant T$,

$$\sum_{\tau=1}^{t} g_\tau(\boldsymbol{\lambda}_{\tau-1}) \geqslant \sum_{\tau=1}^{t} G(\boldsymbol{\lambda}_{\tau-1}) - \sqrt{2\sum_{\tau=1}^{t}\big(1 + 2\|\boldsymbol{\lambda}_{\tau-1}\|_1\big)^2\ln\frac{T^2}{\delta/4}} - 8(1 + 2\gamma t)\ln\frac{T^2}{\delta/4}\,.$$

We relate $\ell^1$–norms to Euclidean norms, resort to a triangle inequality, and substitute (18) to get that on $\mathcal{E}_{\text{Bern-}c}$, for all $1 \leqslant t \leqslant T$,

$$
-\sqrt{2\sum_{\tau=1}^{t}\left(1+2\|\boldsymbol{\lambda}_{\tau-1}\|_1\right)^2\ln\frac{T^2}{\delta/4}} \geqslant -\left(1+2\sqrt{d}\max_{0\leqslant\tau\leqslant t-1}\|\boldsymbol{\lambda}_{\tau-1}\|\right)\sqrt{2t\ln\frac{T^2}{\delta/4}}
$$

$$
\geqslant -\left(1+2\sqrt{d}\,\|\boldsymbol{\lambda}^\star_{\boldsymbol{B}-b\mathbf{1}}\|+2\sqrt{d}\max_{0\leqslant\tau\leqslant t-1}\|\boldsymbol{\lambda}^\star_{\boldsymbol{B}-b\mathbf{1}}-\boldsymbol{\lambda}_{\tau-1}\|\right)\sqrt{2t\ln\frac{T^2}{\delta/4}}
$$

$$
\geqslant -\left(8\sqrt{d}\,\|\boldsymbol{\lambda}^\star_{\boldsymbol{B}-b\mathbf{1}}\|+34\gamma d\,\beta'_{T,\delta/4}+2\sqrt{d}+1\right)\sqrt{2t\ln\frac{T^2}{\delta/4}}
$$

$$
\geqslant -\left(8\sqrt{d}\,\|\boldsymbol{\lambda}^\star_{\boldsymbol{B}-b\mathbf{1}}\|+34\gamma d\,\beta'_{T,\delta/4}+4\sqrt{d}\right)\sqrt{2t\ln\frac{T^2}{\delta/4}}
$$

$$
\geqslant -6\|\boldsymbol{\lambda}^\star_{\boldsymbol{B}-b\mathbf{1}}\|\,\beta'_{T,\delta/4}-25\gamma\sqrt{d}\left(\beta'_{T,\delta/4}\right)^2-2\sqrt{2}\,\beta'_{T,\delta/4}
$$

$$
\geqslant -6\|\boldsymbol{\lambda}^\star_{\boldsymbol{B}-b\mathbf{1}}\|\,\beta'_{T,\delta/4}-28\gamma\sqrt{d}\left(\beta'_{T,\delta/4}\right)^2,
$$

where we performed some crude boundings using the definition of $1 \leqslant \beta'_{t,\delta/4} \leqslant \beta'_{T,\delta/4}$. We also note that

$$
8(1+2\gamma t)\ln\frac{T^2}{\delta/4} \leqslant 8\ln\frac{T^2}{\delta/4}+4\gamma\left(\beta'_{T,\delta/4}\right)^2.
$$

By (4), we have $\text{OPT}(r,\boldsymbol{c},\boldsymbol{B}-b\mathbf{1}) = G(\boldsymbol{\lambda}^\star_{\boldsymbol{B}-b\mathbf{1}}) \leqslant G(\boldsymbol{\lambda})$, for all $\boldsymbol{\lambda} \geqslant \mathbf{0}$. Also,

$$
G(\boldsymbol{\lambda}^\star_{\boldsymbol{B}-b\mathbf{1}}) + b\,\|\boldsymbol{\lambda}^\star_{\boldsymbol{B}-b\mathbf{1}}\|_1
$$

$$
= \mathbb{E}_{\boldsymbol{X}\sim\nu}\left[\max_{a\in\mathcal{A}}\left\{r(\boldsymbol{X},a)-\big\langle\boldsymbol{c}(\boldsymbol{X},a)-(\boldsymbol{B}-b\mathbf{1}),\,\boldsymbol{\lambda}^\star_{\boldsymbol{B}-b\mathbf{1}}\big\rangle\right\}\right] + b\,\|\boldsymbol{\lambda}^\star_{\boldsymbol{B}-b\mathbf{1}}\|_1
$$

$$
= \mathbb{E}_{\boldsymbol{X}\sim\nu}\left[\max_{a\in\mathcal{A}}\left\{r(\boldsymbol{X},a)-\big\langle\boldsymbol{c}(\boldsymbol{X},a)-\boldsymbol{B},\,\boldsymbol{\lambda}^\star_{\boldsymbol{B}-b\mathbf{1}}\big\rangle\right\}\right] \geqslant \text{OPT}(r,\boldsymbol{c},\boldsymbol{B}),
$$

where we used again (4). In particular,

$$
\sum_{\tau=1}^{t} G(\boldsymbol{\lambda}_{\tau-1}) \geqslant t\,\text{OPT}(r,\boldsymbol{c},\boldsymbol{B})-t\,b\,\|\boldsymbol{\lambda}^\star_{\boldsymbol{B}-b\mathbf{1}}\|_1 \geqslant t\,\text{OPT}(r,\boldsymbol{c},\boldsymbol{B})-t\,b\sqrt{d}\,\|\boldsymbol{\lambda}^\star_{\boldsymbol{B}-b\mathbf{1}}\|_1.
$$

Collecting all bounds above and using the definition of $\Upsilon_{T,\delta}$ as $\max\left\{\beta'_{T,\delta/4},\,\alpha_{T,\delta/4}\right\}$ and the fact that $dt \leqslant \left(\Upsilon_{T,\delta}\right)^2$, we proved the following. On $\mathcal{E}_{\text{H-Az}}\cap\mathcal{E}_\beta\cap\mathcal{E}_{\text{Bern-}c}\cap\mathcal{E}_{\text{Bern-}r}$, which is indeed an event with probability at least $1-\delta$, for all $1 \leqslant t \leqslant T$,

$$
\sum_{\tau=1}^{t} r_\tau
$$

$$
\geqslant -3\Upsilon_{T,\delta}-2d\,\gamma t+\sum_{\tau=1}^{t} G(\boldsymbol{\lambda}_{\tau-1})-\sqrt{2\sum_{t=1}^{T}\left(1+2\|\boldsymbol{\lambda}_{t-1}\|_1\right)^2\ln\frac{T^2}{\delta/4}}-8\ln\frac{T^2}{\delta/4}-4\gamma\left(\beta'_{T,\delta/4}\right)^2
$$

$$
\geqslant t\,\text{OPT}(r,\boldsymbol{c},\boldsymbol{B})-\|\boldsymbol{\lambda}^\star_{\boldsymbol{B}-b\mathbf{1}}\|\left(t\,b\sqrt{d}+6\Upsilon_{T,\delta}\right)-36\gamma\sqrt{d}\left(\Upsilon_{T,\delta}\right)^2-8\ln\frac{T^2}{\delta/4}.
$$

This entails in particular the stated result.

## C  Proof of Theorem 1

The proof is divided into three steps: on a favorable event $\mathcal{E}_{\text{meta}}$ of probability at least $1-\delta$, (i) we bound by $i\log T$ the index of the last regime achieved in Box C;

---

**Algorithm 1:** Pseudo-code for the Box C strategy

---

**Input:** number of rounds $T$; confidence level $1 - \delta$; margin $b$ on the average constraints; estimation procedure and error functions $\varepsilon_t$ of Assumption 2; optimistic estimates (2)

**Initialization:** $T_0 = 1$; sequence $\gamma_k = 2^k/\sqrt{T}$ of step sizes;
  sequence $M_{T,\delta,k} = 4\sqrt{T} + 20\sqrt{d}\, \Upsilon_{T,\delta/(k+2)^2}$ of cost deviations

---

1 **for** $k \geqslant 0$ **do**                                                        `// Box C part`
2    $\boldsymbol{\lambda}_{T_k-1} = \mathbf{0}$;
3    **for** $t \geqslant T_k$ **do**                                          `// Box B part`
4      **if** $t = T$ **then**
5        Terminate algorithm;
6      **end**
7      Observe the context $\boldsymbol{x}_t$;
8      Pick an action

$$a_t \in \arg\max_{a \in \mathcal{A}}\Big\{ \widehat{r}^{\,\mathrm{ucb}}_{\delta,t-1}(\boldsymbol{x}_t, a) - \big\langle \widehat{\boldsymbol{c}}^{\,\mathrm{lcb}}_{\delta,t-1}(\boldsymbol{x}_t, a) - (\boldsymbol{B} - b\mathbf{1}),\, \boldsymbol{\lambda}_{t-1}\big\rangle \Big\};$$

9      Observe the payoff $r_t$ and the costs $\boldsymbol{c}_t$;
10      Compute $\boldsymbol{\lambda}_t = \left( \boldsymbol{\lambda}_{t-1} + \gamma_k \big( \widehat{\boldsymbol{c}}^{\,\mathrm{lcb}}_{\delta,t-1}(\boldsymbol{x}_t, a_t) - (\boldsymbol{B} - b\mathbf{1}) \big) \right)_+$;    `// make PGD update`
11      Compute the estimates $\widehat{r}^{\,\mathrm{ucb}}_{\delta,t}$ and $\widehat{\boldsymbol{c}}^{\,\mathrm{lcb}}_{\delta,t}$;
12      **if** $\left\| \left( \sum\limits_{\tau=T_k}^{t} \boldsymbol{c}_\tau - (t - T_k + 1)(\boldsymbol{B} - b\mathbf{1}) \right)_+ \right\| > M_{T,\delta,k}$ **then**    `// Box C part`
13        Break inner for loop;                `// finish phase `$k$` and move to phase `$k+1$`
14        $T_{k+1} = t + 1$;                          `// record beginning of phase `$k+1$`
15      **end**
16    **end**
17 **end**

---

(ii) we bound by $(1 + \mathrm{ilog}\, T)\sqrt{T}$ the excess cumulative costs with respect to $T(\boldsymbol{B} - b_T\mathbf{1})$ and deduce that the cumulative costs are smaller than $B$;

(iii) we provide a regret bound, by summing the bounds guaranteed by Lemma 1 over each regime.

The favorable event $\mathcal{E}_{\mathrm{meta}}$ is defined as follows. By construction, and thanks to the assumption of feasibility for $\boldsymbol{B} - 2b_T\mathbf{1}$, the results of Lemmas 1 and 2 hold with probability at least $1 - \delta/(k+2)^2$ at each round of each regime $k \geqslant 0$ that is actually achieved. Given that

$$\sum_{k \geqslant 0} \frac{1}{(k+2)^2} \leqslant \sum_{k \geqslant 0} \frac{1}{(k+1)(k+2)} = 1\,,$$

we may define $\mathcal{E}_{\mathrm{meta}}$ as the event indicating that the (uniform-in-time) bounds of Lemmas 1 and 2 are satisfied within each of the regimes achieved.

## C.1   Step 1: Bounding the number of regimes achieved

We show that, on $\mathcal{E}_{\mathrm{meta}}$, if regime $K = \mathrm{ilog}\, \|\boldsymbol{\lambda}^\star_{\boldsymbol{B}-b_T\mathbf{1}}\|$ is achieved, then this regime does not stop. Hence, on $\mathcal{E}_{\mathrm{meta}}$, at most $K + 1 \leqslant 1 + \mathrm{ilog}\, \|\boldsymbol{\lambda}^\star_{\boldsymbol{B}-b_T\mathbf{1}}\|$ regimes take place.

Indeed, in regime $K = \mathrm{ilog}\, \|\boldsymbol{\lambda}^\star_{\boldsymbol{B}-b_T\mathbf{1}}\|$, if it is achieved, the meta-strategy resorts to the Box B strategy with

$$\gamma_K = \frac{2^K}{\sqrt{T}} \geqslant \frac{\max\big\{\|\boldsymbol{\lambda}^\star_{\boldsymbol{B}-b_T\mathbf{1}}\|,\, 1\big\}}{\sqrt{T}}\,.$$

Therefore, the bound of Lemma 1 entails that on $\mathcal{E}_{\text{meta}}$, for all $t \geqslant T_K$,

$$\left\| \left( \sum_{\tau=T_K}^{t} \boldsymbol{c}_\tau - (t - T_k + 1)(\boldsymbol{B} - b_T \boldsymbol{1}) \right)_+ \right\| \leqslant \frac{1 + 3\|\boldsymbol{\lambda}^\star_{\boldsymbol{B} - b_T \boldsymbol{1}}\|}{\gamma_K} + 20\sqrt{d}\, \Upsilon_{T, \delta/(K+2)^2}$$

$$\leqslant 4\sqrt{T} + 20\sqrt{d}\, \Upsilon_{T, \delta/(K+2)^2} = M_{T, \delta, K} \,.$$

This is exactly the contrary of the stopping condition of regime $K$: the latter thus cannot be broken.

In some bounds, we will further bound $\text{ilog}\,\|\boldsymbol{\lambda}^\star_{\boldsymbol{B} - b_T \boldsymbol{1}}\|$ by $\text{ilog}\,T$: this holds because the assumption of $(\boldsymbol{B} - 2b_T \boldsymbol{1})$–feasibility entails, by Lemma 3 and as OPT is always smaller than 1, the crude bound

$$\|\boldsymbol{\lambda}^\star_{\boldsymbol{B} - b_T \boldsymbol{1}}\| \leqslant \frac{\text{OPT}(r, \boldsymbol{c}, \boldsymbol{B} - b_T \boldsymbol{1}) - \text{OPT}(r, \boldsymbol{c}, \boldsymbol{B} - (3/2)b_T \boldsymbol{1})}{b_T/2} \leqslant \frac{2}{b_T} \leqslant \frac{\sqrt{T}}{7} \leqslant T \,,$$

where we used that $b_T \geqslant 14/\sqrt{T}$ given its definition. (Of course, sharper but more complex bounds could be obtained; however, they would only improve logarithmic terms in the bound, which we do not try to optimize anyway.)

## C.2    Step 2: Bounding the cumulative costs

We still denote by $K$ the index of the last regime and recall that $K \leqslant \text{ilog}\,\|\boldsymbol{\lambda}^\star_{\boldsymbol{B} - b_T \boldsymbol{1}}\| \leqslant \text{ilog}(T)$, and that for $k \geqslant 0$, regime $k$ starts at $T_k$ and stops at $T_{k+1} - 1$. By convention, $T_0 = 1$ and $T_{K+1} = T + 1$.

By the very definition of the stopping condition of regime $k \geqslant 0$,

$$\left\| \left( \sum_{t=T_k}^{T_{k+1}-2} \boldsymbol{c}_t - (T_{k+1} - T_k - 1)(\boldsymbol{B} - b_T \boldsymbol{1}) \right)_+ \right\| \leqslant M_{T, \delta, k} \,.$$

For rounds of the form $t = T_{k+1} - 1$, we bound the Euclidean norm of $\boldsymbol{c}_t - (\boldsymbol{B} - b_T \boldsymbol{1})$ by $2\sqrt{d}$. Therefore, by a triangle inequality, satisfied both by the non-negative part $(\cdot)_+$ and the norm $\|\cdot\|$ functions, we have

$$\left\| \left( \sum_{t=1}^{T} \boldsymbol{c}_t - T(\boldsymbol{B} - b_T \boldsymbol{1}) \right)_+ \right\|$$

$$\leqslant \sum_{k=0}^{K} \left\| \left( \sum_{t=T_k}^{T_{k+1}-2} \boldsymbol{c}_t - (T_{k+1} - T_k - 1)(\boldsymbol{B} - b_T \boldsymbol{1}) \right)_+ \right\| + \sum_{k=0}^{K} \left\| \boldsymbol{c}_{T_{k+1}-1} - (\boldsymbol{B} - b_T \boldsymbol{1}) \right\|$$

$$\leqslant (K+1)\big(M_{T, \delta, \text{ilog}\,T} + 2\sqrt{d}\big) \leqslant T\, b_T \,,$$

where we used the fact that $M_{T, \delta, k}$ increases with $k$, the bound $K \leqslant \text{ilog}\,T$ proved in Step 1 and holding on the event $\mathcal{E}_{\text{meta}}$, as well as the definition of $b_T$. Therefore, on $\mathcal{E}_{\text{meta}}$, no component of

$$\left( \sum_{t=1}^{T} \boldsymbol{c}_t - T(\boldsymbol{B} - b_T \boldsymbol{1}) \right)_+$$

can be larger than $T\, b_T$, which yields the desired control $\displaystyle\sum_{t=1}^{T} \boldsymbol{c}_t \leqslant T\boldsymbol{B}$.

## C.3    Step 3: Computing the associated regret bound

The total regret is the sum of the regrets suffered over each regime:

$$R_T = \sum_{k=0}^{K} \left( (T_{k+1} - T_k)\text{OPT}(r, \boldsymbol{c}, \boldsymbol{B}) - \sum_{t=T_k}^{T_{k+1}-1} r_t \right) \,.$$

On the favorable event $\mathcal{E}_{\mathrm{meta}}$, the bound of Lemma 2 holds in particular at the end of each regime; i.e., given the parameters $\gamma_k = 2^k/\sqrt{T}$ and $\delta/\big(4(k+2)^2\big)$ to run the Box B strategy in regime $k \in \{0, \dots, K\}$, it holds on $\mathcal{E}_{\mathrm{meta}}$ that

$$(T_{k+1} - T_k)\,\mathrm{OPT}(r, \boldsymbol{c}, \boldsymbol{B}) - \sum_{t=T_k}^{T_{k+1}-1} r_t \leqslant \|\boldsymbol{\lambda}^{\star}_{\boldsymbol{B}-b_T\mathbf{1}}\|\Big((T_{k+1} - T_k)\,b_T\sqrt{d} + 6\Upsilon_{T,\delta/(k+2)^2}\Big)$$

$$+ 36\sqrt{d}\,\big(\Upsilon_{T,\delta/(k+2)^2}\big)^2\,\frac{2^k}{\sqrt{T}} + 8\ln\frac{T^2}{\delta/\big(4(k+2)^2\big)}\,.$$

We now sum the above bounds and use the (in)equalities $\Upsilon_{T,\delta/(k+2)^2} \leqslant \Upsilon_{T,\delta/(K+2)^2}$,

$$\sum_{k=0}^{K}(T_{k+1} - T_k)\,b_T = T\,b_T\,, \qquad \text{and} \qquad \sum_{k=0}^{K} 2^k \leqslant 2^{K+1} \leqslant 2^{1+\mathrm{ilog}\,\|\boldsymbol{\lambda}^{\star}_{\boldsymbol{B}-b_T\mathbf{1}}\|} \leqslant 4\,\|\boldsymbol{\lambda}^{\star}_{\boldsymbol{B}-b_T\mathbf{1}}\|$$

to get

$$R_T \leqslant \|\boldsymbol{\lambda}^{\star}_{\boldsymbol{B}-b_T\mathbf{1}}\|\Big(T\,b_T\sqrt{d} + 6K\Upsilon_{T,\delta/(K+2)^2}\Big) + 144\sqrt{d}\,\|\boldsymbol{\lambda}^{\star}_{\boldsymbol{B}-b_T\mathbf{1}}\|\frac{\big(\Upsilon_{T,\delta/(K+2)^2}\big)^2}{\sqrt{T}}$$

$$+ 8K\ln\frac{T^2}{\delta/\big(4(K+2)^2\big)}\,.$$

The final regret bound is achieved by substituting the inequality $K \leqslant \mathrm{ilog}\,T$ proved in Step 1:

$$R_T \leqslant \|\boldsymbol{\lambda}^{\star}_{\boldsymbol{B}-b_T\mathbf{1}}\|\left(144\sqrt{d}\,\frac{\big(\Upsilon_{T,\delta/(2+\mathrm{ilog}\,T)^2}\big)^2}{\sqrt{T}} + T\,b_T\sqrt{d} + 6\Upsilon_{T,\delta/(2+\mathrm{ilog}\,T)^2}\,\mathrm{ilog}\,T\right)$$

$$+ 8\ln\frac{T^2}{\delta/\big(4(2+\mathrm{ilog}\,T)^2\big)}\,\mathrm{ilog}\,T\,. \quad (19)$$

The order of magnitude is $\sqrt{T}$, up to poly-logarithmic terms, for the quantity $\Upsilon_{T,\delta/(2+\mathrm{ilog}\,T)^2}$, thus for $M_{T,\delta,\mathrm{ilog}\,T}$, thus for $T\,b_T$, therefore, the order of magnitude in $\sqrt{T}$ of the above bound is, up to poly-logarithmic terms,

$$\big(1 + \|\boldsymbol{\lambda}^{\star}_{\boldsymbol{B}-b_T\mathbf{1}}\|\big)\sqrt{T},$$

as claimed.

## D   Proofs of Lemma 3 and Corollary 1

*Proof of Lemma 3.* For $\boldsymbol{\lambda} \geqslant \mathbf{0}$ and $\boldsymbol{C} \in [0,1]^d$, we denote

$$L(\boldsymbol{\lambda}, \boldsymbol{C}) = \mathbb{E}_{\boldsymbol{X}\sim\nu}\left[\max_{a\in\mathcal{A}}\Big\{r(\boldsymbol{X}, a) - \big\langle \boldsymbol{c}(\boldsymbol{X}, a) - \boldsymbol{C},\,\boldsymbol{\lambda}\big\rangle\Big\}\right]$$

so that by (4) and the feasibility assumption, we have, at least for $\boldsymbol{C} = \widetilde{\boldsymbol{B}}$ and $\boldsymbol{C} = \boldsymbol{B} - b\mathbf{1}$:

$$\mathrm{OPT}(r, \boldsymbol{c}, \boldsymbol{C}) = \min_{\boldsymbol{\lambda}\geqslant\mathbf{0}} L(\boldsymbol{\lambda}, \boldsymbol{C}) = L(\boldsymbol{\lambda}^{\star}_{\boldsymbol{C}}, \boldsymbol{C})\,.$$

The function $L$ is linear in $\boldsymbol{C}$, so that

$$\mathrm{OPT}\big(r, \boldsymbol{c}, \widetilde{\boldsymbol{B}}\big) = L\big(\boldsymbol{\lambda}^{\star}_{\widetilde{\boldsymbol{B}}}, \widetilde{\boldsymbol{B}}\big) \leqslant L\big(\boldsymbol{\lambda}^{\star}_{\boldsymbol{B}-b\mathbf{1}}, \widetilde{\boldsymbol{B}}\big) = L\big(\boldsymbol{\lambda}^{\star}_{\boldsymbol{B}-b\mathbf{1}}, \boldsymbol{B}-b\mathbf{1}\big) - \big\langle \boldsymbol{\lambda}^{\star}_{\boldsymbol{B}-b\mathbf{1}},\,\boldsymbol{B}-b\mathbf{1}-\widetilde{\boldsymbol{B}}\big\rangle$$

$$= \mathrm{OPT}\big(r, \boldsymbol{c}, \boldsymbol{B}-b\mathbf{1}\big) - \big\langle \boldsymbol{\lambda}^{\star}_{\boldsymbol{B}-b\mathbf{1}},\,\boldsymbol{B}-b\mathbf{1}-\widetilde{\boldsymbol{B}}\big\rangle\,.$$

The result follows from substituting

$$\big\langle \boldsymbol{\lambda}^{\star}_{\boldsymbol{B}-b\mathbf{1}},\,\boldsymbol{B}-b\mathbf{1}-\widetilde{\boldsymbol{B}}\big\rangle \geqslant \|\boldsymbol{\lambda}^{\star}_{\boldsymbol{B}-b\mathbf{1}}\|_1\,\min\big(\boldsymbol{B}-b\mathbf{1}-\widetilde{\boldsymbol{B}}\big) \geqslant \|\boldsymbol{\lambda}^{\star}_{\boldsymbol{B}-b\mathbf{1}}\|\,\min\big(\boldsymbol{B}-b\mathbf{1}-\widetilde{\boldsymbol{B}}\big)$$

and from rearranging the inequality thus obtained. $\qquad\square$

*Proof of Corollary 1.* We apply Lemma 3 with $\widetilde{\boldsymbol{B}} = \varepsilon\mathbf{1}$ for some $\varepsilon > 0$ sufficiently small and obtain

$$\|\boldsymbol{\lambda}^{\star}_{\boldsymbol{B}-b\mathbf{1}}\| \leqslant \frac{\mathrm{OPT}(r, \boldsymbol{c}, \boldsymbol{B}-b\mathbf{1}) - \mathrm{OPT}\big(r, \boldsymbol{c}, \varepsilon\mathbf{1}\big)}{\min\big(\boldsymbol{B} - (b+\varepsilon)\mathbf{1}\big)}\,.$$

We conclude by substituting $\mathrm{OPT}(r, \boldsymbol{c}, \boldsymbol{B}-b\mathbf{1}) \leqslant \mathrm{OPT}(r, \boldsymbol{c}, \boldsymbol{B})$ and $\mathrm{OPT}\big(r, \boldsymbol{c}, \varepsilon\mathbf{1}\big) \geqslant \mathrm{OPT}\big(r, \boldsymbol{c}, \mathbf{0}\big)$ as well as $\min(\boldsymbol{B}-b\mathbf{1}) \geqslant \min\boldsymbol{B}/2$, and by letting $\varepsilon \to 0$. $\qquad\square$

# E Additional (sketch of) results concerning optimality

We detail here two series of claims made in Section 4 .

## E.1 A proof scheme for problem-dependent lower bounds

We provide the proof scheme for proving the problem-dependent lower bound of order $\left(1+\|\lambda_{\boldsymbol{B}}^{\star}\|\right)\sqrt{T}$ announced in Section 4.

**Step 0: Considering strict cost constraints.** Our aim, as described in Box A, is to make sure that with high probability the cumulative costs are smaller than $T\,\boldsymbol{B}$. If we considered softer constraints, of the form $T\,\boldsymbol{B}+\widetilde{\mathcal{O}}\big(\sqrt{T}\big)$, then $\widetilde{\mathcal{O}}\big(\sqrt{T}\big)$ regret bounds would be possible (see Appendix F); i.e., the factor $\|\lambda_{\boldsymbol{B}-b_T\mathbf{1}}^{\star}\|$ of Theorem 1 could be replaced by a constant. Thus, lower bounds are only interesting in the case of hard constraints stated in Box A.

**Step 1: Necessity of a margin $b_T$ of order $1/\sqrt{T}$.** First, a classical lemma in CBwK (see, e.g., Agrawal and Devanur, 2016, Lemma 1) indicates that a sequence of adaptive cannot perform better than an optimal static policy. Denote by $\pi_{\boldsymbol{B}'}^{\star}$ a (quasi-)optimal policy for the average cost constraints $\boldsymbol{B}'$. Provided that costs are truly random (i.e., do not stem from Dirac distributions, which in particular, does not cover the cases where there is a null-cost action, see Limitation 2 in Section 4), then the law of iterated logarithm shows that when playing $\pi_{\boldsymbol{B}'}^{\star}$ at each round, the cumulative costs must (almost-surely, as $T\to+\infty$) be larger than $T\,\boldsymbol{B}'$ plus a positive term of the order of $\sqrt{T\ln\ln T}$. Therefore, to meet the hard constraints, one should pick $\boldsymbol{B}'$ of the form $\boldsymbol{B}-b_T\mathbf{1}$, where $b_T$ is of order $1/\sqrt{T}$ up to logarithmic terms.

**Step 2: Consequences in terms of regret.** Therefore, the largest average reward a strategy may target is $\mathrm{OPT}(r,\boldsymbol{c},\boldsymbol{B}-b_T\mathbf{1})$. Deviations of the order $\sqrt{T}$ are also bound to happen. Therefore, up to logarithmic terms, the regret lower bound is approximatively larger than something of the order

$$T\big(\mathrm{OPT}(r,\boldsymbol{c},\boldsymbol{B})-\mathrm{OPT}(r,\boldsymbol{c},\boldsymbol{B}-b_T\mathbf{1})\big)+\sqrt{T}\,.$$

Now, an argument similar to the one used in the proof of Lemma 3 shows that

$$\mathrm{OPT}(r,\boldsymbol{c},\boldsymbol{B})-\mathrm{OPT}(r,\boldsymbol{c},\boldsymbol{B}-b_T\mathbf{1})\geqslant L(\lambda_{\boldsymbol{B}}^{\star},\boldsymbol{B})-L(\lambda_{\boldsymbol{B}}^{\star},\boldsymbol{B}-b_T\mathbf{1})=b_T\,\|\lambda_{\boldsymbol{B}}^{\star}\|\,.$$

All in all, the regret lower bound is thus approximatively larger than something of the order of

$$\left(1+\|\lambda_{\boldsymbol{B}}^{\star}\|\right)\sqrt{T}\,.$$

This matches the form of the bound of Theorem 1, but the dual vector $\lambda_{\boldsymbol{B}-b_T\mathbf{1}}^{\star}$ present in the upper bound is replaced by $\lambda_{\boldsymbol{B}}^{\star}$ in our lower bound.

## E.2 Faster rates may be achievable in some specific cases

We explain here why, in some specific cases, faster rates would be achievable—with $\sqrt{T}$ in the regret bound of Theorem 1 replaced by $\sqrt{TB}$ for a problem with scalar costs and total-cost constraints $B$.

Indeed, consider a problem similar to Example 1: with scalar costs, total-cost constraint $B>0$, featuring a baseline action $a_{\mathrm{null}}$ with null cost but also null reward, and additional actions with larger rewards and expected costs $c(\boldsymbol{x},a)\geqslant\alpha>0$. Let $N_T$ denotes the number of times non-null cost actions are played within the first $T$ rounds. Deviation inequalities have it that

$$\sum_{t=1}^{T}c_t\geqslant\alpha N_T-\tilde{O}\big(\sqrt{N_T}\big)\,,$$

where we recall that $\tilde{O}$ is up to poly-logarithmic factors; as a consequence, the total-cost constraint enforces that

$$N_T\leqslant\frac{TB}{\alpha}+\tilde{O}\left(\sqrt{\frac{TB}{\alpha}}\right)\,.$$

In particular, the margin $b_T$ only needs to be of order $\sqrt{N_T}/T$, i.e., $\sqrt{B/(\alpha T)}$, instead of $1/\sqrt{T}$.

Similarly, since at most $N_T$ non-null actions are played, the regret should be lower bounded by something of the order of

$$T\big(\text{OPT}(r,c,B) - \text{OPT}(r,c,B-b_T)\big) + \sqrt{N_T}\,,$$

where, as in Step 2 above,

$$\big(\text{OPT}(r,c,B) - \text{OPT}(r,c,B-b_T)\big) \geqslant b_T |\lambda_B^\star| = \tilde{O}\left(\sqrt{\frac{B}{\alpha T}}\right) |\lambda_B^\star|\,.$$

This suggests a lower bound on the regret of the order (up to poly-logarithmic factors) of

$$\big(|\lambda_B^\star| + 1\big)\sqrt{TB/\alpha} \qquad \text{instead of} \qquad \big(1 + |\lambda_B^\star|\big)\sqrt{T}\,.$$

The difference between the two bounds is significant when $B$ is small, i.e., $B \ll 1$. While proving a matching upper bound with the Box B and Box C strategies looks a bit tricky, we feel that it must be possible to do so with the primal approach of Appendix F, at least when the context space $\mathcal{X}$ is finite. If our intuition holds, then it would be possible to get an upper bound

$$R_T \leqslant \tilde{O}\Big(Tb_T\big(1 + |\lambda_{B-b_T}^*|\big)\Big)$$
$$= \tilde{O}\left(\sqrt{\frac{TB}{\alpha}}\left(1 + \frac{\text{OPT}(r,c,B) - \text{OPT}(r,c,0)}{B}\right)\right) = \tilde{O}\Big(\sqrt{TB/\alpha^3}\Big)\,,$$

where the last bound follows from $\text{OPT}(r,c,B) - \text{OPT}(r,c,0) \leqslant B/\alpha$, as explained in Example 1.

## F  Primal strategy

This section studies the primal strategy stated in Box D, which, at every round, solves an approximation of the primal optimization problem (1). The key issue in running such a primal approach is estimating $\nu$, see comments after Notation 1; this primal approach is essentially worth for the case of finite context sets $\mathcal{X}$. The aim of this section is threefold:

1. In Appendix F.1, we provide a theory of "soft" constraints, when total-cost deviations from $TB$ of order $\sqrt{T}$ up to logarithmic terms are allowed; at least when $\mathcal{X}$ is a finite set, the regret bound then becomes proportional to $\sqrt{T}$ up to logarithmic terms.

2. In Appendix F.2, we revisit and extend the results by Li and Stoltz [2022]. The extension consists of dealing with possibly signed constraints, and the revisited analysis (of the same strategy as in Li and Stoltz, 2022) consists in not directly dealing with KKT constraints (which, in addition, imposed the finiteness of $\mathcal{X}$) but in only relating optimization problems—defined with the true $r$ and $c$ or estimates thereof. We also offer a modular approach and separate the error terms coming from estimating $\nu$ and from estimating $r$ and $c$.

3. In Appendix F.3, we generalize the results of Appendix F.2, which rely on the existence of a null-cost action, and get guarantees that correspond quite exactly to the combination of Theorem 1 and the interpretation thereof offered by Lemma 3, at least in the case of a finite $\mathcal{X}$. Actually, in our research path, we had first obtained these primal results, before trying to obtain them in a more general case of a continuous $\mathcal{X}$, by resorting to a dual strategy. The proof technique in Appendix F.3 also inspired our approach to proving problem-dependent lower bounds presented in Appendix E.1.

Throughout this appendix, we will assume that the context distribution $\nu$ can be estimated in some way. We provide examples and pointers below.

**Notation 1.** *Fix $\delta \in (0,1)$. We denote by $\widehat{\nu}_{\delta,t}$ a sequence of estimators of $\nu$, each constructed on the contexts $\boldsymbol{x}_1, \ldots, \boldsymbol{x}_t$, and by $\xi_{t,\delta}$ a sequence of estimation errors such that, with probability at least $1 - \delta$, for all bounded functions $f : \mathcal{X} \to [-1,1]$,*

$$\left| \mathbb{E}_{\boldsymbol{X} \sim \nu}\big[f(\boldsymbol{X})\big] - \mathbb{E}_{\boldsymbol{X} \sim \widehat{\nu}_{\delta,t}}\big[f(\boldsymbol{X})\big] \right| \leqslant \xi_{t,\delta}\,.$$

*We also denote* $\quad \Xi_{T,\delta} = \sum_{t=1}^{T} \xi_{t-1,\delta}\,.$

We consider the strategy described in Box D, whether $\mathcal{X}$ is finite (as in Li and Stoltz, 2022, where it was first considered) or not. Our simplified analyses do not rely on explicit KKT inequalities and therefore do not require anymore that $\mathcal{X}$ is finite. In Box D, by "when the empirical cost constraints are feasible", we mean that there exists some policy $\boldsymbol{\pi}$ such that

$$\mathbb{E}_{\boldsymbol{X} \sim \widehat{\nu}_{\delta,t-1}} \left[ \sum_{a \in \mathcal{A}} \widehat{\boldsymbol{c}}_{\delta,t-1}^{\text{lcb}}(\boldsymbol{X}, a)\, \pi_a(\boldsymbol{X}) \right] \leqslant \boldsymbol{B} + b_t \mathbf{1} \,.$$

We need to guarantee the existence of a policy achieving the constrained maximum of Step 2 in Box D. This is immediate when $\mathcal{X}$ is finite, as the problem then corresponds to a finite-dimensional linear program. In general, we may note that when (4) holds, we read therein the optimal policy, as a pointwise maximum involving the optimal dual variables. The proofs reveal that up to slightly augmenting the $\xi_{t-1,\delta}$ of Notation 1, we may assume that the strict feasibility sufficient for (4) indeed holds.

We also note that even if the argument of the maximum $\boldsymbol{\pi}_t$ is guaranteed to exist, it may be difficult to compute: the linear program of Box D cannot be solved exactly if the $\widehat{\nu}_{\delta,t}$ are not finitely supported (which happens in general when $\mathcal{X}$ is not finite). We do not see this as a severe issues as the Box D strategy is an ideal strategy anyway, that we study for the sake of shedding lights on our results for the dual approach in the main body of the article.

A final remark on the Box D strategy is that the margins on the average cost constraints $\boldsymbol{B}$ can now be signed, which is why they will be referred to as signed slacks.

---

**BOX D: PRIMAL CBWK STRATEGY, GENERALIZED FROM LI AND STOLTZ [2022]**

**Inputs:** confidence level $1 - \delta$; estimation procedure and error functions $\varepsilon_t$ of Assumption 2; optimistic estimates (2); estimation procedure for $\nu$ as in Notation 1

**Hyperparameter:** signed slacks $b_1, b_2, \ldots, b_T \in \mathbb{R}$

**Initialization:** initial estimates $\widehat{r}_{\delta,0}^{\text{ucb}}(\boldsymbol{x}, a)$ and $\widehat{\boldsymbol{c}}_{\delta,0}^{\text{lcb}}(\boldsymbol{x}, a)$, as well as $\widehat{\nu}_{\delta,0}$

**For rounds** $t = 1, 2, 3, \ldots, T$:

1. Compute a policy $\boldsymbol{\pi}_t$ achieving

$$\max_{\boldsymbol{\pi}: \mathcal{X} \to \mathcal{P}(\mathcal{A})} \quad \mathbb{E}_{\boldsymbol{X} \sim \widehat{\nu}_{\delta,t}} \left[ \sum_{a \in \mathcal{A}} \widehat{r}_{\delta,t-1}^{\text{ucb}}(\boldsymbol{X}, a)\, \pi_a(\boldsymbol{X}) \right]$$

$$\text{under} \quad \mathbb{E}_{\boldsymbol{X} \sim \widehat{\nu}_{\delta,t-1}} \left[ \sum_{a \in \mathcal{A}} \widehat{\boldsymbol{c}}_{\delta,t-1}^{\text{lcb}}(\boldsymbol{X}, a)\, \pi_a(\boldsymbol{X}) \right] \leqslant \boldsymbol{B} + b_t \mathbf{1}$$

   when the empirical cost constraints are feasible, and pick an arbitrary policy $\boldsymbol{\pi}_t$ otherwise;

2. Observe $\boldsymbol{x}_t$ and draw an action $a_t \sim \boldsymbol{\pi}_t(\boldsymbol{x}_t)$;

3. Compute the estimate $\widehat{r}_{\delta,t}^{\text{ucb}}(\boldsymbol{x}, a)$ and $\widehat{\boldsymbol{c}}_{\delta,t}^{\text{lcb}}(\boldsymbol{x}, a)$, as well as $\widehat{\nu}_{\delta,t}$.

---

**Discussion of the estimation of $\nu$.** The simplest case is when $\mathcal{X}$ is a finite set; in this case, we may take

$$\xi_{t,\delta} \qquad \text{of order} \qquad \sqrt{\frac{|\mathcal{X}| \ln(1/\delta)}{t}} \,,$$

where $\mathcal{X}$ denotes the cardinality of $\mathcal{X}$; see [Ai et al., 2022, Section 4.1], see also a less sharp bound based on Hoeffding's inequality in [Li and Stoltz, 2022, Section 5]. This leads to a total error term $\Xi_{T,\delta}$ of order $\sqrt{T}$ up to poly-logarithmic term.

When $\mathcal{X}$ is a continuous subset of $\mathbb{R}^n$, some regularity conditions are put on $\nu$, which is typically assumed to have some smooth density with respect to the Lebesgue measure $\mathfrak{m}$. Estimates $\widehat{\nu}_{\delta,t}$ are obtained by estimating the density $\mathrm{d}\nu/\mathrm{d}\mathfrak{m}$: the criterion in Notation 1, which is proportional to the total-variation distance between $\nu$ and $\widehat{\nu}_{\delta,t}$, is then given by the $\mathbb{L}^1(\mathfrak{m})$ distance between the two

densities. To control the latter with uniform convergence rates, so as to obtain deviation terms $\xi_{t,\delta}$ only depending on $t$ and $\delta$, heavy assumptions on the model to which $\nu$ belongs are in order, e.g., some Hölderian regularity, and uniform estimation rates obtained degrade with the ambient dimension $n$; they are generally much slower than $1/\sqrt{t}$. The total error terms $\Xi_{T,\delta}$ then prevent the bounds stated below in Proposition 1 and Theorems 3 and 4 from sharing the same orders of magnitude than the bounds proved with our dual approach in Theorem 1 and interpreted in Section 3.3. On this topic, see also [Ai et al., 2022, Section 4.1] for the estimation rates as well as a similar description in Han et al. [2022, end of Section 1] of the limitation of the primal approach in CBwK due to the estimation of densities. General references on density estimations are the monographs by Devroye and Györfi [1985] in $\mathbb{L}^1$ and Tsybakov [2008] in $\mathbb{L}^2$.

Note that in the dual approach, the knowledge of (the possibly complex) $\nu$ is replaced by the knowledge of the (finite-dimensional) optimal dual variables $\boldsymbol{\lambda}_{\boldsymbol{B}}^{\star} \in \mathbb{R}^d$, which is easier to learn. This explains the fundamental efficiency of the dual approach compared to the primal approach.

### F.1 Analysis with "soft" constraints

We first provide an analysis for a version of the Box D strategy that may possibly breach the total-cost constraints $T\boldsymbol{B}$. More precisely, we allow deviations to $T\boldsymbol{B}$ of the order of $\sqrt{T}$ times poly-log factors: this is what we refer to as "soft" constraints. Our result is that the regret may then be bounded by a quantity of order $\sqrt{T}$ times poly-log factors, at least when $\mathcal{X}$ is finite.

We do so for two reasons: first, because we do not think that this is a well-known result, and second, for pedagogic reasons, as the proof for "hard" constraints follows from adapting the proof scheme for soft constraints (see Appendix F.2).

**Proposition 1** (soft constraints). *Fix $\delta \in (0,1)$. Under Assumption 2 and with Notation* (1)*, the strategy of Box D, run with $\delta/4$ and positive slacks $b_t = \xi_{t-1,\delta/4}$, ensures that with probability at least $1 - \delta$,*

$$\sum_{t=1}^{T} \boldsymbol{c}_t \leqslant T\boldsymbol{B} + \big(2\alpha_{T,\delta/4} + \beta_{T,\delta/4} + 2\Xi_{T,\delta/4}\big)\mathbf{1} \qquad and \qquad R_T \leqslant 2\alpha_{T,\delta/4} + \beta_{T,\delta/4} + 2\Xi_{T,\delta/4}\,,$$

*where $\alpha_{T,\delta/4} = \sqrt{2T\ln\big((d+1)/(\delta/4)\big)}$.*

In particular, when $\mathcal{X}$ is finite, the deviation terms $2\alpha_{T,\delta/4} + \beta_{T,\delta/4} + 2\Xi_{T,\delta/4}$ are of order $\sqrt{T}$ up to poly-logarithmic terms, so that the bound of Proposition 1 reads: with high-probability,

$$\sum_{t=1}^{T} \boldsymbol{c}_t \leqslant T\boldsymbol{B} + \widetilde{\mathcal{O}}\big(\sqrt{T}\big) \qquad and \qquad R_T \leqslant \widetilde{\mathcal{O}}\big(\sqrt{T}\big)\,.$$

Put differently, soft-constraint satisfaction allows for $\widetilde{\mathcal{O}}\big(\sqrt{T}\big)$ regret bounds when $\mathcal{X}$ is finite.

*Proof.* As in the proofs of Lemmas 1 and 2 in Appendix B, we consider four events, each of probability at least $1 - \delta/4$: two events $\mathcal{E}_{\text{H-Az1}}$ and $\mathcal{E}_{\text{H-Az2}}$, defined below, following from applications of the Hoeffding-Azuma inequality, the favorable event $\mathcal{E}_{\text{TVD}}$ of Notation 1 with $\delta/4$, and the favorable event $\mathcal{E}_\beta$ of Assumption 2 with $\delta/4$. Namely, given the value for $\alpha_{T,\delta/4}$ proposed in the statement (note this value is slightly different from the one considered in Appendix B), we have, on the one hand, on $\mathcal{E}_{\text{H-Az1}}$,

$$\sum_{t=1}^{T} \boldsymbol{c}_t \leqslant \alpha_{T,\delta/4}\mathbf{1} + \sum_{t=1}^{T} \boldsymbol{c}(\boldsymbol{x}_t, a_t) \qquad and \qquad \sum_{t=1}^{T} r_t \geqslant -\alpha_{T,\delta/4} + \sum_{t=1}^{T} r(\boldsymbol{x}_t, a_t)\,, \qquad (20)$$

and on the other hand, on $\mathcal{E}_{\text{H-Az2}}$,

$$\sum_{t=1}^{T} \widehat{\boldsymbol{c}}_{\delta/4,t-1}^{\text{lcb}}(\boldsymbol{x}_t, a_t) \leqslant \alpha_{T,\delta/4}\mathbf{1} + \sum_{t=1}^{T} \mathbb{E}_{\boldsymbol{X}\sim\nu}\left[\sum_{a\in\mathcal{A}} \widehat{\boldsymbol{c}}_{\delta/4,t-1}^{\text{lcb}}(\boldsymbol{X}, a)\,\pi_{t,a}(\boldsymbol{X})\right]$$

and $$\sum_{t=1}^{T} \widehat{r}_{\delta/4,t-1}^{\text{ucb}}(\boldsymbol{x}_t, a_t) \geqslant -\alpha_{T,\delta/4} + \sum_{t=1}^{T} \mathbb{E}_{\boldsymbol{X}\sim\nu}\left[\sum_{a\in\mathcal{A}} \widehat{r}_{\delta/4,t-1}^{\text{ucb}}(\boldsymbol{X}, a)\,\pi_{t,a}(\boldsymbol{X})\right]. \qquad (21)$$

The first two inequalities are obtained by considering conditional expectations with respect to the past, $\boldsymbol{x}_t$ and $a_t$, while the second two inequalities follow from taking conditional expectations with respect to the past and $\boldsymbol{x}_t$; we crucially use that, by definition, $\boldsymbol{x}_t \sim \nu$ is independent from $\boldsymbol{\pi}_t$ and the estimates $\widehat{r}^{\,\mathrm{ucb}}_{\delta/4,t-1}$ and $\widehat{c}^{\,\mathrm{lcb}}_{\delta/4,t-1}$.

We first note that by $\boldsymbol{B}$–feasibility of the problem, by (3), and by Notation (1), a policy $\boldsymbol{\pi}_t$ satisfying the constraints stated in Box D exists at each round $t \geqslant 1$ on the event $\mathcal{E}_\beta \cap \mathcal{E}_{\mathrm{TVD}}$. Indeed, denoting by $\boldsymbol{\pi}^{\mathrm{feas}}$ such a $\boldsymbol{B}$–feasible policy, this existence follows from the inequalities

$$
\mathbb{E}_{\boldsymbol{X}\sim\widehat{\nu}_{\delta/4,t-1}}\left[\sum_{a\in\mathcal{A}} \widehat{\boldsymbol{c}}^{\,\mathrm{lcb}}_{\delta/4,t-1}(\boldsymbol{X},a)\,\pi^{\mathrm{feas}}_a(\boldsymbol{X})\right] \leqslant \mathbb{E}_{\boldsymbol{X}\sim\widehat{\nu}_{\delta/4,t-1}}\left[\sum_{a\in\mathcal{A}} \boldsymbol{c}(\boldsymbol{X},a)\,\pi^{\mathrm{feas}}_a(\boldsymbol{X})\right]
$$

$$
\leqslant \xi_{t-1,\delta/4} + \underbrace{\mathbb{E}_{\boldsymbol{X}\sim\nu}\left[\sum_{a\in\mathcal{A}} \boldsymbol{c}(\boldsymbol{X},a)\,\pi^{\mathrm{feas}}_a(\boldsymbol{X})\right]}_{\leqslant \boldsymbol{B}} \quad (22)
$$

and from the choice $b_t = \xi_{t-1,\delta/4}$.

Cost-wise, we then successively have, by (20), then (3) and Assumption 2, and finally (21) and Notation (1), on the event $\mathcal{E}_{\mathrm{H\text{-}Az1}} \cap \mathcal{E}_\beta \cap \mathcal{E}_{\mathrm{H\text{-}Az2}} \cap \mathcal{E}_{\mathrm{TVD}}$ of probability at least $1-\delta$,

$$
\sum_{t=1}^{T} \boldsymbol{c}_t \leqslant \alpha_{T,\delta/4}\mathbf{1} + \sum_{t=1}^{T} \boldsymbol{c}(\boldsymbol{x}_t,a_t)
$$

$$
\leqslant \big(\alpha_{T,\delta/4}+\beta_{T,\delta/4}\big)\mathbf{1} + \sum_{t=1}^{T} \widehat{\boldsymbol{c}}^{\,\mathrm{lcb}}_{\delta/4,t-1}(\boldsymbol{x}_t,a_t)
$$

$$
\leqslant \big(2\alpha_{T,\delta/4}+\beta_{T,\delta/4}\big)\mathbf{1} + \sum_{t=1}^{T}\mathbb{E}_{\boldsymbol{X}\sim\nu}\left[\sum_{a\in\mathcal{A}} \widehat{\boldsymbol{c}}^{\,\mathrm{lcb}}_{\delta/4,t-1}(\boldsymbol{X},a)\,\pi_{t,a}(\boldsymbol{X})\right]
$$

$$
\leqslant \big(2\alpha_{T,\delta/4}+\beta_{T,\delta/4}+\Xi_{T,\delta/4}\big)\mathbf{1} + \sum_{t=1}^{T}\underbrace{\mathbb{E}_{\boldsymbol{X}\sim\widehat{\nu}_{\delta/4,t-1}}\left[\sum_{a\in\mathcal{A}} \widehat{\boldsymbol{c}}^{\,\mathrm{lcb}}_{\delta/4,t-1}(\boldsymbol{X},a)\,\pi_{t,a}(\boldsymbol{X})\right]}_{\leqslant \boldsymbol{B}+b_t\mathbf{1}},
$$

where the inequalities $\leqslant \boldsymbol{B} + b_t\mathbf{1}$ follow from the definition of $\boldsymbol{\pi}_t$ in Box D. Substituting the choice $b_t = \xi_{t-1,\delta/4}$, we thus proved that on $\mathcal{E}_{\mathrm{H\text{-}Az1}} \cap \mathcal{E}_\beta \cap \mathcal{E}_{\mathrm{H\text{-}Az2}} \cap \mathcal{E}_{\mathrm{TVD}}$,

$$
\sum_{t=1}^{T} \boldsymbol{c}_t \leqslant T\boldsymbol{B} + \big(2\alpha_{T,\delta/4}+\beta_{T,\delta/4}+2\Xi_{T,\delta/4}\big)\mathbf{1}\,,
$$

as claimed.

The control for rewards mimics the steps above for costs (and then resorts to an additional argument). We obtain first that on $\mathcal{E}_{\mathrm{H\text{-}Az1}} \cap \mathcal{E}_\beta \cap \mathcal{E}_{\mathrm{H\text{-}Az2}} \cap \mathcal{E}_{\mathrm{TVD}}$,

$$
\sum_{t=1}^{T} r_t \geqslant -\big(2\alpha_{T,\delta/4}+\beta_{T,\delta/4}+\Xi_{T,\delta/4}\big) + \sum_{t=1}^{T}\mathbb{E}_{\boldsymbol{X}\sim\widehat{\nu}_{\delta/4,t-1}}\left[\sum_{a\in\mathcal{A}} \widehat{r}^{\,\mathrm{ucb}}_{\delta/4,t-1}(\boldsymbol{X},a)\,\pi_{t,a}(\boldsymbol{X})\right].
$$

We denote by $\pi^\star_{\boldsymbol{B}}$ an optimal static policy for the average cost constraints $\boldsymbol{B}$—when it exists, e.g., by (4), as soon as the problem is feasible for some $\boldsymbol{B}' < \boldsymbol{B}$; otherwise, we take a static policy achieving $\mathrm{OPT}(r,\boldsymbol{c},\boldsymbol{B})$ up to some small $e > 0$, which we let vanish in a final step of the proof. As in (22), we have, on $\mathcal{E}_\beta \cap \mathcal{E}_{\mathrm{TVD}}$,

$$
\mathbb{E}_{\boldsymbol{X}\sim\widehat{\nu}_{\delta/4,t-1}}\left[\sum_{a\in\mathcal{A}} \widehat{\boldsymbol{c}}^{\,\mathrm{lcb}}_{\delta/4,t-1}(\boldsymbol{X},a)\,\pi^\star_{\boldsymbol{B},a}(\boldsymbol{X})\right] \leqslant \mathbb{E}_{\boldsymbol{X}\sim\widehat{\nu}_{\delta/4,t-1}}\left[\sum_{a\in\mathcal{A}} \boldsymbol{c}(\boldsymbol{X},a)\,\pi^\star_{\boldsymbol{B},a}(\boldsymbol{X})\right]
$$

$$
\leqslant \xi_{t-1,\delta/4}\mathbf{1} + \underbrace{\mathbb{E}_{\boldsymbol{X}\sim\nu}\left[\sum_{a\in\mathcal{A}} \boldsymbol{c}(\boldsymbol{X},a)\,\pi^\star_{\boldsymbol{B},a}(\boldsymbol{X})\right]}_{\leqslant \boldsymbol{B}},
$$

where the $\leqslant \boldsymbol{B}$ inequality follows from the definition of $\boldsymbol{\pi}_{\boldsymbol{B}}^\star$. Thanks to the choice $b_t = \xi_{t-1,\delta/4}$, we have, by definition of $\boldsymbol{\pi}_t$ as some optimal policy in Box D and on $\mathcal{E}_\beta \cap \mathcal{E}_{\text{TVD}}$,

$$\mathbb{E}_{\boldsymbol{X}\sim\widehat{\nu}_{\delta/4,t-1}}\left[\sum_{a\in\mathcal{A}}\widehat{r}_{\delta/4,t-1}^{\,\text{ucb}}(\boldsymbol{X},a)\,\pi_{t,a}(\boldsymbol{X})\right] \geqslant \mathbb{E}_{\boldsymbol{X}\sim\widehat{\nu}_{\delta/4,t-1}}\left[\sum_{a\in\mathcal{A}}\widehat{r}_{\delta/4,t-1}^{\,\text{ucb}}(\boldsymbol{X},a)\,\pi_{\boldsymbol{B},a}^\star(\boldsymbol{X})\right].$$

Again by (3) and Notation (1), we have, on $\mathcal{E}_\beta \cap \mathcal{E}_{\text{TVD}}$,

$$\mathbb{E}_{\boldsymbol{X}\sim\widehat{\nu}_{\delta/4,t-1}}\left[\sum_{a\in\mathcal{A}}\widehat{r}_{\delta/4,t-1}^{\,\text{ucb}}(\boldsymbol{X},a)\,\pi_{\boldsymbol{B},a}^\star(\boldsymbol{X})\right] \geqslant \mathbb{E}_{\boldsymbol{X}\sim\widehat{\nu}_{\delta/4,t-1}}\left[\sum_{a\in\mathcal{A}}r(\boldsymbol{X},a)\,\pi_{\boldsymbol{B},a}^\star(\boldsymbol{X})\right]$$

$$\geqslant -\xi_{t-1,\delta/4} + \underbrace{\mathbb{E}_{\boldsymbol{X}\sim\nu}\left[\sum_{a\in\mathcal{A}}r(\boldsymbol{X},a)\,\pi_{\boldsymbol{B},a}^\star(\boldsymbol{X})\right]}_{=\,\text{OPT}(r,\boldsymbol{c},\boldsymbol{B})}.$$

Collecting all bounds, we proved that on $\mathcal{E}_{\text{H-Az1}} \cap \mathcal{E}_\beta \cap \mathcal{E}_{\text{H-Az2}} \cap \mathcal{E}_{\text{TVD}}$,

$$\sum_{t=1}^{T} r_t \geqslant T\,\text{OPT}(r,\boldsymbol{c},\boldsymbol{B}) - \left(2\alpha_{T,\delta/4} + \beta_{T,\delta/4} + 2\Xi_{T,\delta/4}\right),$$

which corresponds to the claimed regret bound. $\qquad\qquad\square$

## F.2 Analysis with "hard" constraints and a null-cost action

We now turn our attention to the main kind of result that we want to achieve: when constraints must be strictly satisfied—which we refer to as "hard" constraints. For the sake of simplicity, we do so for now in the presence of a null-cost action; Appendix F.3 explains how the analysis may be generalized to cases without such null-cost actions.

The following result corresponds to the combination of Theorem 1 with Corollary 1, and also, to Li and Stoltz [2022, main result: Theorem 2].

**Theorem 3** (hard constraints). *Fix $\delta \in (0,1)$. We consider the strategy of Box D, run with $\delta/4$ and negative slacks all equal to*

$$b_t \equiv -\overline{\Delta}_{T,\delta/4}, \qquad where \qquad \overline{\Delta}_{T,\delta/4} \stackrel{\text{def}}{=} \frac{2\alpha_{T,\delta/4} + \beta_{T,\delta/4} + \Xi_{T,\delta/4}}{T}$$

*and $\alpha_{T,\delta/4} = \sqrt{2T\ln\big((d+1)/(\delta/4)\big)}$. Assume that a null-cost action exists, that $\overline{\Delta}_{T,\delta/4} < \min\boldsymbol{B}$, that Assumption 2 holds, and use Notation (1). Then, with probability at least $1-\delta$,*

$$\sum_{t=1}^{T} \boldsymbol{c}_t \leqslant T\boldsymbol{B} \quad and \quad R_T \leqslant \left(2\alpha_{T,\delta/4} + \beta_{T,\delta/4} + 2\Xi_{T,\delta/4}\right)\left(1 + \frac{\text{OPT}(r,\boldsymbol{c},\boldsymbol{B}) - \text{OPT}(r,\boldsymbol{c},\boldsymbol{0})}{\min\boldsymbol{B}}\right).$$

*Proof.* We explain how the proof of Proposition 1 may be adapted. We first justify the existence at each round $t \geqslant 1$ of a policy $\boldsymbol{\pi}_t$ satisfying the cost constraints stated in Box D: for the null-cost action $a_{\text{null}}$, we may impose that, for all $\boldsymbol{x} \in \mathcal{X}$, all $t \geqslant 0$, and all $\delta \in (0,1)$,

$$\widehat{\boldsymbol{c}}_t(\boldsymbol{x},a_{\text{null}}) = \boldsymbol{0} \quad \text{and} \quad \varepsilon_t(\boldsymbol{x},a_{\text{null}},\delta) = 0, \qquad \text{so that} \quad \widehat{\boldsymbol{c}}_{\delta,t}^{\,\text{lcb}}(\boldsymbol{x},a_{\text{null}}) = \boldsymbol{0} \text{ a.s.};$$

this shows that at least the static policy $\boldsymbol{\pi}^{\text{null}}$ always playing $a_{\text{null}}$ satisfying the defining cost-constraints for $\boldsymbol{\pi}_t$. (Alternatively, we may note that the policy $\overline{\boldsymbol{\pi}}_t$ defined below also satisfies the cost constraints stated in Box D, on the high-probability event considered.)

We then handle total-cost constraints similarly as in the proof of Proposition 1 and obtain that on the event $\mathcal{E}_{\text{H-Az1}} \cap \mathcal{E}_\beta \cap \mathcal{E}_{\text{H-Az2}} \cap \mathcal{E}_{\text{TVD}}$ of probability at least $1-\delta$,

$$\sum_{t=1}^{T} \boldsymbol{c}_t \leqslant \left(2\alpha_{T,\delta/4} + \beta_{T,\delta/4} + \Xi_{T,\delta/4}\right)\boldsymbol{1} + \sum_{t=1}^{T}\underbrace{\mathbb{E}_{\boldsymbol{X}\sim\widehat{\nu}_{\delta/4,t-1}}\left[\sum_{a\in\mathcal{A}}\widehat{\boldsymbol{c}}_{\delta/4,t-1}^{\,\text{lcb}}(\boldsymbol{X},a)\,\pi_{t,a}(\boldsymbol{X})\right]}_{\leqslant\,\boldsymbol{B}-\overline{\Delta}_{T,\delta/4}\boldsymbol{1}} \leqslant T\boldsymbol{B},$$

where this time, the slacks $b_t = -\overline{\Delta}_{T,\delta/4}$ are negative and were set to cancel out the positive deviation terms. Similarly, on $\mathcal{E}_{\text{H-Az1}} \cap \mathcal{E}_\beta \cap \mathcal{E}_{\text{H-Az2}} \cap \mathcal{E}_{\text{TVD}}$,

$$\sum_{t=1}^{T} r_t \geqslant -T\overline{\Delta}_{T,\delta/4} + \sum_{t=1}^{T} \mathbb{E}_{\boldsymbol{X} \sim \widehat{\nu}_{\delta/4,t-1}} \left[ \sum_{a \in \mathcal{A}} \widehat{r}_{\delta/4,t-1}^{\text{ucb}}(\boldsymbol{X}, a) \, \pi_{t,a}(\boldsymbol{X}) \right]. \tag{23}$$

Now, the main modification to the proof of Proposition 1 is that (with its notation) we rather consider the policy

$$\overline{\boldsymbol{\pi}}_t = (1 - w_t)\boldsymbol{\pi}_{\boldsymbol{B}}^\star + w_t \boldsymbol{\pi}^{\text{null}}, \qquad \text{where} \quad w_t = \min\left\{ \frac{\overline{\Delta}_{T,\delta/4} + \xi_{t-1,\delta/4}}{\min \boldsymbol{B}}, \; 1 \right\}.$$

As in (22), we see that this policy satisfies, on $\mathcal{E}_\beta \cap \mathcal{E}_{\text{TVD}}$:

$$\mathbb{E}_{\boldsymbol{X} \sim \widehat{\nu}_{\delta/4,t-1}} \left[ \sum_{a \in \mathcal{A}} \widehat{\boldsymbol{c}}_{\delta/4,t-1}^{\text{lcb}}(\boldsymbol{X}, a) \, \overline{\pi}_{t,a}(\boldsymbol{X}) \right] \leqslant \mathbb{E}_{\boldsymbol{X} \sim \widehat{\nu}_{\delta/4,t-1}} \left[ \sum_{a \in \mathcal{A}} \boldsymbol{c}(\boldsymbol{X}, a) \, \overline{\pi}_{t,a}(\boldsymbol{X}) \right]$$

$$\leqslant \xi_{t-1,\delta/4} \mathbf{1} + \underbrace{\mathbb{E}_{\boldsymbol{X} \sim \nu} \left[ \sum_{a \in \mathcal{A}} \boldsymbol{c}(\boldsymbol{X}, a) \, \overline{\pi}_{t,a}(\boldsymbol{X}) \right]}_{\leqslant (1 - w_t)\boldsymbol{B}}.$$

By using $\boldsymbol{B} \geqslant (\min \boldsymbol{B})\mathbf{1}$ and since we assumed that $\overline{\Delta}_{T,\delta/4} < \min \boldsymbol{B}$, we may continue the series of inequalities as follows:

$$\xi_{t-1,\delta/4} \mathbf{1} + (1 - w_t)\boldsymbol{B} \leqslant \boldsymbol{B} + \xi_{t-1,\delta/4} \mathbf{1} - w_t (\min \boldsymbol{B})\mathbf{1}$$

$$= \boldsymbol{B} - \min\{\overline{\Delta}_{T,\delta/4}, \; \min \boldsymbol{B} - \xi_{t-1,\delta/4}\}\mathbf{1}$$

$$\leqslant \boldsymbol{B} - \min\{\overline{\Delta}_{T,\delta/4}, \; \overline{\Delta}_{T,\delta/4} - \xi_{t-1,\delta/4}\}\mathbf{1} = \boldsymbol{B} + b_t \mathbf{1},$$

meaning that $\overline{\boldsymbol{\pi}}_t$ is a policy satisfying the constraints stated in Step 2 of Box D on round $t \geqslant 1$. The consequence is that, first by definition of $\boldsymbol{\pi}_t$ as a maximizer and then by the optimistic estimates (3) and Notation (1), on $\mathcal{E}_\beta \cap \mathcal{E}_{\text{TVD}}$,

$$\mathbb{E}_{\boldsymbol{X} \sim \widehat{\nu}_{\delta/4,t-1}} \left[ \sum_{a \in \mathcal{A}} \widehat{r}_{\delta/4,t-1}^{\text{ucb}}(\boldsymbol{X}, a) \, \pi_{t,a}(\boldsymbol{X}) \right]$$

$$\geqslant \mathbb{E}_{\boldsymbol{X} \sim \widehat{\nu}_{\delta/4,t-1}} \left[ \sum_{a \in \mathcal{A}} \widehat{r}_{\delta/4,t-1}^{\text{ucb}}(\boldsymbol{X}, a) \, \overline{\pi}_{t,a}(\boldsymbol{X}) \right]$$

$$\geqslant -\xi_{t-1,\delta/4} + \mathbb{E}_{\boldsymbol{X} \sim \nu} \left[ \sum_{a \in \mathcal{A}} r(\boldsymbol{X}, a) \, \overline{\pi}_{t,a}(\boldsymbol{X}) \right]$$

$$= -\xi_{t-1,\delta/4} + (1 - w_t)\text{OPT}(r, \boldsymbol{c}, \boldsymbol{B}) + w_t \text{OPT}(r, \boldsymbol{c}, \boldsymbol{0})$$

$$= \text{OPT}(r, \boldsymbol{c}, \boldsymbol{B}) - \xi_{t-1,\delta/4} - \min\left\{ \frac{\overline{\Delta}_{T,\delta/4} + \xi_{t-1,\delta/4}}{\min \boldsymbol{B}}, \; 1 \right\} \big(\text{OPT}(r, \boldsymbol{c}, \boldsymbol{B}) - \text{OPT}(r, \boldsymbol{c}, \boldsymbol{0})\big),$$

where $\text{OPT}(r, \boldsymbol{c}, \boldsymbol{0})$ refers to the expect reward achieved by the null-cost action $a_{\text{null}}$. After combination with (23) and summation, we get that on $\mathcal{E}_{\text{H-Az1}} \cap \mathcal{E}_\beta \cap \mathcal{E}_{\text{H-Az2}} \cap \mathcal{E}_{\text{TVD}}$:

$$\sum_{t=1}^{T} r_t - T \, \text{OPT}(r, \boldsymbol{c}, \boldsymbol{B})$$

$$\geqslant -T\overline{\Delta}_{T,\delta/4} - \Xi_{T,\delta/4} - \sum_{t=1}^{T} \min\left\{ \frac{\overline{\Delta}_{T,\delta/4} + \xi_{t-1,\delta/4}}{\min \boldsymbol{B}}, \; 1 \right\} \big(\text{OPT}(r, \boldsymbol{c}, \boldsymbol{B}) - \text{OPT}(r, \boldsymbol{c}, \boldsymbol{0})\big)$$

$$\geqslant -\big(T\overline{\Delta}_{T,\delta/4} + \Xi_{T,\delta/4}\big) \left(1 + \frac{\text{OPT}(r, \boldsymbol{c}, \boldsymbol{B}) - \text{OPT}(r, \boldsymbol{c}, \boldsymbol{0})}{\min \boldsymbol{B}}\right),$$

which corresponds to the stated regret bound. $\qquad \square$

### F.3 General analysis with "hard" constraints

We finally generalize Theorem 3 and get a result corresponding to Theorem 1 combined with Lemma 3.

Here, we "mix" the slacks $+\xi_{t-1,\delta/4}$ and $-\overline{\Delta}_{T,\delta/4}$ of Appendices F.1 and F.2, with a slight modification of $\overline{\Delta}_{T,\delta/4}$ to compensate for the $\xi_{t-1,\delta/4}$: we rather have a $2\Xi_{T,\delta/4}$ term in the numerator of $\overline{\Delta}'_{T,\delta/4}$, compared to the $\Xi_{T,\delta/4}$ term in the numerator of $\overline{\Delta}_{T,\delta/4}$.

**Theorem 4** (hard constraints)**.** *Fix $\delta \in (0,1)$. We consider the strategy of Box D, run with $\delta/4$ and signed slacks (depending on $t \geqslant 1$)*

$$b_t = -\overline{\Delta}'_{T,\delta/4} + \xi_{t-1,\delta/4} \qquad where \qquad \overline{\Delta}'_{T,\delta/4} = \frac{2\alpha_{T,\delta/4} + \beta_{T,\delta/4} + 2\Xi_{T,\delta/4}}{T}$$

*and $\alpha_{T,\delta/4} = \sqrt{2T \ln\big((d+1)/(\delta/4)\big)}$. Assume that the problem is $(\boldsymbol{B} - \boldsymbol{m})$–feasible for some $\boldsymbol{m}$ that does not need to be known by the strategy, with $\boldsymbol{B} - \boldsymbol{m} \leqslant \boldsymbol{B} - \overline{\Delta}'_{T,\delta/4}\mathbf{1}$. Then, under Assumption 2 and with Notation (1), with probability at least $1 - \delta$,*

$$\sum_{t=1}^{T} \boldsymbol{c}_t \leqslant T\boldsymbol{B}$$

*and* $$R_T \leqslant \big(2\alpha_{T,\delta/4} + \beta_{T,\delta/4} + 2\Xi_{T,\delta/4}\big)\left(1 + \frac{\mathrm{OPT}(r,\boldsymbol{c},\boldsymbol{B}) - \mathrm{OPT}(r,\boldsymbol{c},\boldsymbol{B}-\boldsymbol{m})}{\min \boldsymbol{m}}\right).$$

*Proof.* We rather sketch the differences to the proofs of Theorem 3 and Proposition 1. We denote by $\mathcal{E}_{\mathrm{all}}$ the event of probability at least $1 - \delta$ obtained as the intersection of four convenient events of probability each at least $1 - \delta/4$. All inequalities below hold on $\mathcal{E}_{\mathrm{all}}$ but for the sake of brevity, we will not highlight this fact each time.

We denote by $\boldsymbol{\pi}^{\mathrm{feas}}$ a (quasi-)optimal static policy among the ones achieving expected costs smaller than $\boldsymbol{B} - \boldsymbol{m}$; it therefore ensures that its expected costs are smaller than $\boldsymbol{B} - \boldsymbol{m}$ and achieves an expected reward of $\mathrm{OPT}(r,\boldsymbol{c},\boldsymbol{B}-\boldsymbol{m}) - e$, possibly up to a small factor $e \geqslant 0$ which we let vanish. We note that for each round $t \geqslant 1$,

$$\mathbb{E}_{\boldsymbol{X}\sim\widehat{\nu}_{\delta,t-1}}\left[\sum_{a\in\mathcal{A}}\widehat{\boldsymbol{c}}^{\mathrm{lcb}}_{\delta,t-1}(\boldsymbol{X},a)\,\pi^{\mathrm{feas}}_a(\boldsymbol{X})\right] \leqslant \xi_{t-1,\delta/T}\mathbf{1} + \mathbb{E}_{\boldsymbol{X}\sim\nu}\left[\sum_{a\in\mathcal{A}}\boldsymbol{c}(\boldsymbol{X},a)\,\pi^{\mathrm{feas}}_a(\boldsymbol{X})\right]$$

$$\leqslant \xi_{t-1,\delta/T}\mathbf{1} + \boldsymbol{B} - \boldsymbol{m}\mathbf{1} \leqslant \boldsymbol{B} + b_t\mathbf{1},$$

given the definition $b_t = -\overline{\Delta}'_{T,\delta/4} + \xi_{t-1,\delta/4}$ and the assumption $\boldsymbol{B} - \boldsymbol{m} \leqslant \boldsymbol{B} - \overline{\Delta}'_{T,\delta/4}\mathbf{1}$. Thus, on $\mathcal{E}_{\mathrm{all}}$, the strategy $\boldsymbol{\pi}_t$ is indeed defined at each round $t \geqslant 1$ by the optimization problem stated in Step 2 of Box D (and not in an arbitrary manner).

Cost-wise, we thus have

$$\sum_{t=1}^{T} \boldsymbol{c}_t \leqslant \big(2\alpha_{T,\delta/4} + \beta_{T,\delta/4} + \Xi_{T,\delta/4}\big)\mathbf{1} + \sum_{t=1}^{T}\overbrace{\mathbb{E}_{\boldsymbol{X}\sim\widehat{\nu}_{\delta/4,t-1}}\left[\sum_{a\in\mathcal{A}}\widehat{\boldsymbol{c}}^{\mathrm{lcb}}_{\delta/4,t-1}(\boldsymbol{X},a)\,\pi_{t,a}(\boldsymbol{X})\right]}^{\leqslant \boldsymbol{B}+b_t\mathbf{1}}$$

$$\leqslant T\boldsymbol{B} + \big(2\alpha_{T,\delta/4} + \beta_{T,\delta/4} + \Xi_{T,\delta/4}\big)\mathbf{1} - T\overline{\Delta}_{T,\delta/4} + \Xi_{T,\delta/4} = T\boldsymbol{B},$$

where the final equality follows from the definition of $\overline{\Delta}'_{T,\delta/4}$, which involves a $2\Xi_{T,\delta/4}$ term in its numerator.

Reward-wise, we introduce for each $t \geqslant 1$,

$$\overline{\boldsymbol{\pi}}_t = (1 - w_t)\boldsymbol{\pi}^{\star}_{\boldsymbol{B}} + w_t\boldsymbol{\pi}^{\mathrm{feas}}, \qquad where \quad w_t = \min\left\{\frac{\overline{\Delta}'_{T,\delta/4}}{\min \boldsymbol{m}},\ 1\right\},$$

whose empirical expected cost at round $t \geqslant 1$ is smaller than

$$\mathbb{E}_{\boldsymbol{X}\sim\widehat{\nu}_{\delta/4,t-1}}\left[\sum_{a\in\mathcal{A}}\widehat{\boldsymbol{c}}^{\mathrm{lcb}}_{\delta/4,t-1}(\boldsymbol{X},a)\,\overline{\pi}_{t,a}(\boldsymbol{X})\right] \leqslant \xi_{t-1,\delta/4}\mathbf{1} + \mathbb{E}_{\boldsymbol{X}\sim\nu}\left[\sum_{a\in\mathcal{A}}\boldsymbol{c}(\boldsymbol{X},a)\,\overline{\pi}_{t,a}(\boldsymbol{X})\right]$$

$$\leqslant \xi_{t-1,\delta/4}\mathbf{1} + (1 - w_t)\boldsymbol{B} + w_t(\boldsymbol{B} - \boldsymbol{m})$$

$$= \boldsymbol{B} + \xi_{t-1,\delta/4}\mathbf{1} - w_t\boldsymbol{m}.$$

By using $\boldsymbol{m} \geqslant (\min \boldsymbol{m}) \mathbf{1}$ and thanks to the assumption $\boldsymbol{B} - \boldsymbol{m} \leqslant \boldsymbol{B} - \overline{\Delta}'_{T,\delta/4} \mathbf{1}$, which can be equivalently formulated as $\min \boldsymbol{m} \geqslant \overline{\Delta}'_{T,\delta/4}$, we may continue this series of inequalities by

$$
\begin{aligned}
\boldsymbol{B} + \xi_{t-1,\delta/4} \mathbf{1} - w_t \boldsymbol{m} &\leqslant \boldsymbol{B} - \min\Big\{ \overline{\Delta}'_{T,\delta/4} - \xi_{t-1,\delta/4}, \ \min \boldsymbol{m} - \xi_{t-1,\delta/4} \Big\} \mathbf{1} \\
&\leqslant \boldsymbol{B} - \min\Big\{ \overline{\Delta}'_{T,\delta/4} - \xi_{t-1,\delta/4}, \ \overline{\Delta}'_{T,\delta/4} - \xi_{t-1,\delta/4} \Big\} \mathbf{1} = \boldsymbol{B} + b_t \mathbf{1} \,,
\end{aligned}
$$

meaning that $\overline{\boldsymbol{\pi}}_t$ is a policy satisfying the constraints stated in Step 2 of Box D on round $t \geqslant 1$. In particular, by definition of $\boldsymbol{\pi}_t$ as a maximizer,

$$
\begin{aligned}
\mathbb{E}_{\boldsymbol{X} \sim \widehat{\nu}_{\delta/4,t-1}} &\left[ \sum_{a \in \mathcal{A}} \widehat{r}^{\,\mathrm{ucb}}_{\delta/4,t-1}(\boldsymbol{X}, a)\, \pi_{t,a}(\boldsymbol{X}) \right] \\
&\geqslant \mathbb{E}_{\boldsymbol{X} \sim \widehat{\nu}_{\delta/4,t-1}} \left[ \sum_{a \in \mathcal{A}} \widehat{r}^{\,\mathrm{ucb}}_{\delta/4,t-1}(\boldsymbol{X}, a)\, \overline{\pi}_{t,a}(\boldsymbol{X}) \right] \\
&\geqslant -\xi_{t-1,\delta/4} + \mathbb{E}_{\boldsymbol{X} \sim \nu} \left[ \sum_{a \in \mathcal{A}} r(\boldsymbol{X}, a)\, \overline{\pi}_{t,a}(\boldsymbol{X}) \right] \\
&\geqslant -\xi_{t-1,\delta/4} + \big( (1 - w_t) \mathrm{OPT}(r, \boldsymbol{c}\,\boldsymbol{B}) + w_t \mathrm{OPT}(r, \boldsymbol{c}\,\boldsymbol{B} - \boldsymbol{m}) \big) \\
&= \mathrm{OPT}(r, \boldsymbol{c}\,\boldsymbol{B}) - \min\left\{ \frac{\overline{\Delta}'_{T,\delta/4}}{\min \boldsymbol{m}}, \ 1 \right\} \big( \mathrm{OPT}(r, \boldsymbol{c}\,\boldsymbol{B}) - \mathrm{OPT}(r, \boldsymbol{c}\,\boldsymbol{B} - \boldsymbol{m}) \big) - \xi_{t-1,\delta/4} \,. \quad (24)
\end{aligned}
$$

Finally, by combining (23) and (24),

$$
\begin{aligned}
\sum_{t=1}^{T} r_t &\geqslant -\big( \overbrace{2\alpha_{T,\delta/4} + \beta_{T,\delta/4} + \Xi_{T,\delta/4}}^{= T\overline{\Delta}'_{T,\delta/4} - \Xi_{T,\delta/4}} \big) \mathbf{1} + \sum_{t=1}^{T} \mathbb{E}_{\boldsymbol{X} \sim \widehat{\nu}_{\delta/4,t-1}} \left[ \sum_{a \in \mathcal{A}} \widehat{r}^{\,\mathrm{ucb}}_{\delta/4,t-1}(\boldsymbol{X}, a)\, \pi_{t,a}(\boldsymbol{X}) \right] \\
&\geqslant T\,\mathrm{OPT}(r, \boldsymbol{c}\,\boldsymbol{B}) + \Xi_{T,\delta/4} - T\overline{\Delta}'_{T,\delta/4} \left( 1 + \frac{\mathrm{OPT}(r, \boldsymbol{c}\,\boldsymbol{B}) - \mathrm{OPT}(r, \boldsymbol{c}\,\boldsymbol{B} - \boldsymbol{m})}{\min \boldsymbol{m}} \right) - \Xi_{T,\delta/4},
\end{aligned}
$$

as claimed. $\qquad\square$

## G   Numerical experiments: full description

This appendix reports numerical simulations performed on simulated data with the motivating example described in Chohlas-Wood et al. [2021] and alluded at in Section 2.1. These simulations are for the sake of illustration only.

A brief summary of the applicative background in AI for justice is the following. The learner wants to maximize the total number of appearances to court for people of concern. To achieve this goal, the learner is able to provide, or not, some transportation assistance: rideshare assistance (the highest level of help), or a transit voucher (a more modest level of help). There are of course budget limits on these assistance means, and the learner also wants to control how (un)fair the assistance policy is, in terms of subgroups of the population, while maximizing the total number of appearances. Some subgroups take a better advantage of assistance to appear in court, thus, without the fairness control, all assistance would go to these groups. The fairness costs described in Section 2.1 force the learner to perform some tradeoff between spending all assistance on most reactive subgroups and spending it equally among subgroups.

**Outline of this appendix.**   We first recall the experimental setting of Chohlas-Wood et al. [2021]—in particular, how contexts, rewards, and costs are generated in their simulations (we cannot replicate their study on real data, which is not accessible). We then specify the strategies we implemented and how we tuned them—this includes describing the estimation procedure discussed in Section 2.2. We finally report the performance observed, in terms of (average) rewards and costs.

## G.1 The experimental setting of Chohlas-Wood et al. [2021]

We follow strictly the experimental setting of Chohlas-Wood et al. [2021] as provided in the public repository `https://github.com/stanford-policylab/learning-to-be-fair`. This experimental setting, as reverse-engineered from the code, deals with purely simulated data and seems actually inconsistent with the description made in Chohlas-Wood et al. [2021, Section 5.4 and Appendix E], which would anyway rely on proprietary data that could not be made public.

**Context generation.** Each individual is described by four variables $\boldsymbol{x}$: age, proximity, poverty, and group, which are to be read in $\boldsymbol{x}$ in its components, referred to as $x_{\text{age}}$, $x_{\text{prox}}$, $x_{\text{pov}}$ and $\text{gr}(\boldsymbol{x}) = x_{\text{group}}$. The first three variables are assumed to be normalized and are simulated independently, according to uniform distributions on $[0, 1]$. Groups are also simulated independently of these three variables: two groups are assumed, with respective probabilities of $1/2$. This defines the context distribution $\nu$.

As we describe now, assistance has stronger impact on group 0 than in group 1.

**Cost generation.** We recall that three actions are available: offering some rideshare assistance (action $a_{\text{ride}}$), providing a transportation voucher (action $a_{\text{voucher}}$), or providing no help (which is a control situation: action $a_{\text{control}}$). The associated spending costs are deterministic and do not depend on $\boldsymbol{x}$, and there are two separate budgets for rideshares and vouchers. More precisely, at round $t$, upon taking action $a_t$, the following deterministic spending costs are suffered:

$$c_{\text{ride}}(\boldsymbol{x}_t, a_t) = \mathbb{1}_{\{a_t = a_{\text{ride}}\}} \qquad \text{and} \qquad c_{\text{voucher}}(\boldsymbol{x}_t, a_t) = \mathbb{1}_{\{a_t = a_{\text{voucher}}\}} \,,$$

where the corresponding average budgets are $B_{\text{ride}} = 0.05$ and $B_{\text{voucher}} = 0.20$.

We measure the extent of unfair allocation of spendings among groups with the following eight fairness costs: a first series of four fairness costs is given by

$$2\, \mathbb{1}_{\{a_t = a_{\text{ride}}\}} \mathbb{1}_{\{\text{gr}(\boldsymbol{x}_t) = 0\}} - \mathbb{1}_{\{a_t = a_{\text{ride}}\}} \,, \qquad 2\, \mathbb{1}_{\{a_t = a_{\text{ride}}\}} \mathbb{1}_{\{\text{gr}(\boldsymbol{x}_t) = 1\}} - \mathbb{1}_{\{a_t = a_{\text{ride}}\}} \,,$$

$$2\, \mathbb{1}_{\{a_t = a_{\text{voucher}}\}} \mathbb{1}_{\{\text{gr}(\boldsymbol{x}_t) = 0\}} - \mathbb{1}_{\{a_t = a_{\text{voucher}}\}} \,, \qquad 2\, \mathbb{1}_{\{a_t = a_{\text{voucher}}\}} \mathbb{1}_{\{\text{gr}(\boldsymbol{x}_t) = 1\}} - \mathbb{1}_{\{a_t = a_{\text{voucher}}\}} \,,$$

and the second series is given by the opposite values of these costs. We denote by $\tau$ the corresponding average cost constraints $\tau$, and set $\tau = 10^{-7}$ or $\tau = 0.025$ in our experiments.

All in all, the global (spending and fairness) vector costs $\boldsymbol{c}_t$ takes deterministic values in $\mathbb{R}^{10}$. The vector cost function $\boldsymbol{c}$ is fully known to the learner, and no estimation is needed.

**Reward generation.** The rewards are binary: $r_t = 1$ if the $t$–individual appeared in court, and $r_t = 0$ otherwise. That is, the expected reward equals the probability of appearance. A logistic regression model is assumed: denoting by $\Phi(u) = 1/(1 + e^u)$, we assume

$$r(\boldsymbol{x}, a_{\text{control}}) = \Phi(-x_{\text{age}}),$$

$$r(\boldsymbol{x}, a_{\text{voucher}}) = \Phi(-x_{\text{age}} + 2x_{\text{prox}}) \mathbb{1}_{\{\text{gr}(\boldsymbol{x}) = 0\}} + \Phi(-x_{\text{age}} + x_{\text{prox}}) \mathbb{1}_{\{\text{gr}(\boldsymbol{x}) = 1\}} \,,$$

$$r(\boldsymbol{x}, a_{\text{ride}}) = \Phi(-x_{\text{age}} + 4x_{\text{pov}}) \mathbb{1}_{\{\text{gr}(\boldsymbol{x}) = 0\}} + \Phi(-x_{\text{age}} + 2x_{\text{pov}}) \mathbb{1}_{\{\text{gr}(\boldsymbol{x}) = 1\}} \,.$$

We may write these six equalities in a compact format as:

$$r(\boldsymbol{x}, a) = \Phi\big(\varphi(\boldsymbol{x}, a)^{\mathrm{T}} \mu_\star\big)$$

$$\text{where} \qquad \varphi(\boldsymbol{x}, a) = \begin{bmatrix} x_{\text{age}} \\ x_{\text{prox}}\, \mathbb{1}_{\{a = a_{\text{voucher}}\}} \\ x_{\text{prox}}\, \mathbb{1}_{\{a = a_{\text{voucher}}\}} \mathbb{1}_{\{\text{gr}(\boldsymbol{x}) = 0\}} \\ x_{\text{pov}}\, \mathbb{1}_{\{a = a_{\text{ride}}\}} \\ x_{\text{pov}}\, \mathbb{1}_{\{a = a_{\text{ride}}\}} \mathbb{1}_{\{\text{gr}(\boldsymbol{x}) = 0\}} \end{bmatrix} \quad \text{and} \quad \mu_\star = \begin{bmatrix} -1 \\ 1 \\ 1 \\ 2 \\ 2 \end{bmatrix} .$$

The learner ignores the coefficients $\mu_\star$ of this structure and only knows that expected rewards are of the form $\Phi\big(\varphi(\boldsymbol{x}, a)^{\mathrm{T}} \boldsymbol{\theta}\big)$ for some parameter $\boldsymbol{\theta} \in \mathbb{R}^5$. The learner will estimate the reward function $r$ by estimating $\mu_\star$.

**Reward estimation.** We deal with a logistic model and follow the methodology described in Li and Stoltz [2022]; see Modeling 2 in Section 2.2. In particular, the parameters $\mu_\star$ are estimated, after each round $t \geqslant 1$, thanks to the maximum likelihood estimator

$$\widehat{\boldsymbol{\mu}}_t \in \arg\max_{\boldsymbol{\mu} \in \mathbb{R}^5} \sum_{s=1}^{t} r_s \Phi\big(\varphi(a_s, \boldsymbol{x}_s)^{\mathrm{T}} \boldsymbol{\mu}\big) + (1 - r_s) \ln\Big(1 - \Phi\big(\varphi(a_s, \boldsymbol{x}_s)^{\mathrm{T}} \boldsymbol{\mu}\big)\Big) - \frac{\lambda^{\text{logistic}}}{2} \|\boldsymbol{\mu}\|^2 \,,$$

where $\lambda^{\text{logistic}} \geqslant 0$ is a regularization factor. We define as $\widehat{r}_t(\boldsymbol{x}, a) = \Phi\big(\boldsymbol{\varphi}(\boldsymbol{x}, a)^{\mathrm{T}} \widehat{\boldsymbol{\mu}}_t\big)$ the estimated expected rewards, and the associated uniform estimation errors (see Assumption 2) are of the form

$$\varepsilon_t(\boldsymbol{x}, a, \delta) = C_\delta\big(1 + \ln t\big)\sqrt{\boldsymbol{\varphi}(a, \boldsymbol{x})^{\mathrm{T}} V_t^{-1} \boldsymbol{\varphi}(a, \boldsymbol{x})}$$

$$\text{where} \qquad V_t = \sum_{s=1}^{t} \boldsymbol{\varphi}(a_s, \boldsymbol{x}_s)\,\boldsymbol{\varphi}(a_s, \boldsymbol{x}_s)^{\mathrm{T}} + \lambda^{\text{logistic}}\mathrm{I}_5\,,$$

where $\mathrm{I}_5$ is the $5 \times 5$ identity matrix.

In our simulations, we tested a range of values and picked (in hindsight) the well-performing values $C_\delta = 0.025$ and $\lambda^{\text{logistic}} = 0$.

## G.2 Strategies implemented in this numerical study

We run all these strategies not with the total-cost constraints

$$\boldsymbol{B} = (0.05,\ 0.2,\ \tau,\ \tau,\ \tau,\ \tau)\,, \qquad \text{where} \qquad \tau \in \{10^{-7},\ 0.025\}\,,$$

but take a margin $b = 0.005$ and use $\boldsymbol{B}' = \boldsymbol{B} - (b, b, 0, 0, 0, 0)$ instead. This is a slightly different way of taking some margin on the average cost constraint: we do so because we are not aiming for a strict respect of the fairness constraints but rather want to report the level of violation on it. Again, we tried a range of values for $b$ (between 0.001 to 0.01) and this value of 0.005 led to a good balance between (lack of) total-cost constraint violations and rewards.

**Performance of optimal static policies.** We use $\text{OPT}(r, \boldsymbol{c}, \boldsymbol{B})$ as the benchmark in the definition of regret; our methodology also reveals that $\text{OPT}(r, \boldsymbol{c}, \boldsymbol{B}')$ is another benchmark, see the discussion in Section 4. We report both values on our graphs. To compute them, we proceed as follows, e.g., for $\boldsymbol{B}$. As computing directly the minimum stated in (4) is difficult, even when fully knowing the distribution $\nu$, we compute 100 estimates

$$\text{OPT}^{(j)}(r, \boldsymbol{c}, \boldsymbol{B})\,, \qquad \text{which we average out into} \qquad \widehat{\text{OPT}}(r, \boldsymbol{c}, \boldsymbol{B}) = \frac{1}{100}\sum_{j=1}^{100} \text{OPT}^{(j)}(r, \boldsymbol{c}, \boldsymbol{B})\,.$$

For each $j$, we sample $S = 10,000$ contexts from the distribution $\nu$, and denote by $\widehat{\nu}_S^{(j)}$ the associated empirical distribution; we then solve numerically the problem (4) with $\nu$ replaced by this empirical distribution:

$$\text{OPT}^{(j)}(r, \boldsymbol{c}, \boldsymbol{B}) = \min_{\boldsymbol{\lambda} \geqslant 0} \ \mathbb{E}_{\boldsymbol{X} \sim \widehat{\nu}_S^{(j)}}\left[\max_{a \in \mathcal{A}}\Big\{r(\boldsymbol{X}, a) - \big\langle \boldsymbol{c}(\boldsymbol{X}, a) - \boldsymbol{B},\ \boldsymbol{\lambda}\big\rangle\Big\}\right]\,.$$

**Mixed policy knowing $\boldsymbol{\lambda}_{\boldsymbol{B}'}^\star$ but estimating $r$.** For the sake of illustration, we report the performance of a policy that would have oracle knowledge of $\boldsymbol{\lambda}_{\boldsymbol{B}'}^\star$, which is a finite-dimensional parameter that summarizes $\nu$, but would ignore $r$, i.e., the underlying logistic model. That is, this mixed policy would pick, at each round,

$$\max_{a \in \mathcal{A}}\Big\{\widehat{r}_{\delta, t-1}^{\text{ucb}}(\boldsymbol{x}_t, a) - \big\langle \widehat{\boldsymbol{c}}_{\delta, t-1}^{\text{lcb}}(\boldsymbol{x}_t, a) - \boldsymbol{B}',\ \boldsymbol{\lambda}_{\boldsymbol{B}'}^\star\big\rangle\Big\}\,,$$

(and would omit the $\boldsymbol{\lambda}$ update in Box B).

To compute (an approximation of) $\boldsymbol{\lambda}_{\boldsymbol{B}'}^\star$, we proceed 100 times as described below to compute estimates

$$\boldsymbol{\lambda}_{\boldsymbol{B}'}^{\star,(j)}\,, \qquad \text{which we average out into} \qquad \widehat{\boldsymbol{\lambda}}_{\boldsymbol{B}'}^\star = \frac{1}{100}\sum_{j=1}^{100} \boldsymbol{\lambda}_{\boldsymbol{B}'}^{\star,(j)}\,,$$

where we noted that the numerical values obtained for the $\boldsymbol{\lambda}_{\boldsymbol{B}'}^{\star,(j)}$ are rather similar. With the notation above, for each $j$, we sample $S = 10,000$ contexts from the distribution $\nu$, and solve

$$\boldsymbol{\lambda}_{\boldsymbol{B}'}^{\star,(j)} \in \arg\min_{\boldsymbol{\lambda} \geqslant 0} \mathbb{E}_{\boldsymbol{X} \sim \widehat{\nu}_S^{(j)}}\left[\max_{a \in \mathcal{A}}\Big\{r(\boldsymbol{X}, a) - \big\langle \boldsymbol{c}(\boldsymbol{X}, a) - \boldsymbol{B},\ \boldsymbol{\lambda}\big\rangle\Big\}\right]\,.$$

(These estimations are independent from the estimations used to compute the OPT values, i.e., we use different seeds.)

**PGD $\gamma$.** We refer to the Box B strategy as PGD $\gamma$, for which we report the performance for values $\gamma \in \{0.01, 0.02, 0.04, 0.05, 0.1\}$.

We also implemented the Box C strategy, with an alternative value of $M_{T,\delta,k}$, given that the one exhibited based on the theoretical analysis was too conservative, so that no regime break occurred. We resort to a value of the same order of magnitude:

$$M'_{T,\delta,k} = c\,d\sqrt{T\ln\big(T(k+2)\big)}\,,$$

for a numerical constant $c$ that we set to 0.01 in our simulations. We call this strategy PGD Adaptive in Figure 1 and Table 1.

### G.3   Outcomes of the simulations

We take $T = 10,000$ individuals (instead of $T = 1,000$ as in the code by Chohlas-Wood et al., 2021) and set initial 50 rounds as a warm start for strategies (mostly because of the logistic estimation). We were limited by the computational power (see Appendix G.4) and could only perform $N = 100$ simulations for each (instance of each) strategy. We report averages (strong lines in the graphs) as well as $\pm 2$ times standard errors (shaded areas in the graphs or values in parentheses in the table).

**Graphs.** In the first line of graphs in Figure 1, we report the average rewards (over the $N$ runs) of the strategy under scrutiny as a function of the sample size. More precisely, with obvious notation, we plot

$$t \mapsto \frac{1}{N}\sum_{n=1}^{N}\left(\frac{1}{t}\sum_{s=1}^{t} r_s^{(n)}\right).$$

We report the values of $\mathrm{OPT}(r,\boldsymbol{c},\boldsymbol{B})$ and $\mathrm{OPT}(r,\boldsymbol{c},\boldsymbol{B}')$ as dashed horizontal lines.

In the second and third lines of graphs in Figure 1, we report the average costs suffered; again, with obvious notation, we plot

$$t \mapsto \frac{1}{N}\sum_{n=1}^{N}\left(\frac{1}{t}\sum_{s=1}^{t}\mathbb{1}_{\{a_s^{(n)}=a_{\mathrm{ride}}\}}\right) \qquad \text{and} \qquad t \mapsto \frac{1}{N}\sum_{n=1}^{N}\left(\frac{1}{t}\sum_{s=1}^{t}\mathbb{1}_{\{a_s^{(n)}=a_{\mathrm{voucher}}\}}\right).$$

We include the average budget constraints $B_{\mathrm{ride}} = 0.05$ and $B_{\mathrm{voucher}} = 0.20$ as dashed horizontal lines.

Finally, the fourth line of the graphs in Figure 1 reports the fairness costs; we average their absolute values and draw, again with obvious notation,

$$t \mapsto \frac{1}{N}\sum_{n=1}^{N}\left(\frac{1}{4}\sum_{a\in\{a_{\mathrm{ride}},a_{\mathrm{voucher}}\}}\sum_{g\in\{0,1\}}\left|\frac{1}{t}\sum_{s=1}^{t}\Big(2\mathbb{1}_{\{a_s^{(n)}=a\}}\mathbb{1}_{\{\mathrm{gr}(\boldsymbol{x}_s^{(n)})=g\}} - \mathbb{1}_{\{a_s^{(n)}=a\}}\Big)\right|\right).$$

We include the fairness tolerance $\tau$ as a dashed horizontal line.

**Table.** We also report the performance of the strategies at the final round $T$ in following table, with $\pm 2$ times standard errors in parentheses. We note that the performance of the mixed policy is poor, in particular in terms of fairness costs, which is why we omitted it on the graphs, to keep them readable.

**Comments.** When $\gamma$ is well set, i.e., large enough (see the bound of Lemma 1), the PGD strategies control spending costs thanks to targeting $\boldsymbol{B}'$ instead of $\boldsymbol{B}$. We observe that for $\gamma = 0.01$, the PGD strategy does not control the rideshare costs, but for all larger values of $\gamma$, the associated strategies control the three costs considered. The average rewards achieved are coherent with the average spendings: the smaller the average spendings, the smaller the average rewards. There is some lag: the strategies tuned with $\gamma$ parameters in $\{0.04, 0.05, 0.1\}$ could use costly actions more. We also observe that fairness costs remain under the target limits.

The mixed policy does not success in controlling the costs, in particular, the fairness costs. While the optimal dual variables $\boldsymbol{\lambda}_{\boldsymbol{B}'}^{\star}$ summarize the distribution $\nu$, it seems that $\boldsymbol{\lambda}_{\boldsymbol{B}'}^{\star}$ has to be used with care: only with the exact values $r$ and $\boldsymbol{c}$, and not with estimates. On the contrary, the PGD strategies of Box B are more stable, as the dual variables are learned based also on the estimated reward and cost functions.

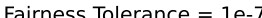

Figure 1: Performance of the PGD strategies.

Table 1: Performance of the strategies as computed at the final round $T$.

| | Average rewards | Rideshare costs | Voucher costs | Fairness costs |
|---|---|---|---|---|
| Fairness tolerance $\tau = 10^{-7}$ | | | | |
| $\mathrm{OPT}(r, \boldsymbol{c}, \boldsymbol{B})$ | 0.4688 (0.0002) | | | |
| $\mathrm{OPT}(r, \boldsymbol{c}, \boldsymbol{B}')$ | 0.4648 (0.0002) | | | |
| PGD $\gamma = 0.01$ | 0.4651 (0.0002) | 0.0519 (<0.0001) | 0.1984 (<0.0001) | 0.0006 (<0.0001) |
| PGD $\gamma = 0.02$ | 0.4613 (0.0002) | 0.0492 (<0.0001) | 0.1967 (0.0004) | 0.0004 (<0.0001) |
| PGD $\gamma = 0.04$ | 0.4571 (0.0002) | 0.0479 (<0.0001) | 0.1962 (0.0002) | 0.0004 (<0.0001) |
| PGD $\gamma = 0.05$ | 0.4554 (0.0002) | 0.0476 (<0.0001) | 0.1961 (0.0002) | 0.0003 (<0.0001) |
| PGD $\gamma = 0.1$ | 0.4502 (0.0002) | 0.0471 (<0.0001) | 0.196 (0.0002) | 0.0003 (<0.0001) |
| PGD Adaptive | 0.4581 (0.0002) | 0.0498 (0.0002) | 0.1971 (0.0002) | 0.0005 (<0.0001) |
| Mixed Policy | 0.4402 (0.0056) | 0.0499 (0.0058) | 0.1056 (0.017) | 0.0411 (0.0052) |
| Fairness tolerance $\tau = 0.025$ | | | | |
| $\mathrm{OPT}(r, \boldsymbol{c}, \boldsymbol{B})$ | 0.4731 (0.0002) | | | |
| $\mathrm{OPT}(r, \boldsymbol{c}, \boldsymbol{B}')$ | 0.4691 (0.0002) | | | |
| PGD $\gamma = 0.01$ | 0.4698 (0.0002) | 0.0518 (0.0002) | 0.1983 (<0.0001) | 0.0246 (0.0002) |
| PGD $\gamma = 0.02$ | 0.4663 (0.0002) | 0.0492 (<0.0001) | 0.1966 (0.0006) | 0.0242 (0.0002) |
| PGD $\gamma = 0.04$ | 0.4621 (0.0004) | 0.0478 (<0.0001) | 0.1958 (0.001) | 0.0223 (0.0004) |
| PGD $\gamma = 0.05$ | 0.4604 (0.0004) | 0.0476 (<0.0001) | 0.1955 (0.0014) | 0.0208 (0.0004) |
| PGD $\gamma = 0.1$ | 0.4538 (0.0002) | 0.0471 (<0.0001) | 0.1958 (0.0004) | 0.0128 (0.0004) |
| PGD Adaptive | 0.4634 (0.0002) | 0.0499 (0.0002) | 0.1972 (0.0002) | 0.0228 (0.0002) |
| Mixed Policy | 0.4466 (0.0054) | 0.0566 (0.0052) | 0.1053 (0.0164) | 0.0473 (0.0054) |

Finally, we note that in our experiments, the regimes in PGD Adaptive strategy typically covered range from $k = 0$ (corresponding to $\gamma_0 = 1/\sqrt{T} = 0.01$) to $k = 2$ (corresponding to $\gamma_2 = 2^2/\sqrt{T} = 0.04$). The PGD Adaptive strategy performs well and is (only) outperformed by the PDG strategy with a fixed $\gamma = 0.02$ (of course difficult to pick in advance). In particular, the PGD Adaptive strategy controls costs, and does so by switching to larger step sizes when needed.

## G.4 Computation time and environment

As requested by the NeurIPS checklist, we provide details on the computation time and environment. Our experiments were ran on the following hardware environment: no GPU was required, CPU is 3.3 GHz 8 Cores with total of 16 threads, and RAM is 32 GB 4800 MHz DDR5. We ran 100 simulations with 10 different seeds on parallel each time. In the setting and for the data described above, the average time spend on each algorithm for a single run was inferior to 10 minutes.

