# OpenReview forum: "Small Total-Cost Constraints in Contextual Bandits with Knapsacks, with Application to Fairness"
_NeurIPS.cc/2023/Conference — NeurIPS 2023 poster_

### Official Review · Reviewer_3rBq · 2023-07-02

**Soundness:** 2 fair
**Presentation:** 2 fair
**Contribution:** 2 fair
**Rating:** 4
**Confidence:** 3

**Summary:**

This paper deals with the CBwK problem with a fairness constraint of equalized average costs between groups. The authors propose a dual algorithm: PGD for CBwK (with adaptive stepsize) and provides regret bounds.

**Strengths:**

The problem statement is pretty clear and the related work is relatively thoroughly presented. The limitations are clearly and explicitly stated.

**Weaknesses:**

The algorithm (PGD) seems to be established in the online optimization and bandit literature, and the proofs also seem standard. Also the regret bound is only superior under some circumstances (e.g. not working for soft constraints, loose bound when for example null action has null reward). My major concern is on the novelty. I think to stand out from the existing papers, the authors may want to specify a more narrow scheme and claim the contributions.

**Questions:**

The authors make an assumption that the respective proportions $\gamma_g$ is known. Could you please discuss the limitations of such assumption? And what analytical convenience this assumption buys you? Are there any less restrictive alternatives?

**Limitations:**

Yes

---

> ### Author Rebuttal · Authors · 2023-08-03
>
> We thank the reviewer for the reading but would respectfully disagree with the evaluation.
>
> For the PGD algorithm (and the proofs) being standard: Yes, the PGD approaches in the CBwK litterature are standard, as we acknowledge and as other rewievers point out (see, e.g., Agrawal and Devanur 2016). Our main point, however, is the adaptive step-size tuning of PGD. We carefully explain in lines 229-244 why and how previous contributions provided suboptimal tunings for PGD, leading to the impossibility of going beyond the T^{3/4} total-cost constraint. Section 3.2 then provides our main contribution: a careful adaptive tuning of the hyperparameters of PGD, allowing to total-cost constraints as small as 1/\sqrt{T} up to logarithmic factors. We agree and acknowledge that its analysis partially relies on some known building blocks.
>
> For the regret bound being only superior under some circumstances (e.g., not working for soft constraints, loose bound when for example null action has null reward): We would like to discuss more in depth this comment, which we object to.
> - First, Section 3.3 explains in detail why our bounds in terms of the norms of \lambda^* are sharper than the usual OPT/min B bounds of the literature. This new form of the bound is important in view of the lower bounds alluded at in lines 330-331 and because it does not require the existence of a null-cost action. So, our bounds do provide improvement upon the literature.
> - Second, the literature was only (and rightfully) interested in hard constraints so far. We introduce in appendix the concept of soft constraints, as we noticed some new phenomena in this setting compared to the usual hard-constraints setting. We actually deal with them in some optimal manner, but with a different algorithm---a primal strategy, see Section F in the supplementary material. Dealing with such (non-classical) soft constraints is an open problem for the PGD strategy considered in the main body. It is also an open problem for all previous strategies in the literature (that did not cover this aspect). To sum up, we have identified tackling soft constraints as a possible extension to CBwK theory, not addressed so far in the literature, and for which we provide partial results (with a primal algorithm). This, in our opinion, cannot be considered as a weakness, on the contrary.
> - Third, the 'loose bound’ is the classical bound in the literature, and it is generally optimal. We mention in Limitation 2 that however, faster rates should be possible in some very specific settings---a possibility which was not identified at all in the literature. The precise study of this very specific, yet interesting, setting would require some space not available in a NeurIPS format. This new insight should then be seen as an open problem stimulating future research, rather than a weakness of our contribution.
>
> For not claiming the contributions: We summarized them in lines 48-59 in Section 1.
>
> For the respective proportions gamma_g being known: In practice (at least in the example by Chohlas-Wood et al. [2021]) this should be a mild restriction, that is not unheard of in the fairness literature. It amounts to having a reasonable knowledge of the breakdown of the population into subgroups. We use this assumption in line 156: the cost function is then known, as required. Replacing T \gamma in line 154 by the empirical number of individuals in the group induces many additional technicalities, while it only presents a limited practical interest, given that the fairness literature has little issues with assuming a reasonable knowledge of the breakdown of the population into subgroups.

---

> > ### Comment · Reviewer_3rBq · 2023-08-17
> >
> > Thank you for your thorough rebuttal. I believe my understanding of this paper is at a reasonable level. And my major concern stays as in the review: that the contribution seems incremental to me (in terms of the algorithm especially, as well as the analysis). Nevertheless I in general agree this paper is well presented, and would like to increase my score to borderline, and leave it to the AC's discretion.

---

### Official Review · Reviewer_Hs7n · 2023-07-06

**Soundness:** 3 good
**Presentation:** 4 excellent
**Contribution:** 3 good
**Rating:** 7
**Confidence:** 4

**Summary:**

The problem studies the contextual bandit with knapsack problem with contexts coming from a continuous set, signed costs, and under the assumption that expected reward and cost functions can be uniformly estimated. The learner aims at maximizing their cumulative rewards while guaranteeing constraints of the form $\sum_{t=1}^T c_t\le TB$, where the rhs $TB$ may be as small as $T^{1/2+\epsilon}$. The proposed algorithm is based on a projected gradient descent scheme with adaptive step size. The author discuss applications of the algorithm to problems related to fairness.

**Strengths:**

The authors present an interesting set of results. The model is well motivated and presents some interesting complications wrt the traditional BwK framework. The paper is generally well written and mostly clear.

**Weaknesses:**

I don't have any major concern.

Some minor comments:

- Line 113: "go go"
- It would be good to mention closer to Assumption 1 that, in order to derive guarantees, you need  strict feasibility.
- Paragraph starting at line 158 lacks convincing examples.
- Assumption 2: why are you defining $\beta$ there?

**Questions:**

Can you provide some further details about the statements of the paragraph starting at line 158 (e.g.  what scenarios do you have in mind when you say "typically")?

**Limitations:**

Yes, limitation are adequately discussed.

---

> ### Author Rebuttal · Authors · 2023-08-03
>
> We thank the reviewer for the careful reading and fully agree with the evaluation.
>
> For the statements of the paragraph starting at line 158, including twice the word "typically": We agree that this paragraph will benefit from some rewriting. For the total spendings B_total, we had in mind the example by Chohlas-Wood et al. [2021], where a constant fraction of the T individuals may benefit from some costly action, hence a linear total budget T B_total, or, put differently, B_total is larger than some positive constant. This is better detailed on page 31 in appendix. For the fairness threshold \tau, we meant that in some cases we would ideally have \tau = 0, but due to central-limit-theorem fluctuations, this is not possible and a \tau with a minimal value of the order of 1/\sqrt{T} has to be considered. In general, Chohlas-Wood et al. [2021] want to ensure some fair treatment among groups, so would pick some \tau that is rather small (though would be ready to have it larger than 1/\sqrt{T} as this entails some positive discrimination and is effective in practice). All in all, we would entirely rewrite this paragraph, by suppressing all occurrences of 'typically' and summarize the ideas above.

---

> > ### Comment · Reviewer_Hs7n · 2023-08-21
> >
> > Thank you for your response. I confirm my positive score.

---

### Official Review · Reviewer_EATd · 2023-07-06

**Soundness:** 3 good
**Presentation:** 3 good
**Contribution:** 3 good
**Rating:** 6
**Confidence:** 3

**Summary:**

This paper studied contextual bandit problems with knapsacks (CBwK). Under this setting, at each time step, a scalar reward is obtained and vector-valued costs are suffered.  The agent aims to maximize the cumulative rewards while ensuring that the cumulative costs are lower than some predetermined cost constraints.
In this setting, total cost constraints had so far to be at least of order $T^{3/4}$, where $T$ is the number of time steps, and were even typically assumed to depend linearly on $T$.
This paper imposed a fairness constraint by applying the CBwK model and introduced a dual algorithm based on projected-gradient-descent updates in order to deal with total-cost constraints of the order of $\sqrt{T}$ up to poly-logarithmic terms.


================

The score is kept unchanged.

**Strengths:**

1. This paper is in general well organized and presents a detailed literature review.
2. The motivation of considering faireness with the CBwK setting is explained and convincing.
3. Beyond the strengths of the proposed algorithms, the authors also listed the limitations of this work in Section 4.

**Weaknesses:**

1. My major concern is that no numerical results are presented in this work.

**Questions:**

I suggest to compare the performance of the proposed algorithm and some other algorithms with numerical experiments.

---

> ### Author Rebuttal · Authors · 2023-08-03
>
> We thank the reviewer for the careful reading and generally agree with the evaluation.
>
> For the weakness raised: We only provide (very) preliminary simulations (see pages 33-35), rather illustrating how to successfully deal with the fairness constraints. What these simulations do not illustrate, however, is how useful adaptive step-sizes are: this is because total-cost constraints are never broken therein during the regime k=0. We were too short on time to provide improved simulations investigating the usefulness of the adaptive step-sizes and offering comparisons to other algorithms. We will provide such simulations in the final version of the paper (including a summary thereof in the main body).

---

> > ### Comment · Reviewer_EATd · 2023-08-20
> >
> > Thanks for your response.

---

### Official Review · Reviewer_aT3N · 2023-07-07

**Soundness:** 3 good
**Presentation:** 4 excellent
**Contribution:** 3 good
**Rating:** 6
**Confidence:** 4

**Summary:**

The paper considers the problem of general contextual bandits with knapsacks in the regime where Omega(sqrt{T}) <= B <= O(T^{3/4}). The paper provides a new algorithm that is based on prior methodologies of primal-dual algorithm, but instead of relying on them as black-boxes, updates the dual variables using adaptive step-sizes (for the same algorithm using fixed step-size reduces to prior works (e.g., Agrawal and Devanur 2016). The main contribution of this paper is to notice this fact, show a scheduling scheme for the adaptive step-size and prove regret bounds in this regime.

**Strengths:**

+ The proposed algorithm is clean and builds on understanding from prior works. Notion of using adaptive step-sizes is common in the optimization literature and bringing that understanding/techniques to contextual bandits with knapsacks is novel. Combining this with existing primal-dual approach is natural.

+ The paper is well-written and easy to understand. The paper also provides problem dependent lower-bounds and surprisingly (for modern papers) honestly discusses the limitations of the algorithm (e.g., not being able to prove a worst case regret bound, when B is actually large).

**Weaknesses:**

- Overall, I do not find the fairness example particularly motivating or the right example. The paper is useful for the mathematical/algorithmic insights. Not sure if this example adds any value to the paper.

- One place where this paper could have really shone is to actually run simulations and show that even in practice the adaptive step-size actually improves the regret in the claimed range. In particular, it would tease out the effect of inability to analyze prior works better vs needing new algorithmic technique (such as adaptive step-size) conclusively.

**Questions:**

Nothing in particular, that could help change my mind. Please see my suggestions above on simulations.

**Limitations:**

This is a mathematical paper and no societal impact.

---

> ### Author Rebuttal · Authors · 2023-08-03
>
> We thank the reviewer for the careful reading and generally agree with the evaluation.
>
> For the fairness example: It was our own motivating example for developing a CBwK theory able to handle small total-cost constraints. (Chohlas-Wood et al., 2021, who first introduced this example, could not propose regret bounds, see lines 62-73.) Admittedly, better examples could be provided and it is clear that breaking the artifical T^{3/4} barrier in the literature was an interesting problem per se.
>
> For simulations: Indeed. We only provide (very) preliminary simulations (see pages 33-35), rather illustrating how to successfully deal with the fairness constraints. What these simulations do not illustrate, however, is how useful adaptive step-sizes are: this is because total-cost constraints are never broken therein during the regime k=0. We were too short on time to provide improved simulations investigating the usefulness of the adaptive step-sizes and offering comparisons to other algorithms. We will provide such simulations in the final version of the paper (including a summary thereof in the main body).

---

### Author Rebuttal · Authors · 2023-08-03

We generally agree with the evaluations by Reviewers aT3N - EATd - Hs7n but respectfully disagree with the evaluation by Reviewer 3rBq. We explain below in detail the main two issues we disagree with:
- The main algorithmic contribution is not the PGD approach per se, but its adaptive step-size tuning (see lines 229-244), for which we present a novel contribution, namely, handling the total-cost regime Omega(sqrt{T}) <= T B <= O(T^{3/4}).
- Our regret bound is sharper than existing bounds in the literature (with some norm of \lambda^* replacing the larger OPT/min B term, see Section 3.3) and does not require a null-cost action. That being said, we have identified possible improvements and extensions (listed in Section 4) that are not addressed in the literature, and that we propose as open problems for future research.

---

### Decision · Program_Chairs · 2023-09-21

**Decision:**

Accept (poster)

**Comment:**

The paper considers the problem of general contextual bandits with knapsacks where the the budget is considerably much smaller than the time horizon. This is a new and challenging setting as the underlying offline knapsack problem is also much harder. In particular, for the case the budget is of magnitude of T and the costs are bounded above by constants, the underlying offline (unbounded) knapsack problem has a computational complexity in P (see the work of Pisinger for more details). Note that this was the common setting that ALL the prior work in knapsack bandits relied on.

The new setting moves away from this rather restrictive assumption and the authors provide a new algorithm that is based on prior methodologies of primal-dual algorithm, but instead of relying on them as black-boxes, updates the dual variables using adaptive step-sizes (for the same algorithm using fixed step-size reduces to prior works (e.g., Agrawal and Devanur 2016). The main contributions of this paper are: (i) to design a new scheduling scheme for the adaptive step-size;  and (ii) prove regret bounds for this setting.

There was a disagreement between the reviewers, as one stated that this paper is rather incremental. I strongly disagree with this statement. As someone very familiar with knapsack bandit literature, and claim that bandits with small total cost constraints is a very new and challenging setting (personally I have been working on this for a few years now, without much success). I am very glad that the authors has managed to provide a first non-trivial result in this open question. Therefore I strongly recommend to accept this paper.